# Genome-scale metabolic modeling of *Aspergillus fumigatus* strains reveals growth dependencies on the lung microbiome

Mohammad H. Mirhakkak[1,12], Xiuqiang Chen [1,12], Yueqiong Ni[1], Thorsten Heinekamp[2], Tongta Sae-Ong [1], Lin-Lin Xu[1], Oliver Kurzai[3,4,5], Amelia E. Barber[6], Axel A. Brakhage [2,7], Sebastien Boutin[8,9], Sascha Schäuble [1] ✉ & Gianni Panagiotou[1,10,11] ✉

*Aspergillus fumigatus*, an opportunistic human pathogen, frequently infects the lungs of people with cystic fibrosis and is one of the most common causes of infectious-disease death in immunocompromised patients. Here, we construct 252 strain-specific, genome-scale metabolic models of this important fungal pathogen to study and better understand the metabolic component of its pathogenic versatility. The models show that 23.1% of *A. fumigatus* metabolic reactions are not conserved across strains and are mainly associated with amino acid, nucleotide, and nitrogen metabolism. Profiles of non-conserved reactions and growth-supporting reaction fluxes are sufficient to differentiate strains, for example by environmental or clinical origin. In addition, shotgun metagenomics analysis of sputum from 40 cystic fibrosis patients (15 females, 25 males) before and after diagnosis with an *A. fumigatus* colonization suggests that the fungus shapes the lung microbiome towards a more beneficial fungal growth environment associated with aromatic amino acid availability and the shikimate pathway. Our findings are starting points for the development of drugs or microbiome intervention strategies targeting fungal metabolic needs for survival and colonization in the non-native environment of the human lung.

Fungal infections are an emerging and costly concern for human health and health care[1,2]. The common mold *Aspergillus fumigatus* is essential for environmental decomposition but poses a serious threat to hospitalized patients, particularly those who are immunocompromised or have pulmonary diseases such as cystic fibrosis[3,4]. Annually, more than 1 million people have invasive aspergillosis (IA), a systemic multi-organ affecting disease starting by *A. fumigatus* infecting the lung, and 3 million have chronic pulmonary

[1]Department of Microbiome Dynamics, Leibniz Institute for Natural Product Research and Infection Biology (Leibniz-HKI), 07745 Jena, Germany. [2]Department of Molecular and Applied Microbiology, Leibniz Institute for Natural Product Research and Infection Biology (Leibniz-HKI), 07745 Jena, Germany. [3]Institute for Hygiene and Microbiology, University of Würzburg, 97080 Würzburg, Germany. [4]Research Group Fungal Septomics, Leibniz Institute of Natural Product Research and Infection Biology (Leibniz-HKI), 07745 Jena, Germany. [5]National Reference Center for Invasive Fungal Infections (NRZMyk), Leibniz Institute of Natural Product Research and Infection Biology (Leibniz-HKI), 07745 Jena, Germany. [6]Junior Research Group Fungal Informatics, Institute of Microbiology, Friedrich-Schiller-University Jena, 07745 Jena, Germany. [7]Institute of Microbiology, Friedrich Schiller University Jena, 07745 Jena, Germany. [8]Department of Infectious Diseases and Microbiology, University of Lübeck, 23562 Lübeck, Germany. [9]Translational Lung Research Center Heidelberg (TLRC), German Center for Lung Research (DZL), University of Heidelberg, 69120 Heidelberg, Germany. [10]Department of Medicine and State Key Laboratory of Pharmaceutical Biotechnology, University of Hong Kong, Hong Kong, China. [11]Friedrich Schiller University, Faculty of Biological Sciences, Jena 07745, Germany. [12]These authors contributed equally: Mohammad H. Mirhakkak, Xiuqiang Chen ✉ e-mail: Sascha.Schaeuble@leibniz-hki.de; Gianni.Panagiotou@leibniz-hki.de

aspergillosis. Both conditions have high mortality rates and diagnosis remains challenging (https://gaffi.org/why/fungal-disease-frequency/, June 2022). In addition, *A. fumigatus* contributes substantially to fatal disease progression in chronic obstructive pulmonary disease, which appears to have a much higher prevalence than previously estimated[5], while *A. fumigatus* is related to as many as half of the worldwide cystic fibrosis cases[6].

The distinct characteristics that *A. fumigatus* isolates possess to cope with external stresses or accessible nutrient profiles in challenging environments such as the human lung remain largely unknown. Recently, we explored the genetic diversity of *A. fumigatus* and found a remarkably low fraction of core genes shared by all members of the species (69% of total genes identified)[7]. How the genetic diversity of *A. fumigatus* influences phenotypic and metabolic heterogeneity, and particularly the ability to thrive in the non-native niche of the human lung, has not been addressed.

One promising approach to studying the metabolic capabilities and growth dependencies of pathogens is genome-scale metabolic model (GEM) reconstruction and analysis[8]. We previously applied GEM analysis to reveal gut microbiome species that influence colonization levels of the opportunistic fungal pathogen *Candida albicans*[9]. The exponentially increasing number of available genome sequences makes the reconstruction of multistrain GEMs possible. The first multistrain-GEM collection of *Escherichia coli* enabled the definition of strain-specific adaptation to nutrition availability and the prediction of nutritional auxotrophies in some strains[10]. Protocols and databases were consequently updated to allow for bacterial GEM reconstruction at strain-level resolution[11,12], although reconstructions of multistrain-GEMs remain to be explored in eukaryotes.

In this study, we provide multistrain-GEM reconstruction using *A. fumigatus* as a fungal model organism. By defining metabolic differences among 203 environmental and 49 clinical strain-specific GEMs we identified metabolic reactions that differ between the two populations. Subsequently, we performed shotgun metagenomics on sputum from 40 cystic fibrosis patients before and after they were diagnosed with an *A. fumigatus* colonization. Based on computationally defined metabolic output of the lung microbiome, we propose that the presence of *A. fumigatus* shapes the metabolic landscape of the lung microbiome to be favorable for fungal growth. Resolving the impact of genetic diversity on *A. fumigatus* metabolism is important to extending our understanding of adaptation mechanisms that likely involve aromatic amino acid metabolism and the shikimate pathway for ultimately guiding the development of new antifungal therapies.

## Results
### Reconstruction of a comprehensive *Aspergillus fumigatus* pan-GEM
To create a template for strain-specific GEM design, we generated a comprehensive pan-GEM for *A. fumigatus* metabolism (Fig. 1a). To start, we combined two available draft reconstructions for *A. fumigatus* with seven automatically derived draft reconstructions for different *Aspergillus* species (see Methods for details)[13,14]. This approach allowed us to acquire as many *Aspergillus*-associated reactions as possible in the core metabolism of *A. fumigatus* (i.e., metabolic reactions present in all strains). It also allowed us to acquire a more comprehensive catalog of optional accessory metabolic reactions by defining strain subset diversity, which enabled subsequent strain-specific gap-filling curation (Fig. 1a). The first draft model comprised 7606 reactions (of which 3233 were responsible for metabolite exchange with the simulated environment) and 3578 metabolites, which we reduced to 3621 (Fig. 1b) during the curation steps described further below.

Next, we adapted 62 metabolic components based on fungal- and *A. fumigatus*- specific literature to create the biomass objective function essential for simulating *A. fumigatus* growth rates (Methods)[15]. The largest fractions of the derived biomass function included

carbohydrates and proteins (43% and 30%, respectively). Additional essential components included lipids, DNA, and energetic co-factors (Fig. 1c, Supplementary Data S1).

Subsequently, we screened available *A. fumigatus* gene information relevant for metabolism and added 1444 unique genes (Figs. 1b) and 2003 corresponding gene-to-reaction rules for metabolic reactions, as defined by KEGG (https://www.kegg.jp/) or MetaCyc (https://metacyc.org/, Methods). The remaining 2370 metabolic reactions (excluding exchange reactions) could not be mapped to any gene in our pan-GEM draft model and were removed from the generic pan-GEM. These reactions were retained for subsequent strain-specific refinement steps, which require accessory information e.g., for gap-filling of fragmented metabolic pathways (Fig. 1a). Concurrently, we incorporated reaction-to-pathway association information from both KEGG and MetaCyc. The largest pathway categories included amino acids and carbohydrates (Fig. 1d). For the 2003 metabolic reactions with gene annotation, we predicted nine compartments in our pan-GEM using WoLF PSORT (Fig. 1e, Methods)[16]. In parallel, we identified and resolved erroneous energy-generating cycles[17] by correcting or removing thermodynamically implausible reactions, for example, that diminished free energy dissipation. After these steps, we analyzed again the consistency in our pan-GEM and identified 210 blocked reactions by flux variability analysis (FVA)[18] based on relaxed flux bounds of exchange reactions.

For the final curation of our pan-GEM, we generated phenotypic growth data for *A. fumigatus* wild-type (Af293 strain) and four mutant strains affecting nitrogen or carbon metabolic components, and adapted the model to publicly available gene essentiality information[19] (Supplementary Data S2). The initial agreement of our pan-GEM to our metabolite-specific growth data was at most 66% on average and required improvement to enable accurate metabolic predictions (Fig. 1f). To optimize the simulation accuracy of our pan-GEM, we manually resolved incompatibilities among our growth data, available gene essentiality data, and our in silico model predictions (Methods). Of note, we did not observe lysine-dependent growth cessation with our phenotypic microarray data of the *lysF* mutant strain (Supplementary Data S2) suggesting a media influence on the environment for growth and virulence[20]. These curation efforts improved growth simulation accuracy from 58% to 84% for all tested carbon sources and improved nitrogen growth simulation accuracy from 55% to 85% (Fig. 1f). The pan-GEM achieved 79% and 82% compatibility for, respectively, the tested phosphorus and sulfur sources (Fig. 1f, Methods). This final model reached 75% agreement with the available gene essentiality data (Fig. 1g). Our final *A. fumigatus* pan-GEM comprised 1,444 unique genes, 3,621 reactions, and 4,046 metabolites distributed across 9 compartments. Of these, 2,798 metabolic reactions and 1,940 metabolites were unique in our pan-GEM across all 9 compartments (Fig. 1b).

Finally, we tested how well our pan-GEM predicted oxygen-dependent growth[21]. After calibrating our model to normoxic growth conditions (0.013 mmol/grams dry weight [grDW]/hr, Fig. 1h), we were able to accurately capture hypoxic growth (predicted 0.010 compared to measured 0.011 mmol/grDW/hr). The same model also predicted the magnitude of experimentally derived secretion rates for acetate (for predicted and measured values, respectively, 0.009 vs. 0.004 for normoxic conditions; 0.020 vs. 0.015 for hypoxic conditions, all mmol/grDW/hr). Measured ethanol and lactate levels were also very low (0.003 mmol/grDW/h or lower), while the predicted theoretical yield by our simulations was 0.008 mmol/grDW/h or lower (Supplementary Data S2, Methods). In summary, our refined pan-GEM was able to recapture experimentally assessed oxygen-dependent growth data and predicted secretion rates of assessed metabolites that were comparable to the publicly available growth data. For the remainder of our analysis, we assumed normoxic growth conditions unless otherwise noted.

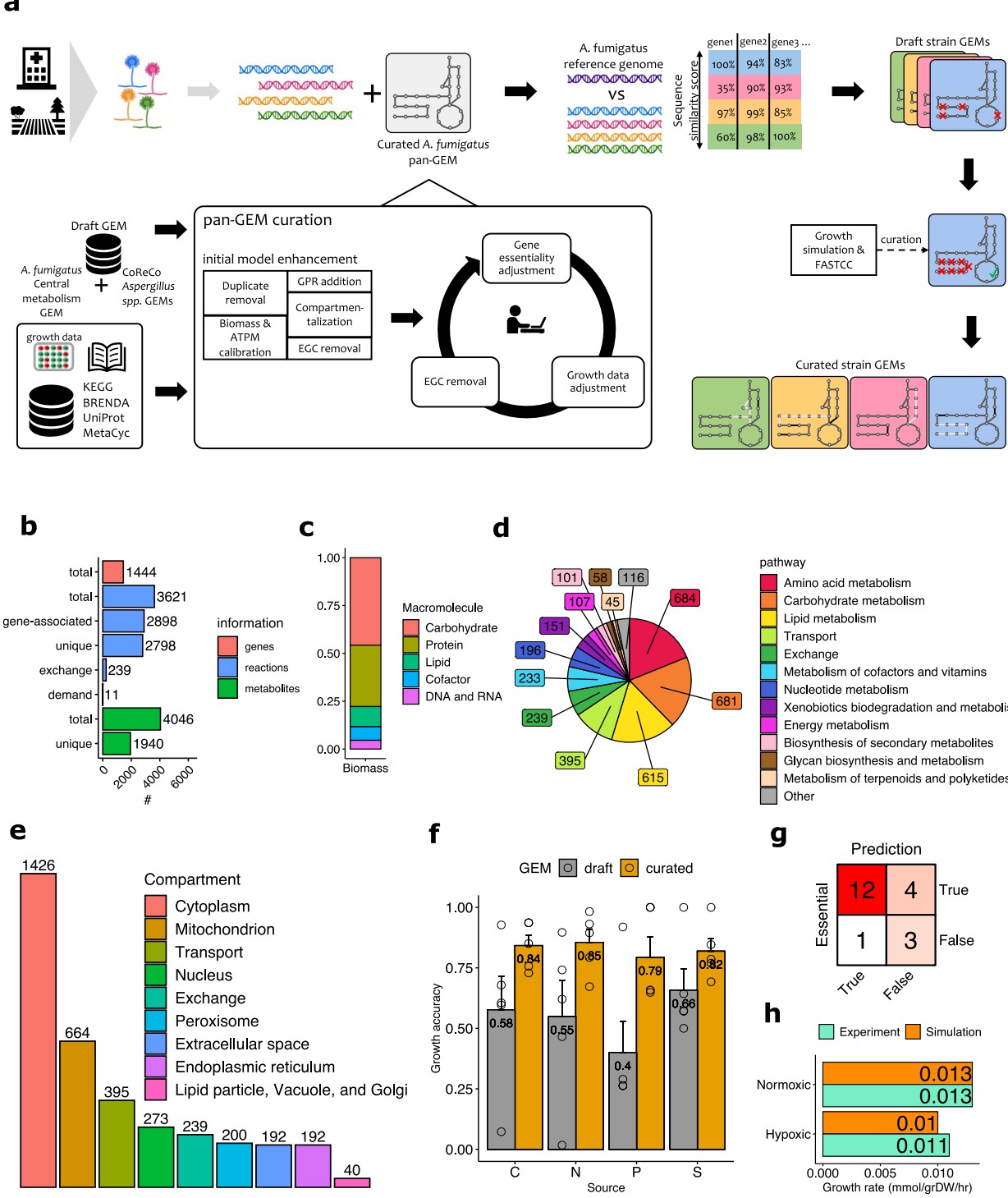

**Fig. 1 | General reconstruction workflow and *A. fumigatus* pan-genome-scale metabolic model (GEM) statistics. a** Workflow for *A. fumigatus* strain-specific GEM reconstructions. Colors indicate strains and associated metabolic models. **b**–**g** Characteristics of pan-GEM reconstruction for *A. fumigatus*. **b** Counts of pan-GEM components for included genes, reactions, and metabolites. **c** Contribution of macromolecules in one unit of biomass (Supplementary Data S1). **d** Distribution of pan-GEM reactions across major pathway categories (Supplementary Data S9). **e** Distribution of pan-GEM reactions across nine compartments (Supplementary Data S9). **f** Growth prediction accuracy of pan-GEM for *A. fumigatus* wild-type (Af293) and four mutant strains using phenotypic microarray data (*n* = 5 in total, bars show mean and standard error of mean, Supplementary Data S2). C: carbon, N: nitrogen, P: phosphorus, S: sulfur. **g** Confusion matrix of pan-GEM accuracy in predicting the essentiality of 20 genes according to the literature (see Results and Methods). **h** Experimental values compared to simulated growth rate values under normoxic and hypoxic conditions (Supplementary Data S2 has experimental and simulated secretion values). Source data for Fig. 1b–h are provided in the Source Data file.

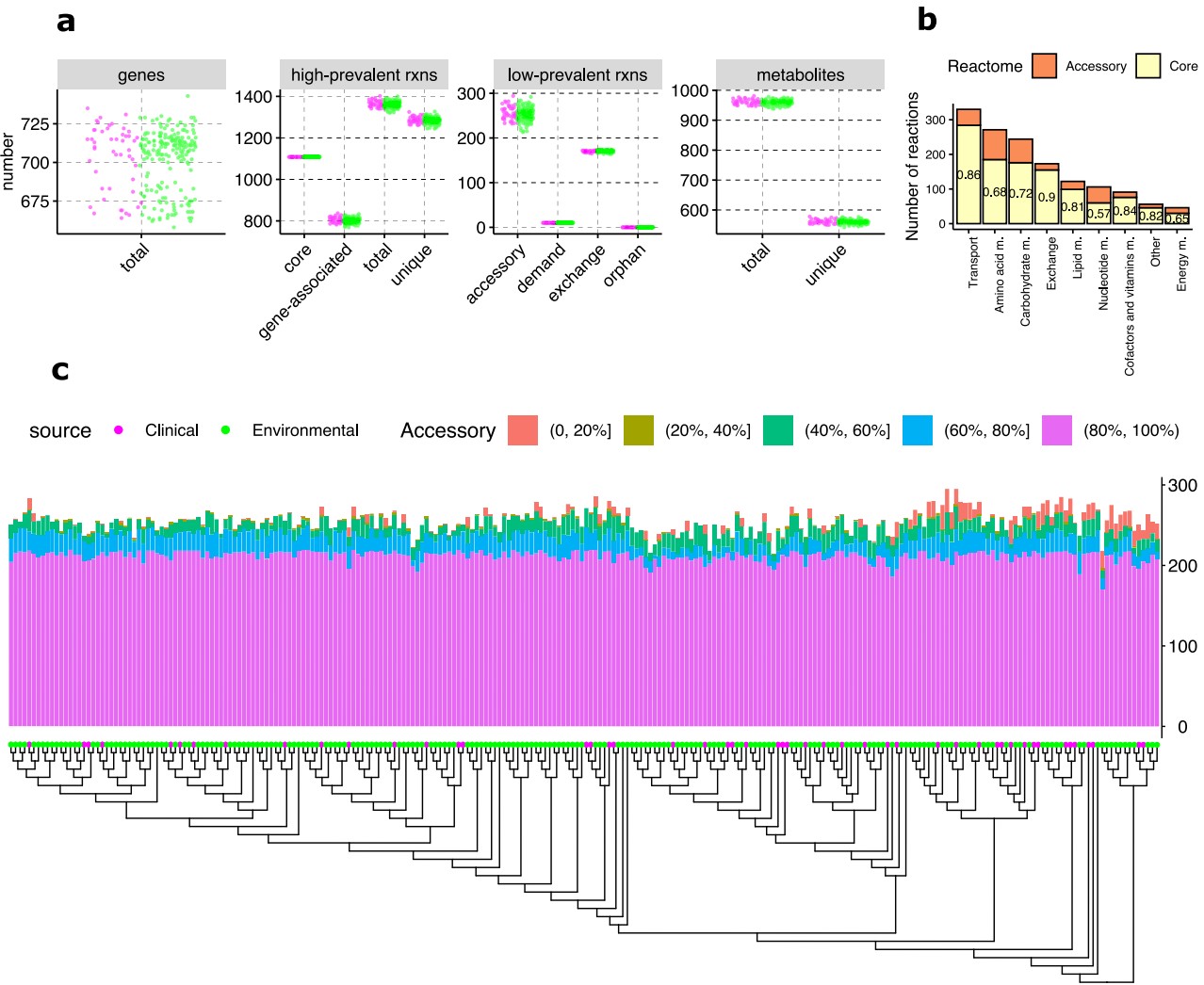

**Fig. 2 | Core and accessory metabolic characteristics of *A. fumigatus* strain-specific genome-scale metabolic models (GEMs).** Core and accessory metabolic content was determined for 252 unique *A. fumigatus* strains of environmental and clinical origin. **a** Number of core and accessory genes, reactions (rxns), and metabolites for reconstructed strain-specific GEMs. **b** Summary of the core and accessory reactome for higher-level metabolic pathway categories. Pathway categories are according to KEGG pathway definition (https://www.kegg.jp/kegg/pathway.html). **c** Distribution of the accessory reactome across strain models. Indicated percentage ranges correspond to accessory reaction presence across strain-specific GEMs. Genetic relationship as originally published is indicated[7]. Source data for Fig. 2 are provided in the Source Data file.

## *A. fumigatus* strains show notable accessory reaction content

Using a genomic dataset of 252 *A. fumigatus* strains from Germany (203 environmental and 49 clinical strains) from our previous study addressing the global diversity of *A. fumigatus*[7], we mapped strain-specific gene profiles to the reference pan-GEM and subsequently derived strain-specific GEMs (Supplementary Data S3). For all strain-specific GEMs, we ensured viable growth was predicted in minimal media with glucose as the carbon source by identifying and resolving minimal sets of essential reactions[22] and crosschecking against blocked reactions with FASTCC (Methods)[23]. Model size varied in the different *A. fumigatus* strains from 1321 to 1402 reactions (mean 1361).

Although all strain-specific GEMs were derived from *A. fumigatus*, we found a low number of core metabolic components shared by all GEMs (Fig. 2a). In line with the considerably high genomic diversity of this organism[7], only 475 metabolic genes (61.7%) and 1108 metabolic reactions (76.9%) were shared by all strain-specific GEMs, resulting in a large degree of metabolic variation across all GEMs (296 accessory genes and 338 accessory reactions). Most relevant accessory content was involved in nucleotide, energy (including oxidative

phosphorylation and nitrogen metabolism), and amino acid metabolic pathways (Fig. 2b). For all reactions in the strain-specific GEMs for these pathways, conservation across all strain models was only 57% for nucleotide, 65% for energy, and 68% for amino acid metabolism, demonstrating considerable metabolic pathway variation among strains (Fig. 2b). The majority of the accessory content (60% for genes, 63% for reactions, Table 1, Fig. 2c) was shared by more than 80% of all strain-specific GEMs. We previously observed that one genetic lineage of *A. fumigatus* possessed significantly fewer accessory genes than the others, including notably fewer metabolic accessory genes[7] (Supplementary Fig. S1). In contrast, metabolic reaction content in the strain-specific GEMs did not show a reduced number of metabolic reactions in this lineage, demonstrating the presence of redundancy among metabolic accessory genes (Fig. 2c). Finally, a small, but notable number of reactions appeared in 40% or fewer of all strain-specific GEMs (Table 1, Fig. 2c), including mostly reactions of amino acid metabolism, but also of lipid and nucleotide metabolism including nitrogen-dependent chorismate pyruvate-lyase or nicotinamidase and acyl-CoA-dependent acyltransferases. The large variability among

**Table 1 | Number of accessory genes and reactions across all strain-specific GEMs**

| Occurrence (in %) | Genes | | Reactions | |
|---|---|---|---|---|
| | Mean | Standard deviation | Mean | Standard deviation |
| [1–20] | 1.9 | 1.8 | 4.7 | 7.7 |
| [21–40] | 6.9 | 2.5 | 1.3 | 1.2 |
| [41–60] | 16.7 | 3.2 | 17.4 | 6.0 |
| [61–80] | 17.8 | 2.4 | 20.2 | 6.3 |
| [81–99] | 296.9 | 8.1 | 219.3 | 6.8 |

Based on 338 accessory reactions, we categorized accessory genes and reactions by percentage of occurrence across all strain-specific GEMs. For each range of percent (e.g., 1–20% of strains), we show the predicted mean number of genes and reactions, with standard deviations. Source data are provided in the Source Data file.

strains in amino acid metabolism was confirmed by cultivation and targeted metabolomics profiling of 20 *A. fumigatus* strains (Supplementary Data S4).

Taken together, our 252 strain models showed notable accessory content and therefore potential metabolic diversity among the strains as well as metabolic robustness despite reduced numbers of accessory metabolically relevant genes. All GEMs of our strain collection are in the BioModels repository (ID MODEL2211100001)[24].

## Metabolic activity of 21 reactions differentiates between environmental and clinical strain-specific GEMs

Calculating the pairwise Jaccard index showed that strain-specific GEMs differed by 15% or less (Fig. 3a). Neither accessory reaction information nor Jaccard distance discriminated between environmental and clinical strains for metabolic capabilities (Figs. 2c, 3a). However, we identified eight metabolic reactions present primarily in either environmental or clinical strain-specific GEMs that, when considered without other reactions, were able to significantly differentiate the two strain origins (Fisher's exact test, $p \leq 0.05$, Fig. 3b). In agreement with the statistical significance of these eight reactions, decision tree machine learning (ML) using the presence or absence of these metabolic reactions, as well as the capability of the strains to grow on different minimal media compositions, required only a few steps to correctly categorize 216 of 252 strains (86%, Fig. 3c). Notably, the presence of chorismate lyase alone allows to categorize 93% of all strain-specific GEMs correctly. Chorismate lyase activity is linked to differential activity in the shikimate pathway, which is associated with virulence in *A. fumigatus*[25,26]. Combining the ability to convert chorismate and glutamine to anthranilate, pyruvate, and glutamate, with amino acid and energy metabolism-associated conversions of methionine, succinate, or tryptamine, and the ability to take up and grow on aspartic acid, appeared sufficient for strain origin classification. Specifically, the ability to add sulfur to methionine as well as the absence of the ability to convert selenocystathione to selenocysteine or tryptamine to Indole-3-acetaldehyde appeared a characteristic of environmental strains, which in part was not present in clinical strains (Fig. 3c). This may hint to altered thioredoxin levels, which have been linked to the fungus' redox homeostasis before[27]. These reactions yielded metabolic discriminators that were complementary to the sole presence/absence statistical analysis of metabolic reactions in our strain-specific GEM collection (Fig. 3a, b).

Given that only a few metabolic reactions were sufficient to differentiate strains by clinical and environmental origin using statistical and decision tree analysis, we explored whether reaction fluxes between strain-specific GEMs could further improve strain origin differentiation. We analyzed feasible reaction flux ranges for all strain-specific GEMs by simulating each on minimal media including glucose as a carbon source and calculating growth-supporting flux ranges

using FVA. Although flux balance analysis and FVA rely on a system assumed at steady state without requiring knowledge of kinetic parameters, both are widely accepted techniques and contribute substantially to biomarker identification and mechanistic insights into disease metabolism[28,29]. The derived flux ranges were used as the input for ML-based classification (Methods). Using information from only 21 reactions, classifying environmental from clinical strains achieved an area under the curve for precision over recall of 0.92 and an accuracy of 0.79 (Fig. 3d, Supplementary Data S5). In addition to previously highlighted chorismate-associated reactions, the ML model also selected features associated with amino acid and energy pathways, especially in the mitochondrial compartment. These included, for example, homoserine succinate-lyase, ribulose-phosphate 3-epimerase, and succinate:CoA ligase, suggesting the contribution of altered amino acid and energy metabolism to differentiation of clinical and environmental *A. fumigatus* strains.

We did not observe major differences in strain origin based on the strain's accessory gene or reaction content (Fig. 2c, Supplementary Fig. S1) or complete metabolic reaction presence (Fig. 3a). In contrast, we identified a small, defined set of reactions mainly associated with amino acid, energy, and chorismate metabolic activity that were largely sufficient to differentiate clinical from environmental origin (Fig. 3b–d).

## Significant alterations in the structure of the lung microbiome upon *A. fumigatus* colonization

To investigate the applicability of the clinical-strain GEMs for predicting metabolic components supporting *A. fumigatus* growth in the human lung, we analyzed sputum samples from 40 cystic fibrosis patients from Germany (see Methods for cohort description). For all patients, we had an initial culture-negative sample and a subsequent sample that was positive for *A. fumigatus* growth. To investigate changes to the lung microbiota after *A. fumigatus* colonization, we performed shotgun metagenomic sequencing for all 80 sputum samples (*A. fumigatus* negative and positive), generating an average of 5.59 Gbp of sequencing data per sample (standard deviation 0.80 Gbp). By combining Kraken 2 and raspir[30] for taxonomic profiling, we identified a total of 67 genera and 200 species. Although spontaneous expectoration of sputum is frequently used for sample acquisition[31,32], this practice may introduce contamination from oropharyngeal flora. By applying FEAST[33], which was developed to partition microbial samples into their source components facilitating the quantification of contamination or other potential source environments, we found a dominant source of our samples to be cystic fibrosis patients' lung microbiome and a significantly higher contribution of them than the oral or clinical environment microbiome (Supplementary Fig. S2). Our study lacked true negative controls, so to minimize the chances of detecting false-positives species, we applied abundance and prevalence filters and compared detected genera and species using the alternative tools KrakenUniq and Centrifuge[34,35] to ensure robustness in lung microbe detection. 198 out of 200 species were detected by at least one of these two alternative tools (Supplementary Data S6). Despite differences among the patient cohort, starting biomaterial, and sequencing method, the taxonomic annotation of the 10 most abundant genera (Fig. 4a, Supplementary Data S6) showed striking similarities to two recent studies. In those papers, the lung microbiome of *A. fumigatus*-infected and control patients was investigated using 16S rRNA sequencing of either sputum samples or bronchoalveolar lavage[36,37].

The prevalence of the most abundant genera was consistently high (Fig. 4a, Supplementary Data S6). Notably, from the 10 most abundant genera, *Campylobacter* and *Capnocytophaga* were highly abundant in, respectively, 25 and 20 of 80 samples, (31.25% and 25%) showing an uneven distribution in the population (Supplementary Data S6). Similarly, the most prevalent species were present in most

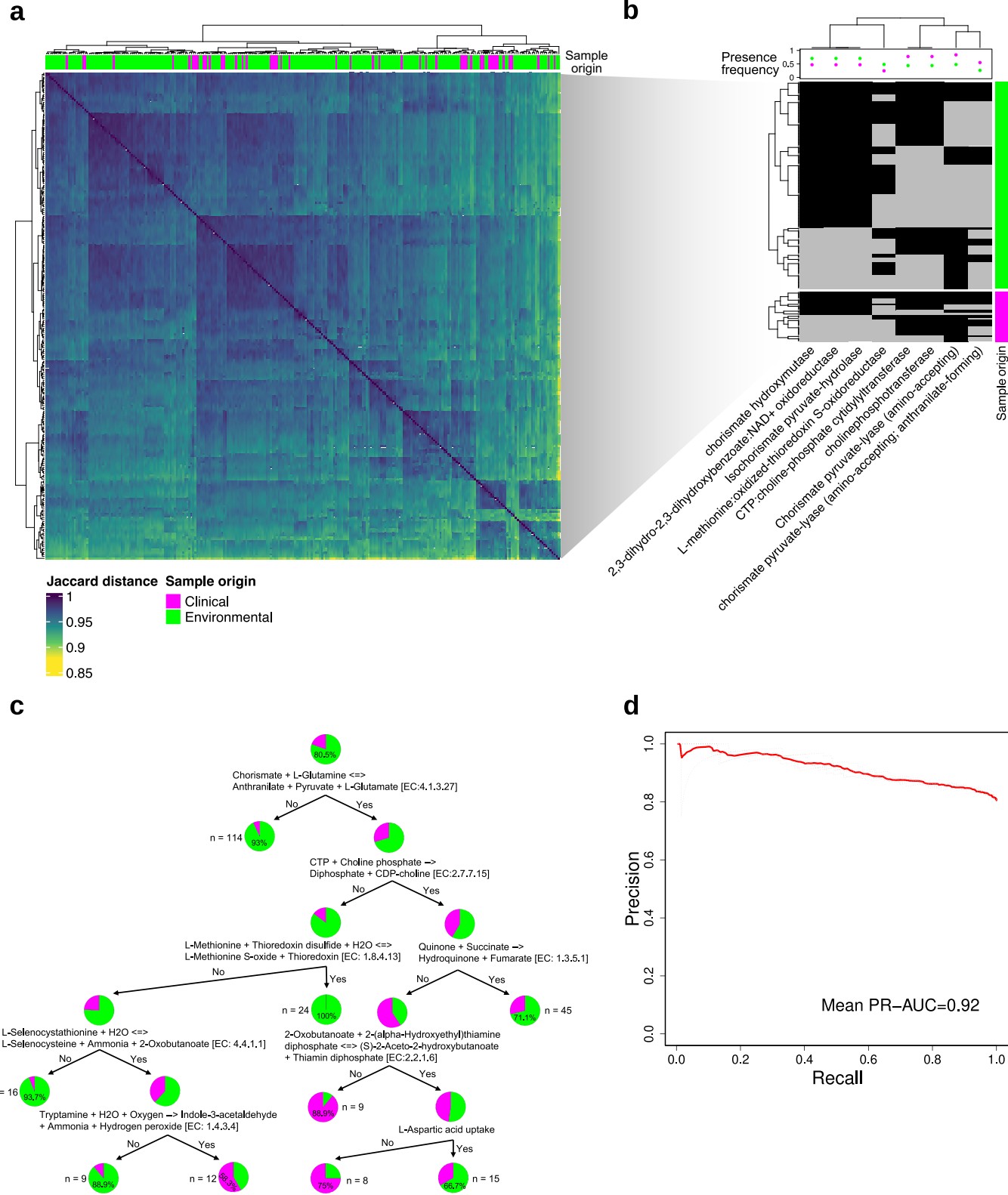

**Fig. 3 | Differentiation by origin of all strain-specific GEMs. a** Heatmap with pairwise Jaccard distance values for isolated GEM pairs based on presence or absence of metabolic reactions. **b** Selected metabolic reactions with the highest statistical significance for differences in presence/absence by sample origin (Fisher exact test, $p \leq 0.05$). Presence frequency indicates the fraction of reaction presence over all investigated strain-specific GEMs. **c** Decision tree optimized for separation of clinical and environmental strain origin. The decision tree is based on absence or presence of metabolic reactions and growth capability on different nutrients for all strain-specific GEMs. **d** Machine learning-derived mean area under the curve for precision over recall based on 21 reactions identified by the model using biomass-supporting flux ranges for all reactions determined with flux variability analysis to classify clinical vs. environmental origin. Source data for Fig. 3 are provided in the Source Data file.

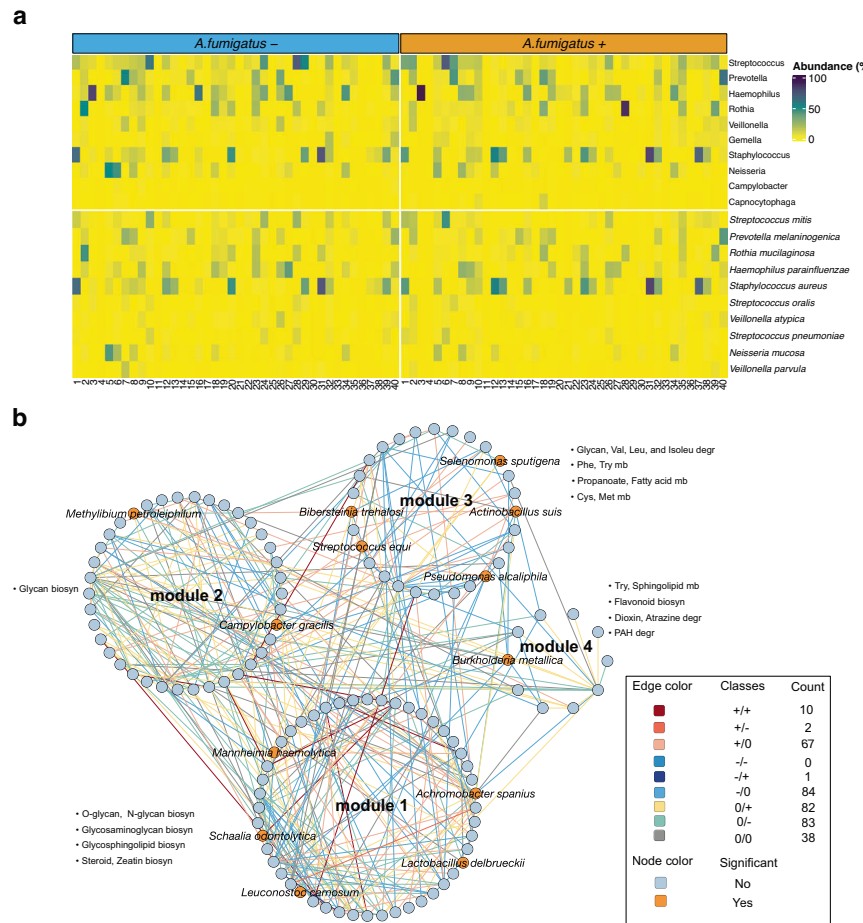

**Fig. 4 | Metagenomics sequencing of 80 total paired sputum samples taken from 40 cystic fibrosis patients before and after *A. fumigatus* colonization.**
**a** Relative abundances of the 10 most abundant genera and species from all samples. *X*-axis is ordered by patient sample number. **b** Differential correlation analysis of species in *A. fumigatus* + (after colonization) relative to *A. fumigatus* - (before colonization) showing changes in the lung microbiome interactome after *A. fumigatus* colonization. Edge colors and class information indicate the direction of correlation for *A. fumigatus* -/*A. fumigatus* + . Associated count indicates the number of species pairs in the network exhibiting this pattern of change. Only species pairs with significant differential correlations were included (permutation test, *p* ≤ 0.05). An orange node background indicates significant differential abundance for species when comparing *A. fumigatus* - vs. *A. fumigatus* + samples (metagenomeSeq, zero-inflated Gaussian mixture model, *p* ≤ 0.05, Supplementary Data S6). Abbreviations: biosyn: biosynthesis; cys: cysteine; degr: degradation; Isoleu: isoleucine; met: methionine; mb: metabolism; Leu: leucine; Phe: phenylalanine; PAH: Polycyclic Aromatic Hydrocarbon; Try: tryptophan; Val: valine. Source data for Fig. 4 are provided in the Source Data file.

samples (≥ 80%) (Supplementary Data S6). Of these species, *Streptococcus mitis* (65% occurrence in the 10 most abundant species), *Haemophilus parainfluenzae* (61.25%), *Rothia mucilaginosa* (57.5%), *Prevotella melaninogenica* (55%), and *Staphylococcus aureus* (52.5%) were most frequently found among the 10 most abundant species (Supplementary Data S6). Intriguingly, *Pseudomonas aeruginosa* was among the 10 most abundant species in 11 samples before and 17 samples after *A. fumigatus* colonization in the same patients. This species is commonly found in cystic fibrosis patients and co-occurs frequently with *A. fumigatus* colonization[38].

No statistically significant differences were seen in alpha or beta diversity (Supplementary Fig. S3a, b), but species co-abundance networks revealed notable compositional changes in the lung microbiome structure following *A. fumigatus* colonization. Using differential gene correlation analysis (DGCA), we generated networks from differentially correlated microbial pairs in paired *A. fumigatus*-negative versus *A. fumigatus*-positive samples from cystic fibrosis patients (Fig. 4b). We analyzed the resulting networks using MEGENA[39] and identified four modules in the global network that contained 13 species (orange in Fig. 4b) with differential abundance (metagenomeSeq, zero-inflated gaussian mixture model, *p* ≤ 0.05) between *A. fumigatus*-negative and subsequently positive patient samples. For the top 5

differentially abundant species (by p-value), existing edges (class +/0 in Fig. 4b, Supplementary Data S6) of *Bibersteinia trehalosi*, *Neisseria elongata* and *Methylibium petroleiphilum* with *Actinobacillus suis*, *Bibersteinia trehalosi* and *Paraburkholderia hospita* in *A. fumigatus*-negative samples were lost upon colonization with *A. fumigatus*. Similar patterns were observed with loss of negative associations between *Bibersteinia trehalosi* with *Xanthomonas vesicatoria*, *Xanthomonas arboricola* and *Stenotrophomonas rhizophila*, *Campylobacter gracilis* with *Pseudomonas resinovorans* (−/0, cyan edges, Fig. 4b, Supplementary Data S6). Additionally, 6 newly formed negative associations (0/−) and 7 newly formed positive associations (0/+) were observed for these species (Fig. 4b, Supplementary Data S6).

To evaluate the functional implications of microbiome restructuring following *A. fumigatus* colonization, we performed KEGG orthology enrichment analysis on the four identified modules of our co-abundance networks (Methods). Interestingly, modules 3 and 4 were enriched in amino acid metabolism, for example, phenylalanine and tryptophan, but also valine and (iso-)leucine. Further enrichments included glycosaminoglycan and steroid biosynthesis (module 1), glycan biosynthesis (module 2), propanoate and fatty acid metabolism (module 3), and flavonoid metabolism (module 4, Supplementary Data S6). Additionally, we performed correlation analysis with clinical

metadata on lung functional capacity as quantified by the forced expiratory volume (FEV, Supplementary Data S7), Among other associations, we found a significant association of FEV and species of module 3 (permutation test, $p \leq 0.05$) and positive correlations between FEV and a number of amino acid-involving pathways including biosynthesis of phenylalanine, tyrosine and tryptophan, and also with glycolysis (Spearman's correlation, $p \leq 0.05$, Supplementary Data S7). These data suggested a relationship between clinical manifestation during *A. fumigatus* colonization and the lung microbial profile, possibly involving metabolic exchanges between the pathogen and lung bacteria.

In summary, we identified a significant change in the structure of the lung microbiome reflected in a distinct set of 319 species co-abundance differences upon *A. fumigatus* colonization. The associated metabolic functions enriched in differential correlation microbial modules pointed again to amino acid pathways, particularly for aromatic amino acids, in addition to fatty acid, nitrogen, and sulfur metabolic pathways, suggesting that lung microbiome metabolic activity is reshaped in the presence of *A. fumigatus*.

## Changes in the lung microbiome metabolic output induced by *A. fumigatus* colonization support pathogen growth

We subsequently investigated if changes in the lung microbial community triggered by the presence of *A. fumigatus* were accompanied by changes in the metabolic output of the microbiome. Since an experimental metabolomics analysis of the sputum samples would reflect dietary molecules and the metabolic output of the host and pathogen as well as the microbiome, we opted for an in silico prediction. For this aim, we used the MAMBO algorithm[40] to predict the most probable lung microbiome metabolic profile that supported the relative abundances of our identified metagenomics species. We found near significant differences in the overall derived metabolite profiles when comparing patient samples before and after *A. fumigatus* colonization (Euclidean distance; PERMANOVA, $p = 0.074$, Fig. 5a, Supplementary Data S8) indicating that the changes in the lung microbiome structure (Fig. 4b) likely had significant functional implications. Of note, more than 86.8% of the variance could be explained by the first 50 dimensions of the principal component analysis. The first two components explained fewer, albeit significant, changes in metabolites between patient samples before and after *A. fumigatus* colonization despite the high complexity of these data.

We next quantified how the changes in the metabolic output of the lung microbiome following *A. fumigatus* colonization might alter the predicted growth of the clinical strain-specific GEMs. Using the MAMBO-derived metabolite profiles present after *A. fumigatus* colonization, we observed that the GEMs of the 49 clinical strains showed a significant increase in the predicted growth rate compared to GEMs simulated on the metabolic outputs from before *A. fumigatus* colonization (3.4% increase, Wilcoxon signed-rank test, $p = 3.55e{-}15$, Fig. 5b). These results suggested that the changes induced by *A. fumigatus* in the lung microbiome led to a nutritional profile that supported its own growth. Notably, the achieved absolute growth rates were higher than previously reported growth rates on, for example, minimal media[21], but nevertheless indicated beneficial growth after functional output from the microbial community changed upon *A. fumigatus* colonization.

To explore if we could identify a connection between the altered lung microbiome and the *A. fumigatus* metabolic capacity, we analyzed feasible flux ranges of reactions that were associated with enriched metabolic subsystems that we identified in the *A. fumigatus* affected lung microbiome (Fig. 4b, Supplementary Data S5). Using the *A. fumigatus* clinical-GEMs simulated with FVA and MAMBO-derived media from before, compared to after, *A. fumigatus* confirmed colonization, we identified 77 metabolic reactions that showed significantly lower or upper flux ranges to support fungal growth (false discovery

rate [FDR] corrected paired Wilcoxon test, $p \leq 0.05$, Fig. 5c, Supplementary Data S8). Most filtered reactions showed significant differences in the upper range, suggesting increased metabolic activity of *A. fumigatus* (Fig. 5c). Affected pathways mainly included amino acid metabolism, primarily for aromatic amino acids, but also nitrogen, sulfur, butanoate or steroid metabolic pathways (Supplementary Data S8), the latter possibly due to antifungal treatments targeting ergosterol biosynthesis[41]. Although *A. fumigatus* negative and positive samples had overlapping predicted flux ranges, the change in direction was mostly consistent on a per-strain GEM level (Fig. 5d). Interestingly, we already identified the reactions of chorismate pyruvate-lyase (EC4.1.3.27) and tryptamine:oxygen oxidoreductase (EC1.4.3.4) as major discriminators between environmental and clinical strains simulated on minimal media before (Fig. 3c, Supplementary Data S8). Only 29 of 77 reactions showed significant differences in both lower and upper flux bounds (Supplementary Data S8). These reactions included those of L-arogenate hydro-lyase (EC4.2.1.51) and L-Phenylalanine:2-oxoglutarate aminotransferase (EC2.6.1.1), which showed notably constrained flux-bound variability across most simulated strain GEMs (Fig. 5d, Supplementary Data S8) suggesting a role for shikimate pathway-associated metabolites and thioredoxin in differentiating *A. fumigatus* origin (Fig. 3c) and in determining colonization status in cystic fibrosis.

To further explore the relevance of amino acid supplementation as either carbon or nitrogen sources based on the predicted pathways, we cultivated four clinical strains and used relative metabolic activity and radial growth assays to test the metabolic benefit of supplementing with glutamine, glycine, phenylalanine, or tryptophan (Methods). Relative metabolic activity and radial growth increases based on minimal media were highest for glutamine (two-tailed $t$ test, adjusted $p$-value $\leq 0.05$, Fig. 5e, Supplementary Data S4). Glycine also led to a significant increase in relative activity but no growth improvement in radial growth assays on minimal media. In contrast, the addition of tryptophan or phenylalanine to *Aspergillus* minimal media (AMM) decreased metabolic activity significantly (34% for tryptophan and 84% for phenylalanine) and growth (0.91% and 0.93%, respectively) on minimal media (Fig. 5e, Supplementary Data S4). On cystic fibrosis-resembling media (SCFM2, Methods), supplementation with the aromatic amino acids phenylalanine or tryptophan resulted in a significant growth increase (4% for phenylalanine, 14% for tryptophan) across all four clinical strains, while adding glutamine or glycine resulted in no significant or steady increase (Fig. 5e, Supplementary Data S4). Our phenotypic microarray data also showed a positive growth effect when most amino acids were tested as a carbon or nitrogen source for the Af293 wild type strain (Supplementary Data S2). Interestingly, phenylalanine provided a growth benefit if provided as the carbon source, but not the nitrogen source in contrast to tryptophan, which provided a growth benefit as the nitrogen source. With the MAMBO predicted cystic fibrosis media composition, we observed a slight, statistically insignificant increase for all four amino acids in the *A. fumigatus*-positive samples (Supplementary Data S8).

Taken together, our data and simulations suggest that aromatic amino acids, in particular, provided by lung bacteria might have beneficial growth implications for *A. fumigatus* in the context of cystic fibrosis. Other amino acids such as glutamine serve as potential nitrogen sources if the media are limited, but appeared not to impact fungal growth in the context of cystic fibrosis. In addition to these metabolites, intermediates such as chorismate, anthralinate, and cholines appeared in multiple ML classification models to differentiate clinical from environmental strains in our study. Given that several of our 53 reactions with predicted significant flux changes in cystic fibrosis were related to the underlying shikimate pathway, our data suggest that aromatic amino acids are of elevated importance for fungal colonization and potentially the severity of this lung disease.

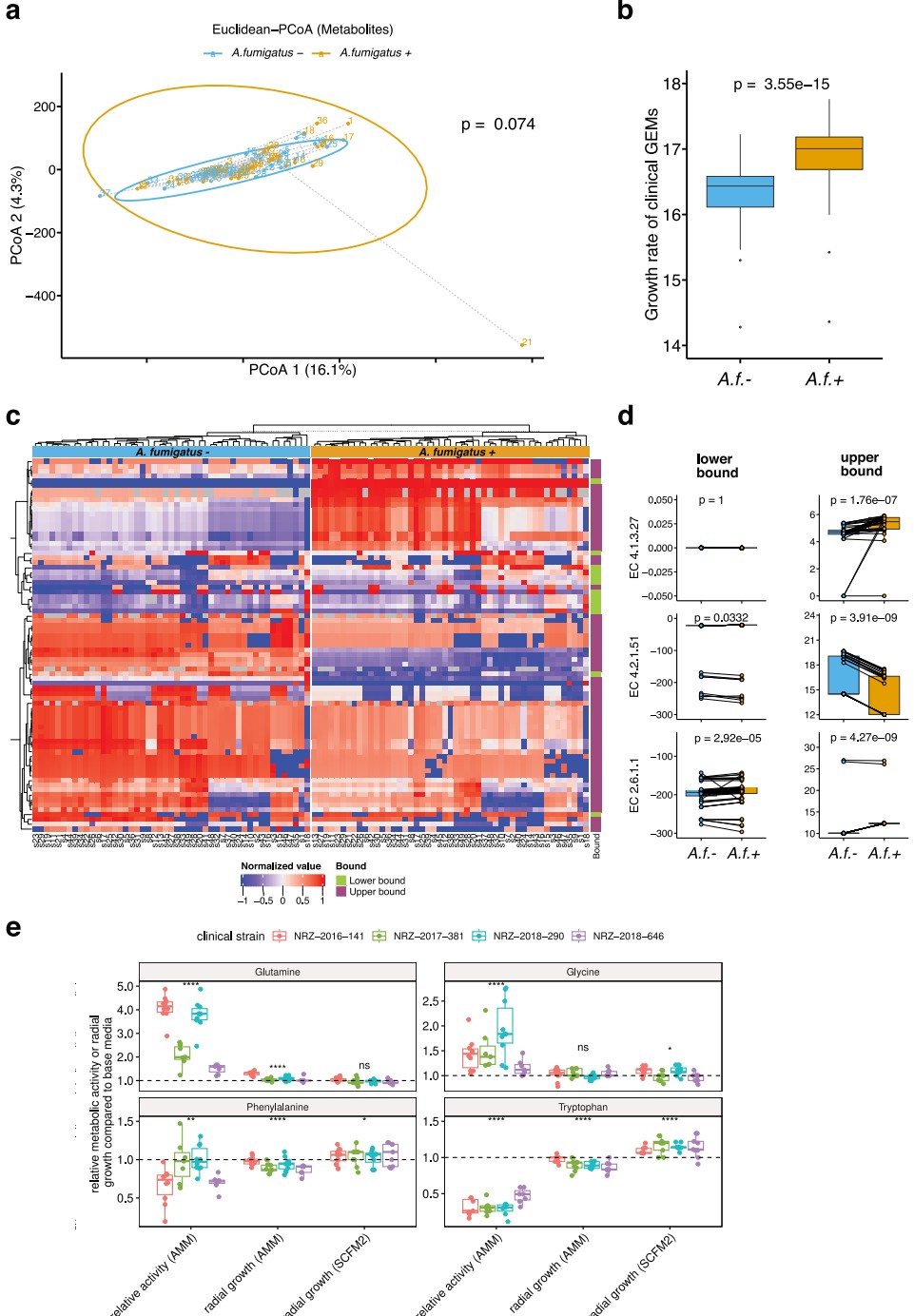

**Fig. 5 | MAMBO-derived metabolite profiles for samples from cystic fibrosis patients. a** Beta diversity (Euclidean distance) for MAMBO-derived media. PER-MANOVA was used to assess the statistical significance of beta diversity comparisons. **b** Growth rate differences from genome-scale metabolic models corresponding to clinical *A. fumigatus* strains based on MAMBO-derived media compositions associated with samples from cystic fibrosis patients before and after *A. fumigatus* colonization ($n = 49$). *P*-value following two-tailed Wilcoxon rank sum test. **c** Significantly different flux ranges (in lower or upper bound) of clinical strain GEMs simulated with flux variability analysis on MAMBO-derived media before and after *A. fumigatus* colonization. Significance was determined by FDR-adjusted paired Wilcoxon signed-rank test ($p \leq 0.05$). **d** Three selected enzymatic reactions with significant flux bound differences for lower or upper bound as in (**c**). Both bounds are indicated. *P*-value following two-tailed Wilcoxon rank sum test.

**e** Relative changes in metabolic activity or radial growth for four clinical strains (Supplementary Data S3) on *Aspergillus* minimal (AMM) or cystic fibrosis media (SCFM2) supplemented with four amino acids as indicated. The dashed line indicates a neutral fold-change of 1. A higher value indicates positive metabolic activity or growth effect when the indicated amino acid was added. Statistical significance was tested with a two-tailed *t* test (****$p \leq 0.0001$, **$p \leq 0.01$, *$p \leq 0.05$, ns: not significant). EC4.1.3.27: chorismate pyruvate-lyase; EC4.2.1.51: L-arogenate hydro-lyase; EC2.6.1.1: L-Phenylalanine:2-oxoglutarate aminotransferase. *A.f.*+, *A. fumigatus* positive, or *A.f.*-, *A. fumigatus* negative. Box-plot elements: center line: median, lower/upper bound: 25th/75th percentile, whiskers: minimum and maximum values within 1.5 × interquartile range (IQR), outliers: points outside ±1.5 × IQR. Source data for Fig. 5 are provided in the Source Data file.

## Discussion

In this study, we built a suite of *A. fumigatus* genome-scale strain-specific metabolic reconstructions originating from 252 environmental and clinical isolates from Germany[7]. We (i) reconstructed a comprehensive pan-GEM of *A. fumigatus* metabolism in a data-driven manner, which we validated against phenotypic microarray and gene essentiality data; (ii) derived 252 strain-specific GEMs from individual genome assemblies and manually curated them towards growth feasibility and minimal fractioned network topologies; and (iii) determined metabolic differences differentiating clinical from environmental strains, such as metabolic reactions involving several amino acids, particularly aromatic amino acids as well as chorismate or thioredoxin. Chorismate is an important precursor for aromatic amino acids and is formed in the shikimate pathway. This seven-step pathway is not present in animals and enables the synthesis of the aromatic amino acids tyrosine, phenylalanine, and tryptophan. Thioredoxin is an important factor for DNA synthesis and is associated with *A. fumigatus* virulence[27,42].

Multistrain-GEMs have been used to elucidate the metabolic diversity of human-pathogenic bacteria. For example, they defined the pan-metabolic capabilities of *Pseudomonas putida*[43], loss of fitness-relevant pathways for survival in the gastrointestinal environment in extraintestinal *Salmonella* spp.[44], and strain-specific metabolic capabilities of *Staphylococcus aureus* linked to pathogenic traits and virulence acquisitions[45]. Here, we applied this strategy to explore metabolic diversity in a eukaryotic fungal pathogen. This strain-specific *A. fumigatus* GEMs platform is publicly available (BioModels ID MODEL2211100001) for investigating the metabolically relevant *A. fumigatus* gene set and its impact on the metabolic diversity influencing growth rate capabilities, metabolic adaptation, and pathogenicity in this important human fungal pathogen.

As a proof-of-concept of the applicability of our fungal GEM collection, we performed metagenomics sequencing analysis of sputum samples from a cohort of 40 cystic fibrosis patients, with samples before and after a confirmed *A. fumigatus* colonization. Clinical isolate-specific simulations and analysis showed significantly increased growth rates in the predicted lung microenvironment of cystic fibrosis patients after a confirmed *A. fumigatus* colonization, suggesting that the lung microbiome is remodeled to a state more favorable to fungal growth. Our analysis predicted 77 metabolic reactions associated with aromatic amino acid metabolism and the shikimate pathway would have significantly different flux ranges after *A. fumigatus* colonization of the lung. These reactions appear not only in aromatic amino acid metabolism but also in sulfur, nitrogen, and lipid metabolic pathways, highlighting the advantage of including topological pathway information when analyzing metabolic activity. Several of these reactions are shown to be essential for the growth of the fungus in knockout experiments, including knockouts for *aroC*, *TrpC*, and *MET16*, making these genes potential targets against the fungus[19,25]. However, we identified a challenge in drug design using potentially targetable biomarkers originating from lethality data of in vitro gene deletions in *A. fumigatus*. Our analyses using metabolic modeling coupled with growth experiments suggest that *A. fumigatus* might be able to grow without the function of targeted genes by obtaining the necessary metabolites externally. This might occur in patients, for example, with *A. fumigatus* acquiring metabolites generated by lung bacteria (Fig. 6), as has been shown for tryptophan and *A. fumigatus* with *TrpC* and *TrpE* deletions[25,46,47]. Therefore, more sophisticated therapeutic approaches targeting both these essential metabolic genes and specific transporters may be required to restrict the growth of this pathogen.

Limitations of our study include genomic and subsequently metabolic differences between the clinical strain collection (isolated mainly from patients with invasive aspergillosis) used to build the strain-specific GEMs and the clinical strains in the cystic fibrosis patients in our study. In our previous study, however, we showed that genomic similarities of clinical strains are relatively high even when the strains originate from different countries[7]. We used the in silico method MAMBO to disentangle the contribution of the microbiome from the host and diet to generate the nutritional profile available for *A. fumigatus* in the human lung, because of a lack of an experimental method. MAMBO relies on the quality of the taxonomic species annotation to infer the metabolic output of the bacterial community. Since the human lung microbiome is much less studied than the human gut[48], the inclusion of false-positive species cannot be excluded despite measures we implemented such as prevalence and abundance filters, and multiple annotation tools.

Altogether, the presented analyses demonstrate that fungal, genome-scale metabolic modeling is feasible at the strain level. The results contribute towards a mechanistic understanding of the impact of genome diversity on the metabolic phenotype of *A. fumigatus* and its metabolic interdependencies with bacterial communities in the human lung.

## Methods

### Biomass formulation

We generated a specific *A. fumigatus* biomass composition based on several sources in the literature. First, we assigned proportions of main biomass components, as described,[49] as 38.8% carbohydrates, 9.9% lipids, 30% proteins, 0.6% DNA, and 3.7% RNA. Since this resource neglected polyols, we added 4% polyols as reported for *Aspergillus oryzae*[50] for a total of 42.8% carbohydrates. After adding 6.6% cofactors, these main components made up the total biomass composition together with the reported 6.4% ash fraction[49]. Next, we screened the literature to specify fractions of carbohydrate subcategories, e.g., glucans or trehalose[51–53]; lipids including sterols; phospholipids; neutral lipid and free fatty acid compositions[54,55]; amino acid composition of the protein content[56,57]; and co-factor content including energy carriers such as NADH and vitamins such as riboflavin[58,59]. After calculating the mmol/g content of each fraction, we added ATP demand according to models developed for *Saccharomyces cerevisae* and *Aspergillus niger*[60,61]. Furthermore, we added a nongrowth-associated ATP maintenance reaction and calibrated flux through it for reported growth rates of the fungus in batch-fermentation process in hypoxic and normoxic glucose-limited conditions (further description below, Supplementary Data S1)[21]. Finally, we modified the proportion of all components to resemble 1 g dry weight (Supplementary Data S1).

### Pan-GEM reconstruction

All reconstruction and analysis efforts used COBRApy (v0.17.1)[62] in Python 3.6.8 and the academic version of the IBM CPLEX solver (v12.8.0.0).

We gathered and combined information from automatically generated draft reconstructions based on the CoRoCo pipeline[13]. We downloaded the *Aspergillus* CoReCo model for *A. fumigatus* (Biomodels ID MODEL1604280029) and further *Aspergillus* models from the CoReCo repository including *A. oryzae* (Biomodels ID MODEL1604280012), *Aspergillus nidulans* (Biomodels ID MODEL1604280008), *A. niger* (Biomodels ID MODEL1604280021), *Aspergillus clavatus* (Biomodels ID MODEL1604280016), *Aspergillus terreus* (Biomodels ID MODEL1604280019), and *Aspergillus gossypii* (Biomodels ID MODEL1604280044) from the BioModels repository (https://www.ebi.ac.uk/biomodels/). In addition, we incorporated metabolite and reaction information from a recently published *A. fumigatus* central metabolism model[14]. This yielded a base model of 7,606 reactions (of which 3,233 were exchange reactions) and 3,578 metabolites. Of note, the high number of initial exchange reactions originated from models created with the CoReCo pipeline, which includes an exchange reaction for each defined metabolite by default. All subsequent curation efforts aimed to keep only reactions for which annotation information was available or that were

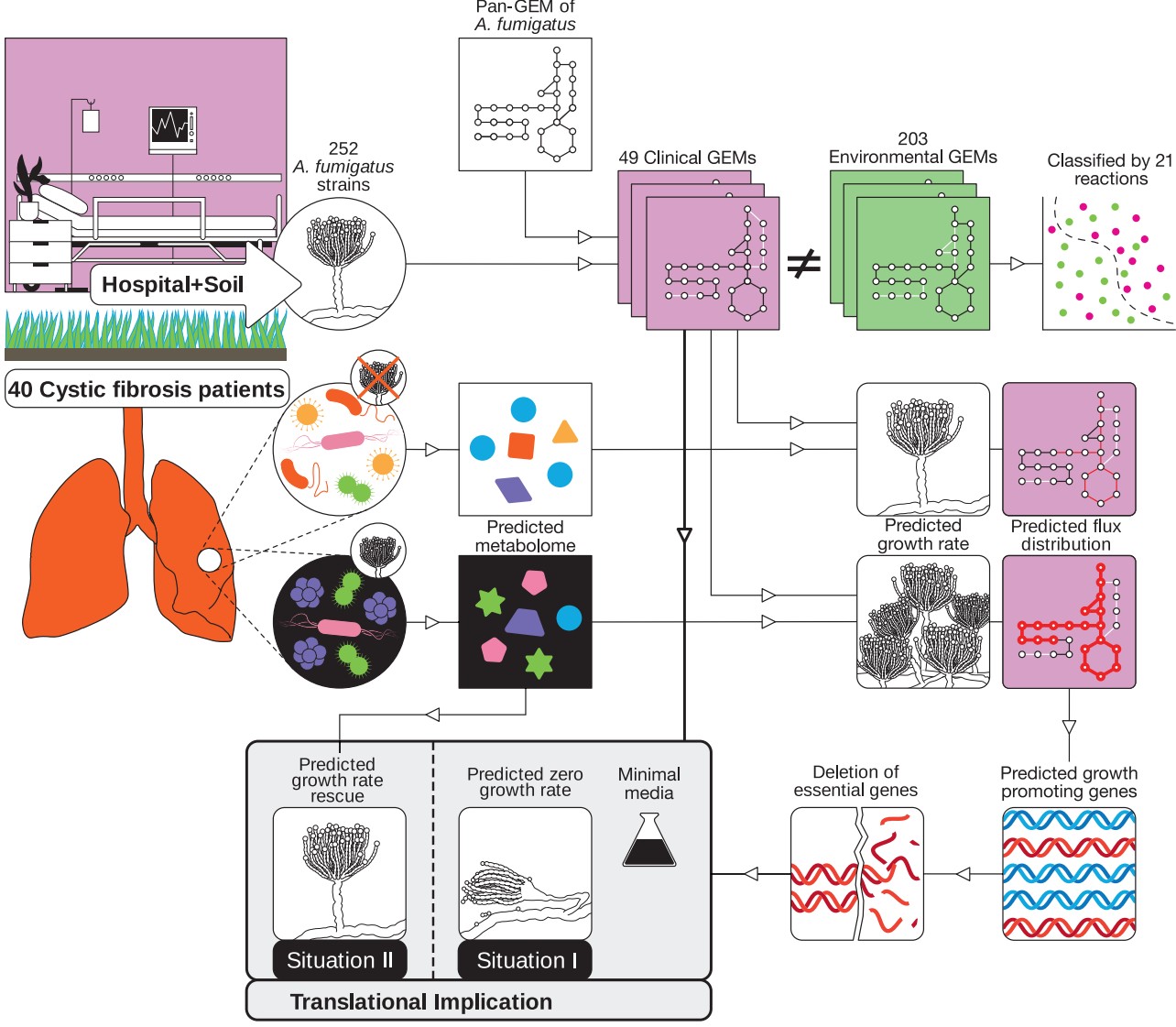

**Fig. 6 | Workflow for strain-specific GEM generation, data acquisition to translational implications.** Beginning with isolating 252 *A. fumigatus* strains from hospitals and soil, we generated a comprehensive pan-GEM for deriving 252 strain-specific GEMs. By including lung microbiome analysis and simulating fungal-lung microbiome interactions, we predicted and experimentally tested nutritional environments for supporting lung bacteria that impact *A. fumigatus* growth. Both lung bacterial communities and metabolites have the potential to affect *A. fumigatus* growth and thus the effectiveness of drugs to treat infections by this opportunistic fungal pathogen.

necessary for model feasibility. Filtering duplicate reactions and metabolites reduced the model by 73 reactions and 210 metabolites. Filtering duplicate reactions was necessary because our initial merge of metabolic information included reactions and converted metabolites from different sources. Only when reactions or metabolites had different naming conventions we reduced to a single naming scheme. When different isozymes encoded the same metabolic reaction, we included all alternative genes as OR relationships (instead of AND) in the genes-to-protein rule of the affected metabolic reaction and removed the duplicate reaction. The biomass formation was modified based on the literature on *A. fumigatus* metabolism and enriched with information from closely related species if we did not find *A. fumigatus*-specific information (*Biomass formulation* section).

Next, we screened the KEGG (https://www.kegg.jp/) and MetaCyc (https://metacyc.org/) databases for gene annotation for *A. fumigatus* metabolism and added 1444 genes to the model. When available, we adopted AND and OR relationships for genes for metabolic reactions and crosschecked with genes encoding reactions in the yeast consensus model[61]. During this step, 2370 reactions could not be mapped to any annotated gene and were removed from the template model but kept for subsequent gap-filling procedures. Further curation efforts were run in parallel, since modifications influenced other curation procedures. Curation included compartmentalization, resolving erroneous energy generating cycles (EGCs)[17] and gene essentiality information[19], and adaptation to phenotypic growth assays (Methods section *Biolog phenotypic microarray*, Supplementary Data S2). To add compartment information for all reactions, we applied subcellular localization prediction using WoLF PSORT[16]. A reaction was allocated to a particular compartment if more than 50% of the associated genes were predicted to be located in that compartment with more than 50% probability. Reactions, including exchange reactions, were associated with nine compartments: cytoplasm, mitochondrion, nucleus, peroxisome, endoplasmic reticulum, lipid particles, vacuole, golgi, and extracellular space. When the prediction was ambiguous or precluded a viable model as measured by biomass production based on defined minimal media, we used concurrent alternative compartment localizations, either as predicted by WoLF PSORT or from the curated

*S. cerevisae* GEM[61]. Compartment-connecting transport reactions were from the yeast consensus model[61]. A minimal set of additional necessary transport reactions were added using gap-filling functionality from COBRApy to allow biomass precursor production based on minimal media with glucose.

In parallel, we resolved EGCs again[17] and adapted our GEM model using publicly available gene essentiality data[19]. EGCs are metabolic reactions running in a potentially nontrivial circle without a net flux except for generating energy carriers. ATP, CTP, GTP, UTP, ITP, NADH, NADPH, FADH2, FMNH2, acetyl-CoA, L-glutamate, ubiquinol-8, demethylmenaquinol-8, and menaquinol-8 were found in at least one EGC (Supplementary Data S9). The directions of 44 reactions were refined using reaction directionalities from the BiGG[63] and BRENDA[64] databases and Gibbs free energy of the reactions from the MetaCyc database[65]. Incompatible gene-essentiality information was resolved by either correcting to a feasible thermodynamically reaction direction or removing erroneous reactions without gene annotation.

Next, we ran phenotypic microarrays with the *A. fumigatus* reference strain Af293 and four mutant strains (experimental details below). We considered 0.001 mmol/grDW/h as an absolute minimum for growth rate to account for reported low-oxygen growth conditions, e.g., in glucose-limited media[21]. We identified essential carbon, nitrogen, sulfur, and phosphorus components (Supplementary Data S2). This step included plausible correction of reaction directions for thermodynamically feasibility and removal of reactions without gene annotation. When our data showed growth in a certain condition that our pan-GEM could not predict, we investigated the direction of related reactions, potential gaps in the metabolic network, and connectivity of the involved compartments. When our data showed no growth under a given condition but our GEM predicted a nonzero growth rate, we investigated if the metabolic reactions in our initial, draft pan-GEM were not associated with the genome and needed to be removed. We used similar curation efforts to resolve incompatibilities in the gene-essentiality data. When incompatibilities could not be resolved in this way, we screened our catalog of initially removed reactions without gene annotation and used gap-filling procedures from the COBRApy gap-fill functionality.

Next, we calibrated our pan-GEM to reflect experimental growth and byproduct secretion rates in glucose-limited environments in hypoxic and normoxic conditions. We retrieved published glucose, acetate, ethanol, lactate, and cell dry weight concentration values from Barker et al.[21] and converted these values to plausible units (e.g., mmol/grDW/h for metabolites and 1/h for growth) for subsequent GEM analysis (Fig. 1). We constrained our pan-GEM for the experimental glucose consumption rates (i.e., fixed glucose uptake of 0.250 mmol/grDW/h for normoxic and 0.206 mmol/grDW/h for hypoxic conditions) and calibrated flux through the nongrowth-associated maintenance reaction with different $O_2$ uptake rates resembling hypoxic to normoxic conditions. The data fitted best when setting the flux through the nongrowth-associated maintenance reaction of the pan-GEM to 0.1 mmol/grDW/h (Supplementary Data S9 and S10).

Next, we analyzed re-introduced non-genome annotated reactions during curation to identify reactions that re-introduced metabolic redundancy after all curation steps were done. We identified 609 putative transport reactions between compartments and 79 further reactions occurring in only one compartment, which we analyzed further. 210 were blocked based on analyzing relaxed influx through all defined exchange reactions. The remainder of 478 reactions were analyzed for essentiality, that is, identifying reactions that require carrying flux to support a non-zero biomass value given relaxed metabolite influx. Towards this goal, we conducted a minimal cut set analysis[22] assuming hypoxic or normoxic growth conditions and otherwise again unconstrained exchange reaction fluxes. 314 reactions were individually essential, or essential as a set of reactions identified by MCS analysis. For 164 reactions the essentiality status remained

unknown with relaxed exchange flux bounds and biomass optimization. In addition, we investigated the 79 non-transport reactions for identity in metabolite conversion except co-factors. We identified only two pairs of reactions that differed in the use of co-factors NAD/NADH or NADP/NADPH, but had different directionality due to curation of energy-generating cycles. These were kept accordingly and we removed the 210 blocked reactions from our generic draft pan-GEM. We finally analyzed our pan-GEM for redundant reactions using CHESHIRE[66]. We used the tool to identify 200 reactions based on similarity score and identified only one further internal transport redundancy for CoA, comprising two reactions with different directionality and resolved this by allowing bidirectional transport. No further redundant reaction was identified by CHESHIRE, which concluded our refinement efforts for our pan-GEM.

Last, we checked the compatibility of our model with metabolic modeling standards by running MEMOTE tests[67]. The overall MEMOTE score we achieved was 73%, which is in the range reported for the most-curated yeast GEM (https://sysbiochalmers.github.io/yeast-GEM/release_report.html). Simulation scripts for all analyses are at Github: https://github.com/mohammadmirhakkak/A_fumigatus_GEM/, which also holds the MEMOTE report (Github folder: https://mohammadmirhakkak.github.io/A_fumigatus_GEM/memote_report.html). Simulation-specific constraints are in Supplementary Data S10.

## Strain-specific GEM reconstruction and curation

Recently, the pan-genome of *A. fumigatus* was derived for 300 environmental and clinical strains with a global distribution[7]. By mapping the genomes for 252 strains to the Af293 *A. fumigatus* reference genome annotation, we identified metabolically relevant genes by requiring at least 95% sequence identity under the rationale that high sequence identity preserves metabolic function. Small deviations from the chosen sequence identity threshold did not change the results. The similarity analysis was done by BLAST analysis of protein sequences using diamond (v0.9.24.125)[68]. The presence or absence of metabolic reactions was deduced for each strain using the associated gene-protein-reaction rules from the pan-GEM and the relevant identified genes. The metabolic core comprised reactions and genes that occurred in all strains with other reactions and genes defining, respectively, the accessory reactome and genome. To ensure that all strain-specific GEMs showed nonzero growth capability on minimal media with glucose as the carbon source, we identified and resolved minimal sets of essential reactions required to operate when adapted to the minimal cut set concept[22]. Finally, we guaranteed a consistent network property by identifying and discarding blocked reactions for each isolate using FASTCC[23]. All models were deposited in BioModels[24] and assigned the identifier MODEL2211100001.

## Phenotypic microarrays

The fungal strains Af293 (wild type reference), CEA17 *pyrG*, *ΔlysF*, *Δmet2*, and *ΔniaD* were grown at 25 °C for 7 days before experimental assays on malt agar supplemented with 5 mM uracil. Uracil supplementation was required for growth of the CEA17 *pyrG* strain (a uridine auxotroph) and was added to all media to ensure comparable growth. Mature conidia were harvested by rubbing plates with sterile distilled water and filtering the resulting solution through a 30-μm cell strainer to remove mycelial fragments. Spore purity was assessed and confirmed by microscopy. Spore solutions were adjusted to a transmittance of 75%. Phenotypic microarrays were performed using Biolog Phenotypic Microarray plates PM1, PM2, PM3, and PM4 (Biolog Inc., Hayward, CA, USA) prepared following the manufacturer's protocol for filamentous fungi, including resuspending conidia in filamentous fungi (FF) media and the addition of 0.16 ml of Biolog Redox Dye D to the master mix of each plate to quantify fungal metabolic activity. By using PM1-PM4 we could investigate 379 different growth conditions rather than only 95 as available with Biolog's FF plate (Fungi

Identification Test Panel). The plates were incubated at 37 °C for 3 days and metabolic activity was measured colorimetrically using an Omni-Log microplate reader with readings taken every 15 min. The incubation temperature was changed from the initial 25 °C to 37 °C for the phenotypic microarray experiments to mimic the change in temperature that occurs in the transition from the environment to the human host as used before[69–71]. Experiments were performed as biological duplicates or triplicates (Supplementary Data S2). Phenotypic microarray results were analyzed in R, and statistical comparison used Dunnett-type comparison of growth signals of negative controls against all other wells in a plate. All wells with signals greater than the negative control and $p$-value ≤ 0.05 were considered growth cases.

### Cystic fibrosis sample acquisition

This study was approved by the Ethics Committee of the University of Heidelberg and written informed consent was obtained from all patients or their parents or legal guardians (S-370/2011). Patients were treated according to the standard of care[72]. The diagnosis of cystic fibrosis was verified by established diagnostic criteria[73,74]. Spontaneously expectorated sputum was collected during visits to the Cystic Fibrosis Center at the University Hospital Heidelberg and frozen in liquid nitrogen on the day of visit. Sputum samples were collected from 40 cystic fibrosis patients before and after they had positive *A. fumigatus* colonization. Samples were frozen within 24 h after reception at the microbiology department. The cohort for 80 total samples was 15 females and 25 males aged 23.6 ± 4.96 years (mean ± standard deviation) before *A. fumigatus* colonization. Samples were taken during visits from patients without exacerbation or intravenous antibiotic treatment in the previous 3 months. Supplementary Data S7 has clinical data on cystic fibrosis patients including age, body mass index, sex, forced expiratory volume-one second in pulmonary function testing (PFTFEV1) and percent predicted PFTFEV1 (PFTFEV1pred). Sputum samples were prepared for microscopy using lactophenol anilin blue solution to detect fungi. In parallel, samples were plated on: Columbia agar (with 5% sheep blood) (BD Diagnostic, Heidelberg, Germany), chocolate agar (BioMérieux, Nürtingen, Germany), McConkey agar (BioMérieux, Nürtingen, Germany), *Burkholderia cepacia* special agar (7 days, 36 °C) (BD Diagnostic, Heidelberg, Germany), and Sabouraud agar (7 days, 36 °C) (BD Diagnostic, Heidelberg, Germany). Two other media were used for anaerobic isolation (36 °C): Schaedler agar (Bio-Mérieux, Nürtingen, Germany) and kanamycin-vancomycin agar (BD Diagnostic, Heidelberg, Germany). Colonization was defined as positive culture from a specimen from the sputum samples on at least one of the agar plates.

### DNA extraction and fragmentation

DNA extraction procedures followed the QIAamp DNA Mini and Blood Mini Handbook. The QIAampl DNA minikit was used for DNA extraction. Sputum samples were placed in a 1.5 ml microcentrifuge tube and 180 μl Buffer ATL was added. Proteinase K (20 μl) was added, and samples were mixed by vortexing and incubated at 56 °C. After centrifuging, 4 μl RNase A (100 mg/ml) was added, followed by mixing by pulse-vortexing for 15 s, and incubating 2 min at room temperature (15–25 °C). After centrifuging to remove drops from inside the lid, 200 μl Buffer ATL was added. Samples were mixed by pulse-vortexing for 15 s and incubated at 70 °C for 10 min. After briefly centrifuging, 200 μl Buffer AL was added to samples, followed by mixing by pulse-vortexing for 15 s, and incubating at 70 °C for 10 min. After brief centrifugation, 200 μl ethanol (96–100%) was added and samples were mixed by pulse-vortexing for 15 s. After brief centrifugation, QIAamp Mini spin columns and 500 μl Buffer AW1 were added before centrifuging at 6000*g* (8000 rpm) for 1 min. To QIAamp Mini spin columns 500 μl Buffer AW2 was added before centrifuging at full speed (20,000*g*; 14,000 rpm) for 3 min. To the QIAamp Minispin columns, 200 μl Buffer AE or distilled water was added before incubating at

room temperature for 1 min and centrifuging at 6000*g* (8000 rpm) for 1 min. Genomic DNA was randomly fragmented by sonication.

### Library preparation and DNA sequencing

DNA fragments were end-polished, A-tailed, and ligated with full-length adapters for Illumina sequencing, followed by PCR amplification with P5 and indexed P7 oligos. PCR products for final construction of libraries were purified with the AMPure XP system (Beckman, Krefeld, Germany). Libraries were checked for size distribution with an Agilent 2100 Bioanalyzer (Agilent Technologies, CA, USA), and quantified by real-time PCR to ensure an amount of at least 3 nM). Qualified libraries were inserted into Illumina sequencers (MiSeq system). For quality checking, 1 ethidium bromide negative control was added for every 11 samples and treated with the same handling procedure as the experimental samples.

### Metagenomics and MAMBO analysis

Trimmomatic was used to clip adapter and low-quality bases (v0.36, ILLUMINACLIP:TruSeq3-PE-2.fa:2:30:10:1:TRUE, LEADING:3, TRAILING:3, SLIDINGWINDOW:4:15, MINLEN:30). Remaining reads shorter than 30 base pairs were discarded. BWA (v07.17) was used to align quality-filtered reads to the human reference genome (hg38) for the removal of human-derived reads. From 1.9e + 07 ± 2.7e + 06 (mean ± standard deviation) metagenomic reads, 7.8e + 05 ± 8.9e + 05 remained after preprocessing samples. To estimate the taxonomic composition of the nonhuman reads, Kraken 2 (v2.0.7, default parameters) was used with its standard database as the reference. To further control false positive rate, raspir (v1.0.2)[30] was used to support our Kraken 2 results. Only the species that can be detected in at least one sample by raspir were kept (Supplementary Data S6). KrakenUniq (v0.5.7, default parameters)[34] and Centrifuge (v1.0.4, default parameters)[35] were also used to assess the reliability and for comparison of Kraken 2 results. FEAST (v0.1.0, default parameters)[33] was used to check for possible contamination and to estimate the contribution of potential source environments using the cystic fibrosis lung microbiome dataset from three published studies (European Nucleotide Archive [ENA] project IDs PRJEB38221, PRJNA316588, PRJEB32062)[75–77], an oral microbiome dataset (ENA project ID PRJEB28422)[78] and a clinical environmental dataset (ENA project ID PRJNA376580)[79]. Taxonomic assignment of the datasets followed the same pipeline used for our dataset. FEAST results (Supplementary Fig. S2) indicated a dominant source of our samples was the lung microbiome of cystic fibrosis patients with a significantly higher contribution from them than from the oral microbiome and clinical environment. We applied EukDetect (v1.3)[80] to test and confirm that patient samples contained no other fungi such as *Scedosporium* that may be clinically relevant to *A. fumigatus*-associated disease phenotypes such as allergic bronchopulmonary aspergillosis. For functional composition annotation, the MG-RAST (v4.0.3) pipeline was used to assign nonhuman reads to KEGG pathways. R packages vegan (v2.5) and picante (v1.8.2) were used to calculate alpha diversity with a Shannon and phylogenetic diversity index for each sample using the read counts of species. Statistical differences between samples before (*A. fumigatus* -) and after (*A. fumigatus* +) colonization with *A. fumigatus* were obtained by Wilcoxon signed-rank test. For beta diversity, R package coda.base (v0.3.1) was used to calculate the pairwise Aitchison distance for samples using the relative abundance of species. Statistical differences between samples before and after colonization with *A. fumigatus* were calculated by PERMANOVA.

The co-abundance network was constructed based on the relative abundance values of species (prevalence filter: 10%). DGCA (v2.0.0) was used to construct the network from differentially correlated microbial pairs in paired cystic fibrosis samples, comparing before to after *A. fumigatus* colonization (empirical *p*-value ≤ 0.05). Subsequently, MEGENA (v1.3.7) was used to identify co-expressed modules

(module p-value ≤ 0.05) in the constructed network using differentially correlated microbial pairs. To identify molecular functions, we investigated the enrichment of KEGG pathway information (https://www.genome.jp/kegg/pathway.html) by permutation testing to determine if correlations between modules and KEGG orthologies were possible by chance[81]. First, for a given module, all correlation coefficients and p-values for a KEGG orthology and all species in the module were obtained using the Spearman correlation method. The sum of absolute correlation coefficients in the module was then calculated. Following that, a random set of species of the same size as any given module was chosen 1000 times from all species, calculating the sum of absolute correlation coefficients each time. Finally, the sum of correlation values for each module was evaluated. If higher than 95% of the sums of correlation values in the 1000 repeats of randomly selected species, we inferred a significant correlation between modules and KEGG orthologies.

To determine the most likely metabolite abundance profiles associated with our metagenomic samples, we applied the MAMBO algorithm[40]. In brief, MAMBO optimizes a highly correlated metabolic profile with a given relative abundance profile using growth rate simulations of bacterial GEMs associated with a particular metagenomic sample. For 200 raspir-confirmed bacterial species, we downloaded 143 matching bacterial GEMs from the AGORA2 (https://vmh.life/files/reconstructions/AGORA2/version2.01/)[82] and CarveMe collection (https://github.com/cdanielmachado/embl_gems/tree/master/models)[83]. Optimizations were run in a python environment (v3.7) using a high-performance cluster (192 cores, 1 TB RAM). Finally, 929 metabolites were obtained using MAMBO (Supplementary Data S8). For imputation of metabolites, only the metabolites that appeared in less than 80% of the samples were kept. Missing metabolite abundance values in any remaining samples were imputed with MICE (miceRanger, v1.4.0 with m = 1 and maxiter = 50).[84-86]

## Machine learning

Unless otherwise noted, we used the following ML methodology. When group sizes were unbalanced (e.g., unequal numbers with environmental and clinical labels), we randomly sampled 50% of the majority group and oversampled the minority group using ADASYN implemented in R package imbalance (v1.0.2.1). Subsequent feature selection used Boruta (v7.0.0), VSURF (v1.1.0), MUVR (v0.0.973), and sPLS-DA (mixOmics, v6.16.0). These steps were repeated 50 times and selected features and their selection frequency were recorded. Finally, the Extra Trees algorithm from PyCaret (v2.3.2) was run for the feature sets, scanning frequency cutoffs to optimize the cutoff value for ML performance. The best hyperparameters of the Extra Trees model were automatically selected by scikit-optimize (v0.8.1, Bayesian optimization).

## Metabolic and radial growth assays

For conidia production, A. fumigatus was cultivated on Aspergillus minimal medium (AMM)[84] agar plates for 5 days at 37 °C. Conidia were harvested with sterile phosphate buffered saline containing 0.01 % (v/v) Tween 80 and filtered through a 40-μm pore-size cell strainer (BD Biosciences, Germany). Conidia numbers were determined using a CASY cell counter (Roche Innovatis, Germany). Spore purity was assessed and confirmed by microscopy.

Metabolic activity was determined based on a resazurin assay as described[85]. In brief, in a 96-well cell culture microplate (F-bottom, clear) 190 μl AMM/resazurin containing, when indicated, 5 mM glutamine, glycine, phenylalanine or tryptophan, was inoculated with 10 μl of conidial suspension at 12 different concentrations, for 1.6 × $10^6$ to 0.78 × $10^3$ conidia per well. After 18 h, resazurin reduction was measured as fluorescence using a microplate reader (Tecan Infinite M200 Pro) at wavelengths 570 nm (excitation) and 615 nm (emission). Relative metabolic activity was determined as the number of conidia needed as inoculum to reach a fluorescence of 5000 at an optimal gain.

Radial growth was determined with 5 μl conidial suspension ($10^6$ conidia ml$^{-1}$) point-inoculated on AMM or cystic fibrosis media (SCFM2)[86] agar plates supplemented with 5 mM glutamine, glycine, phenylalanine or tryptophan, when indicated. Radial colony growth of A. fumigatus cultures at 37 °C was determined after 24 h using a caliper.

## Targeted metabolomics

A. fumigatus conidia from 10 clinical and 10 environmental strains were cultured in AMM at 3 × $10^6$ per ml for 24 h at 37 °C on a rotary shaker. Aliquots of the sterile, filtered media were frozen in liquid nitrogen and stored at −80 °C until analysis. To adjust for differences in the growth rates of A. fumigatus strains, mycelium dry mass was determined for each sample after lyophilization.

Sample analysis was performed by MS-Omics (Vedbæk, Denmark) as follows. Samples were derivatized with methylchloroformate using a slightly modified version of the Smart et al. protocol[87]. Samples were analyzed in random order by gas chromatography (7890B, Agilent) coupled with a quadropole detector (5977B, Agilent) controlled by ChemStation (Agilent). Raw data were converted to netCDF format using Chemstation (Agilent), before data were imported and processed in Matlab R2018b (Mathworks, Inc.) using PARADISe software[88].

## Statistics and reproducibility

All statistical test procedures are described in the respective methods sections. No statistical method was used to predetermine the sample size. No data were excluded from the analyses. Testing group differences included two-tailed Wilcoxon or t test using a p-value cutoff of 0.05. Multiple test correction as described by Benjamini−Hochberg was applied when appropriate. Gaussian mixture model was used to identify significantly abundant lung species. Functional enrichment of KEGG terms was done using over-representation analysis. Spearman correlation was applied to report correlation of KEGG modules with forced expiratory volume of cystic fibrosis patients (uncorrected $p \leq 0.05$). All codes and data to reproduce the presented results are described in the code and data availability section of this manuscript and are stored in zenodo, github, BioModels and ENA repositories.

## Reporting summary

Further information on research design is available in the Nature Portfolio Reporting Summary linked to this article.

# Data availability

The authors declare that the data supporting the findings of this study are available within the paper and its Supplementary Information files. Information on metabolic models generated in this study is provided in Supplementary Data S1 and S9. All metabolic models are available at the BioModels repository (ID MODEL2211100001). Information on phenotypic growth assays generated in this study is provided in Supplementary Data S2. Information on experimental data including metabolomics, radial growth and metabolic activity generated in this study is provided in Supplementary Data S4. Information on metagenomics of cystic fibrosis samples generated in this study is provided in Supplementary Data S6 and S7. All shotgun metagenomics of sputum from 40 cystic fibrosis patients are available at the European Nucleotide Archive (project ID PRJEB54014). External microbiome datasets analyzed in this study were retrieved from the European Nucleotide Archive (project URLs: https://www.ebi.ac.uk/ena/browser/view/PRJEB38221, https://www.raw.ebi.ac.uk/ena/browser/view/PRJNA316588, https://www.ebi.ac.uk/ena/browser/view/PRJEB32062, https://www.ebi.ac.uk/ena/browser/view/PRJEB28422, https://www.ebi.ac.uk/ena/browser/view/PRJNA376580). Source data are provided with this paper.

## Code availability

The source code and necessary data for results generated in this manuscript are available at zenodo: https://zenodo.org/record/8034128 and github: https://github.com/mohammadmirhakkak/A_fumigatus_GEM/.

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

## Acknowledgements

This work was supported by the Deutsche Forschungsgemeinschaft (DFG—www.dfg.de) CRC/Transregio 124 "Pathogenic fungi and their human host: Networks of interaction" (DFG project number 210879364, subproject INF to M.M., S.S., and G.P. and subproject A1 to A.A.B.), by the DFG funded Germany's Excellence Strategy—EXC 2051—Project-ID 390713860 (A.A.B., A.B., and G.P.) and the German Ministry for Education and Research (82DZL004B1 to S.B.). We thank Natascha Wilker for excellent technical assistance and Chris Tachibana for proofreading our manuscript.

## Author contributions

Conceptualization: M.M., X.C., S.S., G.P. Investigation: M.M., X.C., T.S., A.B., T.H. Methodology: M.M., X.C., Y.N., L.X., S.S., S.B., A.A.B., G.P. Resources: S.B., O.K., A.A.B., G.P. Supervision: G.P. Writing—original draft: S.S., X.C., G.P. Writing—review and editing: all authors.

## Funding

## Competing interests

The authors declare no competing interests.
