## [Peer Review File · Nature Communications]

REVIEWER EXPERTISE

Reviewer #1. Systems biology, genome-scale metabolic models.

Reviewer #2. Fungal genomics, *Aspergillus*.

Reviewer #3. Lung microbial metagenomics, cystic fibrosis.

REVIEWER COMMENTS

Reviewer #1 (Remarks to the Author):

1- A diagram including general information about generated GEMs would help readers quickly receive basic information. This should cover the number of gaps and orphan reactions, number of gene-associated reactions, number of exchange reactions, number of total reactions, number of metabolites, number of genes, number of reactions in core Reactome, and number of reactions in accessory Reactome for each GEM.

2- Template based GEM reconstruction requires a gold standard template, though authors provided phenotypic growth analysis there are no results on quantitative validation of generated models (such as comparing experimental and predicted growth rates, substrate consumption, and product formation rates or comparing metabolic flux analysis with flux balance analysis), which is a crucial step in producing a gold standard model. Given that experimental data for this organism are publicly available (PMID: 12734751) It is highly recommended to include supporting results for quantitative validation of resulted GEMs.

3- The author stated that duplicated reactions were removed from final pan-GEM during pan-GEM reconstruction. considering that, duplicated reactions could result from, 1) allelic variation among selected models. 2) different Isozymes among selected models. In the second case, before removing a reaction, locus tag of the corresponding Isozyme has to be included in the GPR of the remaining reaction. Otherwise, information for such genes will be missed from all the generated models. In this regard, the main question would be whether author has removed all duplicated reactions or isozymes were kept and transformed to a new GPR rule of remaining representative reactions?

4- Reliability of GEM-driven results is highly dependent on the quality of GEMs. Given that, it is highly recommended to include the MEMOTE report (<https://memote.io/>) on generated models.

5- For model validation using phenotypic microarray (biology), growth rate cut-off has not been considered (Supplementary Table S2)

6- For having a reliable *invivo* simulation, publicly available transcriptome on the target organism (<https://www.ncbi.nlm.nih.gov/geo/query/acc.cgi?acc=GSE61974>) could be integrated into GEMs to

generate context-specific models. In this case, it has been proven that during IPA, *Aspergillus fumigatus* would face hypoxia condition, given that, changes in O₂ uptake rate could play an important role in metabolism shift in this organism during pathogenesis. Regarding that, have authors simulated aerobic/anaerobic/microaerophilic conditions during IPA. If not, what is the logic behind it?

7- It has been shown that *lysF* deficiency would result in Lysine auxotrophy in *Aspergillus fumigatus*, and subsequently stops pathogenicity (PMID: 15052376). This might indicate that Lysine availability during IPA becomes a bottleneck in virulence factor synthesis for *Aspergillus fumigatus*. Could GEMs predict any significant metabolic changes comparing wild type and *lysF* mutant?

8- Growth rates shown in fig.4-b are in an unrealistic range compared to the maximum experimental growth rate reported in the literature, it's also in contrast to the growth rate prediction reported in (Supplementary Table S2). How do authors explain this inconsistency? Is it because of unconstrained exchange reactions during MAMBO analysis? If so, why did exchange reactions remain unbound during this analysis?

9- Providing a supplementary table containing all the constraints required for in silico simulation (exchange reactions bounds), and a file containing scripts that have been used for GEMs generation and analysis, would help the reproducibility of results.

10- The criteria for defining the core and accessory genes (e.g. cut-off threshold, method, etc.) are not well defined.

11- It would be more informative to discuss the results of the 'unique' metabolites and genes found in the study.

Reviewer #2 (Remarks to the Author):

This paper aims to investigate the growth dependencies of the important human fungal pathogen *Aspergillus fumigatus* on the lung microbiome. This paper contains a wealth of information, but I do have some concerns about fundamental experimental techniques carried out, mainly for the phenotyping via omnilog. This is (at least, to my knowledge) the first time this has been used on Af I believe? If so, this is extremely novel and should be more of a focus! However, it is not clear whether biological duplicates/triplicates were performed (although I didn't have access to the supplementary information) - replicates are needed to confirm the findings. If they weren't performed, these need to be redone to include replicates. If they were, please include these data. The methods for the omnilog need clarifying: refrained two of the strains because of incompatibility issue for pm04 and two incubation temperature for the phenotype 25 for 7 days and then move it to 37 for 3 days are issues I believe that could impact the results (more below on this). Why was additional dye added to the suspension with the existing dye in the plates? As the omnilog reader system is sensitive to the dye that would cause too much noise in the curves, skewing the results. Also, how was the omnilog

optimised for *Aspergillus*? The omnilog protocol is optimised for yeast, so were there additional optimisation?

Further comments on the omnilog methods include:

Line 149 onwards: 'we manually resolved any incompatibility between our growth data' - needs more information.

Line 531/532: 'we refrained from resolving sulfur growth accuracy' - why? Earlier lines state it would have caused a notable performance drop - that is concerning. Is this an issue with the platform or the mutants, or another issue? Not resolving raises concerns on the findings.

Line 549: why grown at 25 degrees? and then transferred to higher temperature (37 degrees) for 3 days? This could be responsible for growth changes seen

Line 550: why was uracil added?

The bioinformatics analysis seems comprehensive, but I do question why only 252 of the 300 genomes in Barber et al. were mapped to Af293 for identification of metabolically relevant genes - what is the rationale for this reduced number? Surely additional information would yield a 'gold standard' set of genes?

How reliable are GEMs compared to experimental metabolomics?

Overall, whilst this contains a lot of information that would be relevant to the mycology community, I found the paper hard to follow and read, being confusing in a number of places. For instance, the Results are hard to follow, and the sections seem to jump with no flow. I appreciate that word limits are placed on these manuscripts, but I would suggest a hard edit to enable an easier read to enhance these data presented.

Reviewer #3 (Remarks to the Author):

Mirhakkak et al.

A pan-genome resembling genome-scale metabolic model platform of 262 *Aspergillus fumigatus* strains reveals growth dependencies on the lung microbiome.

Metabolic models are a prevalent topic of research. Within this theoretical frame the authors applied methods that are state of the art (lines 445 – 545). According to the reviewer's gut feeling the field still looks like a fancy Glass Bead Game.

Please find below my comments on methods, Figures and Tables that need substantial improvements. In other words, I will not comment on the theoretical framework but will confine myself to the bread-and-butter methodology.

Lines 551-552, lines 682-686. Demonstrate the purity of your spore solution.

Line 554. Provide the source of the phenotypic microarrays.

Lines 565-606.

a. The clinical literature differentiates between

- Airway colonization with *A. fumigatus*,
- ABPA
- aspergilloma
- Invasive aspergillosis.

Please provide the individual patient's diagnosis in a Table in the supplement and link with the respective sputum sample.

b. The authors collected spontaneously expectorated sputa. This mode of sampling is prone to contamination by the oropharyngeal flora. The leading symptom of ABPA is bronchial obstruction. Hence, the mode of drainage of sputum may be shaped by the obstruction of the conducting airways, which may impair comparability of sample pairs collected prior and during *A. fumigatus* detection. Authors should provide the patient's spirometry data at the days of collection.

c. How did you define *A. fumigatus* colonization (line 572)?

d. Sensitivity, specificity, validity and reliability of airway metagenome data critically depend on sample collection and processing (see e.g. Gut Pathogen 2016;8:24; BMC Biol 2014;12:87; Mol Ecol Resourc 2019;19:982-96). Please describe the

- mode of sampling,
- the latent period between sampling and freezing,
- the standard cleaning procedure of the laboratory environment,

- number and handling of negative controls during DNA extraction and library preparation.

Lines 615-617. Why did you use Kraken2? KrakenUniq is a more accurate metagenome binning tool surpassing Kraken and Kraken2. Back up your k-mer approach by an index classifier such as Centrifuge or MetaPhlan3.

Lines 617-618. Low abundance species are meanwhile known to be crucial for the metabolic competence of an airway metagenome. Please process your datasets with a tool that removes false negative and false positive assignments.

Lines 615-618. Unfortunately, reference eukaryotic genomes are often contaminated, making it hard to detect and quantify eukaryotic microbes in shotgun metagenomic data. You may use a tool like EukDetect, which is specifically built to deal with these problems. In the context of *Aspergillus*, one need to know whether patients' samples contained other fungi like *Scedosporium* that are suspected to be clinically more relevant for the ABPA phenotype than *A. fumigatus*.

Line 631. How did you define infection with *A. fumigatus*?

Line 629. The rationale for the thresholds of prevalence and abundance filters should be given. If the dataset is appropriately curated, less prevalent and less abundant taxa can be included.

Figure 3A. *Burkholderia* spp. and particularly *Sphingomonas* spp. are rare members of the CF airway community. Demonstrate that Af positive patients at the Heidelberg CF clinic are more frequently harboring these proteobacteria than patients seen at any other CF clinic in the world.

Figure 3B shows numerous taxa that have not been detected in previous 16S rRNA or metagenome studies of CF airway specimens. Verify these species by another pipeline (Centrifuge, MetaPhlan3) and by a tool that flags false positive and false negative taxonomic assignments. In other words, the datasets need to be curated. The reviewer assumes that the figure shows about 17, 14, 7 and 9 false positive assignments in modules 1, 2, 3, 4, respectively.

Figure 4. Authors should discuss why the first two dimensions in the PCA just explains 13-14% of the variance.

Table S1. ATP demand. Provide error estimates and restrict numbers to physically plausible digits.

Table S2. Quantify the term 'optimal' for growth prediction and explain your terminology in more depth in the Read me section.

Table S3. Metadata of the *A. fumigatus* strains should be given (date of collection, habitat, geographic origin, collector (name or institution)). I did not find this standard information in the authors' ref. 6 (Barber et al., 2021) cited throughout the table.

Table S4. Description. Provide reference for reaction ID, spell out abbreviation (FVA). Provide error estimates for the ratio $\text{mean}(\text{clinical})/\text{mean}(\text{environmental})$. Five or six digits do not make sense.

Table S5 lists the presence of 600 taxa the majority of which never seen before in any CF patient's airways. The dataset must be rigorously curated and checked for false positive assignments and contaminations from other sources (lab, environment, other samples).

Table S6. Were P values corrected for multiple testing?

Reviewer #1 (Remarks to the Author):

1- A diagram including general information about generated GEMs would help readers quickly receive basic information. This should cover the number of gaps and orphan reactions, number of gene-associated reactions, number of exchange reactions, number of total reactions, number of metabolites, number of genes, number of reactions in core Reactome, and number of reactions in accessory Reactome for each GEM.

Response:

We agree that a diagram indeed improves entry into the complexity of our endeavor of strain GEM generation and downstream multi-data driven analyses. Accordingly, we extended the given information in Supplementary Table S3, sheet "afu_strain_metadata" for all our strain-GEMs as well as the base pan-GEM. This includes the number of all and unique genes, reactions and metabolites as well as differentiating reactions further into e.g. core, accessory, orphan, gene-related, exchange, demand or exchange reactions. Additionally, we represented the requested information as a new Fig. 1b. We moved all original subfigures from originally Fig. 1b-f to the updated Fig. 1c-g accordingly.

Moreover, an overview over all strain-GEMs was added as new Fig. 2a. To keep figures concise and topic oriented we split our originally submitted Fig. 2 into two new Figures: Fig. 2, now representing strain-GEM information and a new Fig. 3 representing details of our analysis of clinical and environmental strain-GEMs.

The originally submitted Fig. 3 and 4 became Fig. 4 and 5 accordingly.

The figure caption for the new Fig. 1b reads:

"b Counts of pan-GEM components for included genes, reactions and metabolites."

2- Template based GEM reconstruction requires a gold standard template, though authors provided phenotypic growth analysis there are no results on quantitative validation of generated models (such as comparing experimental and predicted growth rates, substrate consumption, and product formation rates or comparing metabolic flux analysis with flux balance analysis), which is a crucial step in producing a gold standard model. Given that experimental data for this organism are publicly available (PMID: 12734751) It is highly recommended to include supporting results for quantitative validation of resulted GEMs.

Response:

Thank you for the suggestion. Since the experimental growth rate data measured as colony radius in millimeters in the indicated paper cannot be directly converted into growth units commonly used in GEMs (i.e. mmol/grDW/hr) we opted for a more suitable/translatable publicly available data set (Barker et al. 2012) to compare the reported growth rates with the GEM predicted growth rate values. In brief, Barker et al. (2012) performed a batch fermentation culture of the fungus in normoxic and hypoxic conditions. We converted the experimental gr/grDw/hr unit to mmol/grDW/hr for the glucose consumption. The batch fermentation was done in an oxygen-controlled fermenter without indication of the actual oxygen consumption by the fungus in the article and its extended information. Given the glucose uptake rate in normoxic condition (0.25 mmol/grDW/hr opposed to 0.206 mmol/grDW/hr for hypoxic condition), we fitted our O₂ dependent growth curve of our pan-GEM (i.e. tracking achievable growth rate by simulating increasing values of O₂ influx) to the experimental data by calibrating the flux through the non-growth associated maintenance reaction to 0.1 mmol/grDW/hr (see Fig. R1.2 below). We hypothesized that the normoxic oxygen uptake rate must relate to a value within the section of our simulated growth curve, where growth rate cannot be increased anymore and thus does not depend on further

increases of oxygen uptake (0.2+ mmol/gDW/hr, Fig. R1.2). In contrast, under hypoxic conditions (i.e. 1% O₂ available compared to normoxic conditions, cf. Barker et al., 2012) the oxygen uptake is a growth limiting factor. Indeed, after calibrating the maintenance reaction and oxygen uptake given the available glucose sources, we found that the oxygen uptake rates 14 and 0.14 mmol/gDW/hr for normoxic and hypoxic conditions could convincingly resemble the experimentally measured growth rates. The FBA-derived growth rates were 0.01 for hypoxic (compared to experimental growth of 0.011 h⁻¹) and 0.013 mmol/grDW/hr for normoxic conditions (experimental growth = 0.013 h⁻¹). Of note, the calculated maximum oxygen uptake rate of 14 mmol/grDW/hr was well beyond the plateau of necessary oxygen for the maximum feasible growth rate. This indicated that in our simulations *A. fumigatus* appeared to require only a fraction of the available oxygen to convert the available glucose in this setup (cf. Fig. R1.2 below for O₂ dependent growth simulations). We added the calibrated maximum O₂ uptake rate to our pan-GEM and consequently to our strain-GEMs. O₂ uptake in our downstream analyses were in concordance with the determined maximum uptake rate. After calibration of O₂ uptake values, we investigated whether we can recapitulate reported secretion levels for acetate, ethanol and lactate under different O₂ conditions (Barker et al. 2012). The secretion of acetate and lactate increased under low levels of O₂, which we also saw with our pan-GEM simulations (from 0.009 to 0.02 for acetate and 0.006 to 0.008 mmol/grDW/hr for lactate, cf. Supplementary Table S2, new sheet 'Quantitative prediction'). These low secretion values reflected well the experimental data, which also showed low production of these metabolites (0.004/0.015 for acetate, 0.003/0 for ethanol, 0/0.002 mmol/grDW/hr for lactate secretion under normoxic/hypoxic conditions). Of note, steady state assumption-based genome-scale metabolic modeling always shows a theoretical maximum yield given growth and media conditions which in reality might not be reached due to e.g. transient regulatory effects that cannot be captured by metabolic model simulations. This may explain, why under normoxic conditions for lactate and under hypoxic conditions for ethanol our model allowed low theoretical secretion rates, whereas experiments did not show any yield. We also cannot rule out a detection limit in the experimental device given the low observed lactate traces. In light of these simulations our pan-GEM is able to recapture experimentally observed growth under different O₂ conditions, thus adding an orthogonal calibration to our initially used phenotypic microarray data. We are grateful for this remark by the reviewer and added recaptured experimentally determined growth rates by simulations as new Fig. 1h and an additional sheet to include the secretion rate simulations to Supplementary Table S2 to our main manuscript. We added a brief description to our text to include these additional calibrating efforts in the results and methods:

Results: "Finally, we tested how well our pan-GEM predicted oxygen-dependent growth²⁰. After calibrating our model to normoxic growth conditions (0.013 mmol/grams dry weight [grDW]/hr, Fig. 1h), we were able to accurately capture hypoxic growth (predicted 0.010 compared to measured 0.011 mmol/grDW/hr). The same model also predicted the magnitude of experimentally derived secretion rates for acetate (for predicted and measured values, respectively, 0.009 vs. 0.004 for normoxic conditions; 0.020 vs. 0.015 for hypoxic conditions, all mmol/grDW/hr). Measured ethanol and lactate levels were also very low (0.003 mmol/grDW/hr or lower), while the predicted theoretical yield by our simulations was 0.008 mmol/grDW/hr or lower (Supplementary Table S2, Methods). In summary, our refined pan-GEM was able to recapture experimentally assessed oxygen-dependent growth data and predicted secretion rates of assessed metabolites that were comparable to the publicly available growth data."

Methods: "Next, we calibrated our pan-GEM to reflect experimental growth and byproduct secretion rates in glucose-limited environments in hypoxic and normoxic conditions. We retrieved published glucose, acetate, ethanol, lactate, and cell dry weight concentration values from Barker et al.²⁰ and converted these values to plausible units (e.g., mmol/grDW/hr for metabolites and 1/hr for growth) for subsequent GEM analysis (Fig. 1). We constrained our

pan-GEM for the experimental glucose consumption rates (i.e., fixed glucose uptake of 0.250 mmol/grDW/hr for normoxic and 0.206 mmol/grDW/hr for hypoxic conditions) and calibrated flux through the nongrowth-associated maintenance reaction with different O₂ uptake rates resembling hypoxic to normoxic conditions. The data fitted best when setting the flux through the nongrowth-associated maintenance reaction of the pan-GEM to 0.1 mmol/grDW/hr (Supplementary Table S9 and S10)."

Fig. 1h caption: "h Experimental values compared to simulated growth rate values under normoxic and hypoxic conditions (Supplementary Table S2 has experimental and simulated secretion values)."

*Figure R1.2: Growth rate simulations of *A. fumigatus* pan-GEM for different O₂ uptake rates and available glucose as described in Barker et al. (2012).*

3- The author stated that duplicated reactions were removed from final pan-GEM during pan-GEM reconstruction. considering that, duplicated reactions could result from, 1) allelic variation among selected models. 2) different Isozymes among selected models. In the second case, before removing a reaction, locus tag of the corresponding Isozyme has to be included in the GPR of the remaining reaction. Otherwise, information for such genes will be missed from all the generated models. In this regard, the main question would be whether author has removed all duplicated reactions or isozymes were kept and transformed to a new GPR rule of remaining representative reactions?

Response:

We apologize for our imprecise description of this important aspect. In fact, the group of duplicated gene-associated reactions in all cases was initially assigned to the same set of genes. Hence, duplicated reactions and metabolites were identical and appeared while we aimed at generating a most comprehensive pan-GEM due to usage of different IDs for the same metabolite and reaction in different available draft reconstructions. For instance, the following duplicated reactions initially occurred in the draft reconstruction:

*RXN9501MetaCyc: C00019 + Cluster6767 <=> C00021 + Cluster4007 (grRule: AFUA_5G02640)
r1078ANIDULANS: C00019 + C20445 <=> C00021 + C03944 (grRule: AFUA_5G02640)*

The duplicated metabolites are Cluster6767 | C20445 and Cluster4007 | C03944. The names of the duplicated metabolites are alphabetically different and as follows:

Cluster6767 -> dihydrosterigmatocystin [cytosol]
C20445 -> dihydrosterigmatocystin [cytosol]
Cluster4007 -> dihydro-O-methylsterigmatocystin [cytosol]
C03944 -> dihydro-O-methylsterigmatocystin [cytosol]

RXN9501MetaCyc was removed from the draft GEM and *r1078ANIDULANS* was kept accordingly. After the curation of the pan-GEM, the reaction IDs were changed in order to have a unified nomenclature for the network reactions in our final pan-GEM. Accordingly, 'r1078ANIDULANS' was converted to 'R45' which can be found in the pan-GEM reactions (Supplementary Table S9 [formerly S7]).

Therefore, removing the duplicated reactions and metabolites did not result in losing gene information from the pan-GEM and accordingly for reconstruction of the strain GEMs. We adapted our text to improve our description:

"Filtering duplicate reactions was necessary because our initial merge of metabolic information included reactions and converted metabolites from different sources. Only when reactions or metabolites had different naming conventions did we reduce to a single naming scheme. When different isozymes encoded the same metabolic reaction, we included all alternative genes as OR relationships (instead of AND) in the genes-to-protein rule of the affected metabolic reaction and removed the duplicate reaction."

4- Reliability of GEM-driven results is highly dependent on the quality of GEMs. Given that, it is highly recommended to include the MEMOTE report (<https://memote.io/>) on generated models.

Response:

*Indeed another very important note, thank you. We acknowledge this comment from the reviewer that resulted in additional manual curation and subsequent quality improvement of our reconstruction. We benchmarked the performance of our submitted pan-GEM of *A. fumigatus* by the MEMOTE pipeline. Before any curation efforts, the reported MEMOTE score of our initial pan-GEM was 30%. To improve the MEMOTE score and thus the quality of our pan-GEM, we substantially curated our pan-GEM originally and additionally during the revision. Importantly, during our revision our efforts did not require to change the pan-GEM's stoichiometric matrix, which was important as otherwise downstream analyses might have been affected. After our curation efforts, the overall MEMOTE score increased to 78%. MEMOTE's consistency score reported 72% for our reconstruction. The html file of the summarized MEMOTE test results is available in our github repository¹. In addition, we compared the MEMOTE report of our GEM with the consensus Yeast v8.6.2 GEM (Lu et al. 2019), one of the very few highly curated fungal GEMs with updates over more than a decade. The summarized MEMOTE score is reported by the consensus Yeast team at github². Interestingly, with 70% the MEMOTE score for Yeast v8.6.2 was lower compared to our curated GEM as was the reported consistency score of 55%. This exemplifies in part the difficulty and challenges in fulfilling all MEMOTE conventions for fungal GEMs compared to e.g. bacterial GEMs where data availability and standardisation is much more progressed.*

Nevertheless, given our additional efforts we could improve the MEMOTE score of our pan-GEM to a convincing and ready-to-use level. Of note, our curation steps including standardized naming nomenclature and ensuring network consistency diluted to all subsequently generated strain-

¹ https://github.com/mohammadmirhakkak/A_fumigatus_GEM/blob/main/memote_report.html

² https://sysbiochalmers.github.io/yeast-GEM/release_report.html

GEMs, which thus benefit from the considerable high score of our pan-GEM. We added a note to applying MEMOTE to our methods section accordingly:

“Last, we checked compatibility of our model with metabolic modeling standards by running MEMOTE tests⁶². The overall MEMOTE score we achieved was 78%, with a 72% consistency score, which is in the range reported for the most-curated yeast GEM (https://sysbio-chalmers.github.io/yeast-GEM/release_report.html). Simulation scripts for all analyses are at github: https://github.com/mohammadmirhakkak/A_fumigatus_GEM/, which also holds the MEMOTE report (github folder: blob/main/memote_report.html).”

5- For model validation using phenotypic microarray (biology), growth rate cut-off has not been considered (Supplementary Table S2)

Response:

We considered 0.001 mmol/grDW/hr as a minimum growth rate for considering growth vs. no growth. Since we aimed to capture also very low, but non-zero growth rates, we set this value below the hypoxic growth rate in glucose-limited media as reported by Barker et al. (2012), which we used for quantitative prediction (cf. point 2 of this reviewer above). We stated the considered cut-off related to the “Prediction” in the “description” sheet of the Supplementary Table S2 and added a further note to the Methods section of our manuscript:

“We considered 0.001 mmol/grDW/hr as an absolute minimum for growth rate to account for reported low-oxygen growth conditions, e.g., in glucose-limited media²⁰.”

6- For having a reliable in vivo simulation, publicly available transcriptome on the target organism (<https://www.ncbi.nlm.nih.gov/geo/query/acc.cgi?acc=GSE61974>) could be integrated into GEMs to generate context-specific models. In this case, it has been proven that during IPA, *Aspergillus fumigatus* would face hypoxia condition, given that, changes in O₂ uptake rate could play an important role in metabolism shift in this organism during pathogenesis. Regarding that, have authors simulated aerobic/anaerobic/microaerophilic conditions during IPA. If not, what is the logic behind it?

Response:

*We acknowledge that indeed O₂ availability might be important for metabolism in pathogenesis. However, it has been shown that at least gene expression data may not necessarily reflect metabolic activity (cf. e.g. Glanemann et al. 2002 for disparity in the model organism *C. glutamicum*) challenging the reliability of the addition of transcriptomic data. Moreover, integrating transcriptomic data towards data-driven context-specific models has been debated before and showed among others, very diverse outcomes for different algorithmic approaches. In fact, predictions obtained by simple flux balance analysis using growth maximization and parsimony criteria were as good or better than those obtained after using transcriptome integrating methods, which impact notably the received and to be expected prediction capabilities (Machado & Herrgård, 2014, Opdam et al., 2017). Therefore, we did not believe that generating context-specific networks in the context of our manuscript would improve the quality of our first collection of reconstructed strain-GEMs of *A. fumigatus*, but contrarily would raise further questions about the validity of the (whichever chosen) applied approach. The ability of our pan-GEM to predict the growth in normoxic and hypoxic conditions was shown above and explained in the response of the second comment of reviewer 1. For further downstream analyses we relied on normoxic growth conditions, since our experimental data derived from cystic fibrosis patients originated*

from lung tissue via sputum and did not resemble established IPA. We added this notion to our results as follows:

Results: “For the remainder of our analysis we assumed normoxic growth conditions unless otherwise noted.”

7- It has been shown that *lysF* deficiency would result in Lysine auxotrophy in *Aspergillus fumigatus*, and subsequently stops pathogenicity (PMID: 15052376). This might indicate that Lysine availability during IPA becomes a bottleneck in virulence factor synthesis for *Aspergillus fumigatus*. Could GEMs predict any significant metabolic changes comparing wild type and *lysF* mutant?

Response:

*In the context of our Biolog phenotypic microarray data, deletion of *lysF* (AFUA_5G08890) did not result in lysine-auxotrophy. As shown in Supplementary Table S2, *A. fumigatus* grew in well C09 on plate PM1 where glucose resembled the carbon source and lysine did not exist. The same occurred in other wells for testing different sources where Lysine is lacking (Supplementary Table S2).*

*Following the reviewer’s suggestion, we constrained the pan-GEM with respect to a minimal media and performed FVA across all reactions simulating the *lysF* knock-out and compared it with the same simulation in wild-type condition. However, we observed no difference between the two simulated conditions suggesting that this observation cannot be reflected using minimal media. We note this observation in our results:*

*“Of note, we did not observe lysine-dependent growth cessation with our phenotypic microarray data of the *lysF* mutant strain (Supplementary Table S2) suggesting a media influence on the environment for growth and virulence¹⁹.”*

8- Growth rates shown in fig.4-b are in an unrealistic range compared to the maximum experimental growth rate reported in the literature, it’s also in contrast to the growth rate prediction reported in (Supplementary Table S2). How do authors explain this inconsistency? Is it because of unconstrained exchange reactions during MAMBO analysis? If so, why did exchange reactions remain unbound during this analysis?

Response:

*For the FBA simulations, the exchange reactions of the GEMs were constrained using the metabolite levels predicted by MAMBO before and after colonization of *A. fumigatus*. However, the overall predicted metabolite levels by MAMBO were an order of magnitude higher than the minimal media used in Supplementary Table S2. Moreover, the predicted metabolome environment by MAMBO contained a wide variety of metabolites (Supplementary Table S8 [formerly S6]) and was not limited to the low number of nutrients used in each Biolog corresponding simulation case in Supplementary Table S2 (minimal media plus a target nutrient). Nevertheless, constraining the feasible uptake by the MAMBO derived metabolite distributions had substantial impact on achieved growth rates as can be compared to simulations when uptake rates would generally have been unconstrained (Fig. R1.8)*

Figure R1.8: Predicted growth rates of clinical GEMs by unconstrained uptake rates with respect to the MAMBO predicted available metabolite profiles. Blue: before *A. fumigatus* colonization; orange: after *A. fumigatus* colonization.

We added an explicit note of the elevated growth to our Results section:

“Notably, the achieved absolute growth rates were higher than previously reported growth rates on, for example, minimal media²⁰, but nevertheless indicated beneficial growth after functional output from the microbial community changed upon *A. fumigatus* colonization.”

9- Providing a supplementary table containing all the constraints required for in silico simulation (exchange reactions bounds), and a file containing scripts that have been used for GEMs generation and analysis, would help the reproducibility of results.

Response:

We agree that this will be valuable for further modeling efforts. We added Supplementary Table S10 containing individual sheets for the constraints used in each simulation. This includes constraints for reproducing the results in Fig. 1f (FBA associated with growth data), Fig. 1g (FBA associated with gene essentiality data), Fig. 1h (FBA associated with normoxic and hypoxic growth data), Fig. 3c (FBA associated with metabolite consumption by the strain GEMs), Fig. 3d (FVA associated with strains’ growth in minimal media), Fig. 5b (FBA associated with growth of the clinical GEMs in MAMBO-derived media), and Fig. 5c (FVA associated with growth of the clinical GEMs in MAMBO-derived media). Furthermore, we deposited all our code to Github³ alongside our manuscript.

We added this information to our methods:

“Simulation scripts for all analyses are at github: https://github.com/mohammadmirhakkak/A_fumigatus_GEM/, which also holds the MEMOTE report (github folder: blob/main/memote_report.html). Simulation specific constraints are in Supplementary Table S10.”

³ https://github.com/mohammadmirhakkak/A_fumigatus_GEM

10- The criteria for defining the core and accessory genes (e.g. cut-off threshold, method, etc.) are not well defined.

Response:

We apologize for our insufficient description of our definition of the core and accessory Genes and Reactome. We modified the respective paragraph in the methods section:

*“By mapping the genomes for 252 strains to the Af293 *A. fumigatus* reference genome annotation, we identified metabolically relevant genes by requiring at least 95% sequence identity under the rationale that high sequence identity preserves metabolic function. Small deviations from the chosen sequence identity threshold did not change the results. The similarity analysis was done by BLAST analysis of protein sequences using diamond (v0.9.24.125)⁶³. The presence or absence of metabolic reactions was deduced for each strain using the associated gene-protein-reaction rules from the pan-GEM and the relevant identified genes. The metabolic core comprised reactions and genes that occurred in all strains with other reactions and genes defining, respectively, the accessory reactome and genome.”*

11- It would be more informative to discuss the results of the ‘unique’ metabolites and genes found in the study.

Response:

*We apologize if we did not fully grasp the intention of this comment. In case the reviewer means metabolites and genes that were not known to be relevant for *A. fumigatus* metabolism before, we note that our modeling endeavor was not tailored towards identifying novel *A. fumigatus* relevant metabolic compounds, but rather to provide the most comprehensive metabolic knowledge (molded and summarized into our pan-GEM) which we further used to derive strain-specific *A. fumigatus* GEMs of clinical and environmental origin. Whenever appropriate and when we describe the number of metabolites or genes we modified the text by explicitly adding information about the total and unique number of GEM associated genes and metabolites.*

Reviewer #2 (Remarks to the Author):

This paper aims to investigate the growth dependencies of the important human fungal pathogen *Aspergillus fumigatus* on the lung microbiome. This paper contains a wealth of information, but I do have some concerns about fundamental experimental techniques carried out, mainly for the phenotyping via omnilog. This is (at least, to my knowledge) the first time this has been used on Af I believe? If so, this is extremely novel and should be more of a focus!

Response:

We would like to thank the reviewer for these positive remarks about the novelty and richness of our study.

1- However, it is not clear whether biological duplicates/triplicates were performed (although I didn't have access to the supplementary information) - replicates are needed to confirm the findings. If they weren't performed, these need to be redone to include replicates. If they were, please include these data.

Response:

We apologize for the unavailability of our supplementary material in case this was due to us. We uploaded seven Supplementary Tables and they should have been available via the journal's management system. During this revision process 3 further Supplementary Tables were added for a total of 10. Regarding the phenotypic microarray experiments, these were done either in triplicates or duplicates. We completed full triplicates for the sulfur plates to improve our accuracy for these resources also in response to further comments below. The exact number of replicates per tested C, N, P or S source is shown in the "Description" sheet of Supplementary Table S2.

2- The methods for the omnilog need clarifying: refrained two of the strains because of incompatibility issue for pm04 and two incubation temperature for the phenotype 25 for 7 days and then move it to 37 for 3 days are issues I believe that could impact the results (more below on this).

Response:

Indeed we originally refrained from using two strains for our model calibration with respect to sulfur metabolites. In order to address all points raised by the reviewers and the editor we reinvestigated our pan-GEM and improved this weakness. After adding full triplicate information for the PM4 plate alongside a careful model re-evaluation, we used also all information of all strains for PM4 and substantially improved the accuracy for this plate (please cf. the updated Fig. 1f and Supplementary Table S2 of this revision).

The incubation temperature was changed from 25°C to 37°C to mimic the change in temperature that occurs in the transition from the environment to the human host as used before (Slesiona, Gressler et al., 2012, Slesiona, Ibrahim-Granet et al., 2012, and Lambou et al., 2010).

We modified our text to account for the improvements and better describe the necessity of our experimental design accordingly:

Results: "The pan-GEM achieved 79% and 82% compatibility for, respectively, the tested phosphorus and sulfur sources (Fig. 1f, Methods)."

Methods:

Deletion of: "Resolving growth compatibility for two of our mutants (Δ niaD and Δ lysF, cf. Biolog phenotypic microarray) on sulfur would have caused a notable performance drop in the overall growth prediction for all investigated growth media and gene essentiality performance. Since growth accuracy on sulfur was very good for the remaining wild-type and two mutant strains and because optimizing growth on carbon and nitrogen sources was very good over all mutant data, we refrained from resolving Δ niaD and Δ lysF sulfur growth accuracy (Fig. 1b)."

Addition of: "The incubation temperature was changed from the initial 25°C to 37°C for the phenotypic microarray experiments to mimic the change in temperature that occurs in the transition from the environment to the human host as used before⁶⁴⁻⁶⁶."

3- Why was additional dye added to the suspension with the existing dye in the plates? As the omnilog reader system is sensitive to the dye that would cause too much noise in the curves, skewing the results. Also, how was the omnilog optimised for Aspergillus? The omnilog protocol is optimised for yeast, so were there additional optimisation?

Response:

The Biolog phenotypic microarray plates did not contain the dye themselves. Instead, it was added to the media mix because variable dyes were used with the same base plate based on the input species being tested. The dye used (Redox dye D) was used based on consultation with a company representative as it was the dye that showed the best and most robust results with fungi. We modified our methods description to:

"Phenotypic microarrays were performed using Biolog Phenotypic Microarray plates PM1, PM2, PM3, and PM4 (Biolog Inc., Hayward, CA, USA) prepared following the manufacturer's protocol for filamentous fungi with the modification of 0.16 ml of Biolog Redox Dye D added to the master mix of each plate to ensure robust quantification of metabolic fungal activity."

Further comments on the omnilog methods include:

4- Line 149 onwards: 'we manually resolved any incompatibility between our growth data' - needs more information.

Response:

*We apologize for the unclear explanation of this part in the manuscript. This sentence aimed to explain the curation of the pan-GEM to increase its quality for more accurate predictions. The predictions of our draft reconstruction did not reflect the capability of *A. fumigatus* to grow on a number of different carbon, nitrogen, phosphor, and sulfur sources as identified by our phenotypic microarray experiments. Hence, to improve predictability of our model, we included more curation steps, e.g. gap-filling approaches to adjust the GEM predictions to the experimental data. We elaborated and improved this part in the manuscript with the addition of the following methods section, which we reference in the mentioned sentence now:*

"When our data showed growth in a certain condition that our pan-GEM could not predict, we investigated the direction of related reactions, potential gaps in the metabolic network, and connectivity of the involved compartments. When our data showed no growth under a given condition but our GEM predicted a nonzero growth rate, we investigated if the metabolic reactions in our initial, draft pan-GEM were not associated with the genome and needed to be removed. We used similar curation efforts to resolve incompatibilities in the gene-essentiality data."

5- Line 531/532: 'we refrained from resolving sulfur growth accuracy' - why? Earlier lines state it would have caused a notable performance drop - that is concerning. Is this an issue with the platform or the mutants, or another issue? Not resolving raises concerns on the findings.

Response:

We are very thankful to the reviewer for this constructive comment. We investigated the issue from two aspects: I) capability of our pan-GEM to predict growth/ no growth based on the growth data; II) precision of produced phenotypic microarray growth data.

I) We investigated our pan-GEM whether further curation can increase its accuracy to predict the growth/ no growth conditions. We found Aminoacetaldehyde in cytoplasm causing a blocked reaction, i.e., R2596: Taurine, 2-oxoglutarate: O₂ oxidoreductase (sulfite-forming); Taurine + 2-Oxoglutarate + Oxygen  Sulfite + Aminoacetaldehyde + Succinate + CO₂, in Taurine and Hypotaurine metabolism (Supplementary Table S9 [formerly S7]).

*According to the KEGG database, the compound is associated with three enzymes, i.e., 1.1.1.276, 1.1.1.387, and 1.14.11.17. Our reconstruction encompasses only 1.14.11.17 which is only responsible for catalyzing the identified blocked reaction. As the other two EC numbers do not occur in the reconstruction and 1.14.11.17 is a specific enzyme, we concluded that there is no missing reaction metabolizing Aminoacetaldehyde in our network through different metabolic routes. Moreover, searching in the literature, we did find experimental results indicating secretion of the compound from *A. fumigatus*.*

*Considering the aforementioned inspections and following the protocol for generating high-quality genome-scale metabolic models (Thiele & Palsson 2010), we added a Demand reaction for this particular compound to our reconstruction, i.e., DM_C06735[c], to make the reaction able to carry flux. Of note, the same demand reaction is also present in other high quality reconstructions available in the BiGG database (<http://bigg.ucsd.edu/>) such as different *Escherichia coli* strains and *Shigella* species GEMs. This curation resulted in enhancement of the GEM to predictions for sulfur sources by 8% in average for all four mutants and the wild-type (73% accuracy). The accuracy for the problematic mutations, i.e., Δ lysF and Δ niaD, increased to 46%. Since this value was still below 50% and thus inadequate, we checked the precision of our Biolog experiments described in bullet point II) below. Of note, we regenerated all the strain GEMs and repeated all related analyses and figures due to this modification in the pan-GEM. Although a number of figures had to be recomputed, this did not affect the main results and conclusions in our study. The only notable difference we identified were different sets of machine learning selected reaction features to differentiate clinical from environmental strains affecting both the decision tree (updated Fig. 3c) or the necessary number to hold a high precision-recall curve (cf. updated Fig. 3d) which reduced to 21 reactions (cf. Supplementary Table S5 [formerly S4]). However, our main observation that chorismate and amino acid-associated reactions are involved in the predicted differentiation of simulated strain-GEMs still holds. Hence, our updated prediction appeared robust with respect to these metabolic components, despite the involved sensitivity and dependence of predictions on even a few, but apparently relevant model changes. Considering the new arrangement of the figures in this revision, the following figures were changed: Fig. 1d-f, Fig. 2b,c, Fig. 3a,c,d, Fig. 5b-d. We updated our text for the machine learning descriptions accordingly:*

*“Notably, the presence of chorismate lyase alone allows to categorize 93% of all strain-specific GEMs correctly. Chorismate lyase activity is linked to differential activity in the shikimate pathway, which is associated with virulence in *A. fumigatus*^{24,25}. Combining the ability to convert chorismate and glutamine to anthranilate, pyruvate and glutamate, with amino acid and energy metabolism-associated conversions of methionine, succinate or tryptamine, and the ability to take up and grow on aspartic acid, appeared sufficient for strain origin classification. These reactions yielded metabolic discriminators that were complementary to the sole presence/absence statistical analysis of metabolic reactions in our strain-specific GEM collection (Fig. 3a,b). [...] In addition to previously*

highlighted chorismate-associated reactions, the ML-model also selected features associated with amino acid and energy pathways, especially in the mitochondrial compartment. These included, for example, homoserine succinate-lyase, ribulose-phosphate 3-epimerase and succinate:CoA ligase, suggesting the contribution of altered amino acid and energy metabolism to differentiation of clinical and environmental *A. fumigatus* strains.”

II) In order to increase the precision of our experiments, we completed third replicates for all sulfur source associated Biolog experiments for all used strains (except MET2 which was already done in triplicates in the original submission, cf. response to point 2, reviewer 2 above and Supplementary Table S2). Consequently, this includes full triplicates of tested sulfur sources in plate PM04 for Af293, CEA17, Δ lysF, and Δ niaD in our revision. Next, we repeated our statistics with Dunnett's tests to compare respective negative controls with all the tested sulfur sources per strain. Significantly higher growth signals than negative controls were taken as growth conditions (p -value ≤ 0.05). Adding these complementary experiments improved the accuracy of our GEM prediction for sulfur sources to a convincing level of 82% on average across all the mutants and the wild-type. The respective accuracy for Δ lysF and Δ niaD increased to 77% and 70%, respectively. We subsequently incorporated all the results for the sulfur sources and updated the new Fig. 1f and simplified our results description accordingly:

Results: “The pan-GEM achieved 79% and 82% compatibility for, respectively, the tested phosphorus and sulfur sources (Fig. 1f, Methods).”

6- Line 549: why grown at 25 degrees? and then transferred to higher temperature (37 degrees) for 3 days? This could be responsible for growth changes seen

Response:

We adhered to this setup, since it reflects best the knowledge of the community in letting *Aspergillus* strains pre-grow in 25°C, because it mimics the transition from environment to human lung. Growing the strains this way also allowed us to standardize the results with how strains were grown prior to infections using a mouse model. Please see also Slesiona, Gressler et al. (2012), Slesiona, Ibrahim-Granet et al. (2012), and Lambou et al. (2010).

We modified our methods description to better justify our experimental setup with showing our phenotypic microarray data for the first time (cf. response to point 2 of Reviewer 2):

“We switched the temperature from the initial 25°C to 37°C to adhere to community-acquired standards^{64–66}, mimic the change in temperature from environment to human host, and prevent fungal filamentation during growth experiments.”

7- Line 550: why was uracil added?

Response:

One of the mutants (Δ pyrG) is a uridine auxotroph and required supplementation with uracil to grow. For standardization, all strains were grown on the same media of Malt agar containing uracil. We added this information to our methods:

“Uracil supplementation was required for growth of the CEA17 pyrG⁻ strain (a uridine auxotroph) and was added to all media to ensure comparable growth.”

8- The bioinformatics analysis seems comprehensive, but I do question why only 252 of the 300 genomes in Barber et al. were mapped to Af293 for identification of metabolically relevant genes - what is the rationale for this reduced number? Surely additional information would yield a 'gold standard' set of genes?

Response:

We used our bioinformatic approach to build and cross-check 48 additional strain-GEMs with our 252 GEMs. Of note at the time of creating our strain-GEMs and subsequent studies we did not have access to these 48 additional genomes from different locations outside of Germany, which were added during the revision process of our Barber et al. (2021) study. Following your comment, we reconstructed the GSMMs of the remaining 48 strains and we analyzed whether they would yield or improve separation by comparing the Jaccard distance (Fig. R2.8) of the affected metabolic reactions to all other 252 models in resemblance of our hierarchical clustering of Jaccard distances in Fig. 2c.

Figure R2.8: Jaccard distance heatmap across considered GEMs from German landsides compared to further international strains.

We neither observed any improvement in differentiating clinical from environmental strains nor did we see a separate cluster of these additional strain models by Jaccard distance over all metabolic reactions. The notion of a gold standard of a metabolically relevant gene set is certainly valuable. However, we identified only one additional metabolically relevant gene (AFUA_6G11210, 3-oxoacyl-(acp) reductase), which did not yield any new metabolically relevant reaction activity, as isozymes exist in the already existing 252 models. Henceforth, given that reaction-wise these models were not distinct from our previous 252 strain-GEM set and the metabolically relevant gene gold-standard was virtually not affected, we refrained from including these additional models in our setup. We explicitly mention metabolically relevant genes in our discussion now: “This strain-specific *A. fumigatus* GEMs platform is publicly available (BioModels ID MODEL2211100001) for investigating the metabolically relevant *A. fumigatus* geneset and its impact on the metabolic diversity influencing growth rate capabilities, metabolic adaptation and pathogenicity in this important human fungal pathogen.”

9- How reliable are GEMs compared to experimental metabolomics?

Response:

In fact, genome-scale metabolic modeling became a serious alternative to e.g. kinetic modeling for host and disease modeling in light of limited knowledge of intricate metabolic mechanics and parameters. Without requiring kinetic parameters and solely based on stoichiometric information, metabolic modeling enables analyses at the genome-wide scale for simulating and predicting metabolism. It could celebrate numerous victories to not only improve product yield with simple bacterial organisms, but also to understand metabolic pathway deterioration – and potential recovery – in human disease (cf. Yang et al., 2018, Björnson et al., 2015, Agren et al., 2014, or Wu et al., 2017 to name a few).

To the best of our knowledge the present study is the first work that tried putting GEMs on strain level resolution for an important fungal, and thus eukaryotic, pathogen. As with other pioneering works, this came with a number of obstacles that needed to be overcome such as high quality GEM generation for the fungus, since neither an adequate base (or template) model nor a sufficient data or even established naming convention was available when we started our work. During our revision we substantially further improved the accuracy of our GEMs for predicting phenotypic growth data. Furthermore, following the excellent suggestion of Reviewer 1 we evaluated the quality of constructed GEMs using the MEMOTE pipeline. MEMOTE is a software suite ensuring metabolic models adhere to metabolic modeling community standards. Our pan-GEM, of which all strain-specific GEMs are derived, scored 78% and outperformed the consensus Yeast v8.6.2 GEM (Lu et al. 2019), a highly curated fungal metabolic model (compare answer to reviewer 1, point 4).

One of our findings when we reconstructed the 252 strain-specific models was the high variability observed in the amino acid potential (see section “A. fumigatus strains show notable accessory reaction content”). Therefore, we cultivated 20 A. fumigatus strains on A. fumigatus minimal media for this revision. By acquiring and analyzing targeted amino acids found in the resulting supernatants after 24 hr, we found a considerable variability of metabolites that reflect the metabolic variability of at most 40% shared metabolic reactions across all strain-GEMs. We added this information to Supplementary Table S4 and in our results:

“The large variability among strains in amino acid metabolism was confirmed by cultivation and targeted metabolomics profiling of 20 A. fumigatus strains (Supplementary Table S4).”

10- Overall, whilst this contains a lot of information that would be relevant to the mycology community, I found the paper hard to follow and read, being confusing in a number of places. For instance, the Results are hard to follow, and the sections seem to jump with no flow. I appreciate that word limits are placed on these manuscripts, but I would suggest a hard edit to enable an easier read to enhance these data presented.

Response:

We thank the reviewer for this remark which certainly may affect how readers perceive our work. During this revision we heavily modified our text, including clarification of multiple sections in response to this and the other reviewers. In addition, we included a professional for proofreading into our revision process and modified our manuscript substantially without changing the content. We added our gratitude for professional editing towards Chris Tachibana for doing so in the acknowledgements.

Reviewer #3 (Remarks to the Author):

Mirhakkak et al.

A pan-genome resembling genome-scale metabolic model platform of 262 *Aspergillus fumigatus* strains reveals growth dependencies on the lung microbiome.

1- Metabolic models are a prevalent topic of research. Within this theoretical frame the authors applied methods that are state of the art (lines 445 – 545). According to the reviewer's gut feeling the field still looks like a fancy Glass Bead Game.

Response:

We would like to thank the reviewer for finding our methodology state of the art. We understand, the in silico aspects of metabolic modeling may appear complex. We are confident, however, that they have their merits. Not only did we reconstruct high quality models using vast amount of data for calibration (that may be further used by the scientific community for a number of different applications), but we also showed translational aspects by combining in silico prediction and experimental data.

Our simulations pinpointed to considerable metabolic variations of the strains, both shown by differences of the individual strain-GEMs and thus their metabolic capabilities. We showed from multiple angles the importance of aromatic amino acids and suggested known and novel metabolic targets for potential drug interventions. By investigating fungal growth on minimal, but also on cystic fibrosis-mimicking media we finally could show that clinical strains may be capable to modulate their lung microbial environment towards improved growth. Towards this aim we included e.g. analysis of MAMBO-derived metabolite profiles, FVA based flux activity analysis as well as amino acid-supplemented growth experiments. Notably, we added tryptophan growth and metabolic activity experiments given its reported importance for virulence and the closely connected biosynthetic anabolic pathway to phenylalanine, which we already investigated in our original submission (Zelante et al. 2021, Choera et al., 2018).

*Moreover, during the revision of our manuscript we also investigated clinical metadata of our cystic fibrosis samples including the forced expiratory volume (FEV) parameter. This readout for the functional capacity of the lung correlated positively with module 2 of our identified lung microbial species confirming that the in silico predicted metabolic dependencies of *A. fumigatus* by lung bacteria translate to actual worsening of the patient health status. Therefore, the integration of metabolic modelling with shotgun metagenomics data allowed us to support our initial hypothesis that identifying drug targets based on lethality data from in vitro gene deletions in *A. fumigatus* is a risky strategy since *A. fumigatus*'s growth can be probably rescued by obtaining the necessary metabolites from the lung bacteria. Therefore, multimodal strategies targeting both metabolic genes as well as transporters of *A. fumigatus* would probably be more efficient drug interventions (Discussion, lines 458-466).*

In summary, we believe metabolic model simulations at the genome-scale are far away from being a glass bead game. Quite the contrary, they can be used to shed light on metabolic components that otherwise remain infeasible to study by experiment alone. Given our extensive efforts for this revision we hope that we could address all the reviewer's requests. Our addition of further analyses and experimental data substantially strengthened our manuscript and we are grateful for the reviewer's thoughts on improving our manuscript.

Please find below my comments on methods, Figures and Tables that need substantial improvements. In other words, I will not comment on the theoretical framework but will confine myself to the bread-and-butter methodology.

2- Lines 551-552, lines 682-686. Demonstrate the purity of your spore solution.

Response:

Fig. R3.2 is the microscopic image demonstrating the spore purity of A. fumigatus in our experiments. We added a notion to both methods sections:

“Spore purity was assessed and confirmed by microscopy.”

Figure R3.2: Microscopic image of A. fumigatus conidia suspension.

3- Line 554. Provide the source of the phenotypic microarrays.

Response:

We apologize for this missing information. The phenotypic microarrays were purchased from Biolog Inc. (Hayward, CA, USA). Adapted methods description:

“Spore solutions were adjusted to a transmittance of 75%. Phenotypic microarrays were performed using Biolog Phenotypic Microarray plates PM1, PM2, PM3, and PM4 (Biolog Inc., Hayward, CA, USA) prepared following the manufacturer’s protocol for filamentous fungi, including the addition of 0.16 ml of Biolog Redox Dye D to the master mix of each plate to quantify fungal metabolic activity.”

4- Lines 565-606.

The clinical literature differentiates between

- Airway colonization with *A. fumigatus*,
- ABPA
- aspergilloma
- Invasive aspergillosis.

Please provide the individual patient's diagnosis in a Table in the supplement and link with the respective sputum sample.

Response:

*We thank the reviewer for the remark and we added this information accordingly to our Supplementary materials (Supplementary Table S7, updated order). Of note, all patients were considered as colonized with *A. fumigatus*, if we could cultivate the fungus from the sputum samples. Further disease classification was neither tracked, nor was it necessary for our purposes, where we aimed at exploring the lung microbiome affected by *A. fumigatus* presence (or not). ABPA, aspergilloma and pulmonary invasive aspergillosis are certainly important manifestations of *A. fumigatus*-driven disease. Since we identified first colonisations with *A. fumigatus* we considered it very unlikely, our patients suffered from aspergilloma or IPA. We could not rule out ABPA due to the lack of tracked data and hence, we opted for categorizing these patient samples as positive *A. fumigatus* culture. Elucidating these disease types was out of the scope of our current manuscript, which focused on providing and applying our strain resolved *A. fumigatus* genome-scale metabolic models, but warrants further studies in the future.*

5- The authors collected spontaneously expectorated sputa. This mode of sampling is prone to contamination by the oropharyngeal flora. The leading symptom of ABPA is bronchial obstruction. Hence, the mode of drainage of sputum may be shaped by the obstruction of the conducting airways, which may impair comparability of sample pairs collected prior and during *A. fumigatus* detection. Authors should provide the patient's spirometry data at the days of collection.

Response:

*We appreciate the reviewer's view on expectorated sputa and potential contamination. We did not aim for identifying ABPA, however, but only needed detection of *A. fumigatus* colonization and subsequent differentiation of fungus positive and negative samples for our downstream strain-GEM analysis. For this we considered the expectorated sputum as a gold standard for microbiological diagnostic as reported also elsewhere (cf. Jones et al., 2021 or Xiao et al., 2018). Following the reviewer's suggestion, we added Supplementary Table S7 (updated order) containing clinical data of the CF patients including age, BMI, sex as well as PFTFEV1 (forced expiratory volume-one second in pulmonary function testing) and PFTFEV1pred (percent predicted PFTFEV1). Spirometry data is shown as "FEV1pred" (if available). We noted this addition in our methods and also added a piece of caution mentioning the oropharyngeal flora contamination in the results:*

Methods: "Supplementary Table S7 has clinical data on the cystic fibrosis patients including age, body mass index, sex, forced expiratory volume-one second in pulmonary function testing (PFTFEV1) and percent predicted PFTFEV1 (PFTFEV1pred)."

Results: "Although spontaneous expectoration of sputum is frequently used for sample acquisition^{29,30}, this practice may introduce contamination from oropharyngeal flora."

6- How did you define *A. fumigatus* colonization (line 572)?

Response:

*Colonization was defined by positive culture from the specimen of our sputum samples. Sputum samples were prepared for microscopy testing using Lactophenol Anilin Blue solution to detect fungi. In parallel, samples were plated on different agar: Columbia-Agar (with 5% sheep blood) (BD Diagnostic, Heidelberg, Germany), Chocolate-Agar (BioMérieux, Nürtingen, Germany), McConkey- Agar (BioMérieux, Nürtingen, Germany), Burkholderia cepacia-Spezial-Agar (7 days, 36°C) (BD Diagnostic, Heidelberg, Germany), and Sabouraud- Agar (7 days, 36°C) (BD Diagnostic, Heidelberg, Germany). Additionally, two media were used for anaerobic isolation (36°C): Schaedler-Agar (BioMérieux, Nürtingen, Germany) and Kanamycin-Vancomycin-Agar (BD Diagnostic, Heidelberg, Germany). Since all samples included in this study showed the growth of *A. fumigatus* on at least one of those agar plates, further testing by e.g. ITS sequencing was not applied. We added this piece of information to our methods:*

“Sputum samples were prepared for microscopy using lactophenol anilin blue solution to detect fungi. In parallel, samples were plated on: Columbia agar (with 5% sheep blood) (BD Diagnostic, Heidelberg, Germany), chocolate agar (BioMérieux, Nürtingen, Germany), McConkey agar (BioMérieux, Nürtingen, Germany), Burkholderia cepacia special agar (7 days, 36°C) (BD Diagnostic, Heidelberg, Germany), and Sabouraud agar (7 days, 36°C) (BD Diagnostic, Heidelberg, Germany). Two other media were used for anaerobic isolation (36°C): Schaedler agar (BioMérieux, Nürtingen, Germany) and kanamycin-vancomycin agar (BD Diagnostic, Heidelberg, Germany). Colonization was defined as positive culture from a specimen from the sputum samples on at least one of the agar plates.”

7- Sensitivity, specificity, validity and reliability of airway metagenome data critically depend on sample collection and processing (see e.g. Gut Pathogen 2016;8:24; BMC Biol 2014;12:87; Mol Ecol Resourc 2019;19:982-96). Please describe the

- mode of sampling,
- the latent period between sampling and freezing,
- the standard cleaning procedure of the laboratory environment,
- number and handling of negative controls during DNA extraction and library preparation.

Response:

Samples were obtained by spontaneous expectoration during routine visit in the CF center and were frozen within 24h after reception at the microbiology department. Regarding the cleaning procedure and laboratory environment, we delivered our samples to Novogene (<https://en.novogene.com/>) for metagenomic sequencing, The establishment of the quality management system of the Novogene laboratory is in accordance with the CAP laboratory certification, the international standard ISO 15189:2012 "Guidelines for the Quality and Capability of Medical Laboratory" and the international standard ISO 15190:2003 "Guidelines for the Safety Approval of Medical Laboratories". These ensured a clean laboratory environment. The whole process from DNA extraction, quality check, library construction to library pooling does not require intervention by the operators and is automated and intelligent, thus ensured a clean sequencing procedure.

DNA extraction and library preparation were performed by Novogene UK. During library preparation, we had a negative control (ethidium bromide solution) for quality check for every 11 samples, and it followed exactly the same handling procedure as our experimental samples. For the automated DNA extraction, there was no negative control. We modified our methods description accordingly:

Cystic Fibrosis sample acquisition: "Samples were frozen within 24 hr after reception at the microbiology department."

Library preparation and DNA sequencing: "For quality checking, 1 ethidium bromide negative control was added for every 11 samples and treated with the same handling procedure as the experimental samples."

8- Lines 615-617. Why did you use Kraken2? KrakenUniq is a more accurate metagenome binning tool surpassing Kraken and Kraken2. Back up your k-mer approach by an index classifier such as Centrifuge or MetaPhlan3.

Response:

We thank the reviewer for this suggestion. We agree that Kraken 2 has a slight false-positive rate that KrakenUniq does not have. However, as documented by the official guidelines (<http://ccb.jhu.edu/software/choosing-a-metagenomics-classifier>), KrakenUniq and Kraken 2 are uniquely useful depending on the project goal. For diagnosis of infections, where the goal is to identify a very small number of reads, KrakenUniq is superior to Kraken 2. This, however, was not the goal of our study. Given that Kraken 2 has also been successfully applied in many large microbiome studies (Almeida et al., 2021, Stacy et al., 2021, McCulloch et al., 2022) and in benchmarking studies for metagenomic profiling tools (with good performance) (Sun et al., 2021, Ye et al., 2019), we opted to keep our Kraken 2-derived results (including abundance and prevalence filtering).

Following the reviewer's valuable suggestion, we also assessed the reliability of and backed up our Kraken 2 results by applying KrakenUniq and Centrifuge, a non-Kraken index classifier (Table R3.5a and R3.5b below, updated Supplementary Table S6 [formerly S5]). The top 10 identified most abundant species were exactly the same for Kraken 2 and KrakenUniq, while for Centrifuge, 9 out of top 10 abundant species were the same (shown in Table R3.5a below). Furthermore, the top 10 abundant genera from Kraken 2 were also found by Centrifuge, while for KrakenUniq 9 out of 10 were the same (shown in Table R3.5b below). For all 5688 species that Kraken 2 detected (without any filtering), 4822 species (85%) can be detected by KrakenUniq (total 4950 species), while 4328 species (76%) can be detected by Centrifuge (total 4684 species). For the species that we listed in Supplementary Table S6 (after filtering), 574 out of 598 species (96%) that Kraken 2 detected could also be detected by KrakenUniq, while Centrifuge also detected 445 (74%). We adapted our manuscript and added these additional results accordingly:

*Results: "Despite differences among the patient cohort, starting biomaterial, and sequencing method, the taxonomic annotation of the 10 most abundant genera (Fig. 4a, Supplementary Table S6) showed striking similarities to two recent studies. In those papers, the lung microbiome of *A. fumigatus*-infected and control patients was investigated using 16S rRNA sequencing of either sputum samples or bronchoalveolar lavage^{31,32}. The genera were also found using Centrifuge as an alternative taxonomic profiling algorithm, while KrakenUniq also identified 9 of the 10 genera (Methods)."*

Methods: "KrakenUniq (v0.5.7, default parameters) and Centrifuge (v1.0.4, default parameters) were used to assess reliability and for comparison of Kraken 2 results."

Table R3.5a: Top 10 detected species by different classifiers (by relative median abundance).

Kraken 2	KrakenUniq	Centrifuge
Haemophilus parainfluenzae	Streptococcus mitis	Streptococcus mitis
Streptococcus mitis	Haemophilus parainfluenzae	Haemophilus parainfluenzae
Rothia mucilaginosa	Rothia mucilaginosa	Prevotella melaninogenica
Prevotella melaninogenica	Prevotella melaninogenica	Rothia mucilaginosa
Staphylococcus aureus	Staphylococcus aureus	Staphylococcus aureus
Neisseria mucosa	Veillonella atypica	Veillonella atypica
Streptococcus pneumoniae	Neisseria mucosa	Streptococcus pneumoniae
Streptococcus oralis	Streptococcus oralis	Streptococcus oralis
Veillonella atypica	Streptococcus pneumoniae	Veillonella dispar
Veillonella parvula	Veillonella parvula	Veillonella parvula

Table R3.5b: Top 10 detected genera by different classifiers (by relative median abundance).

Kraken 2	KrakenUniq	Centrifuge
Streptococcus	Streptococcus	Streptococcus
Prevotella	Prevotella	Haemophilus
Haemophilus	Haemophilus	Prevotella
Veillonella	Veillonella	Veillonella
Rothia	Rothia	Rothia
Neisseria	Neisseria	Neisseria
Gemella	Gemella	Gemella
Staphylococcus	Staphylococcus	Staphylococcus
Pseudomonas	Actinomyces	Pseudomonas
Actinomyces	Capnocytophaga	Actinomyces

9- Lines 617-618. Low abundance species are meanwhile known to be crucial for the metabolic competence of an airway metagenome. Please process your datasets with a tool that removes false negative and false positive assignments.

Response:

Since airway microbiome is a relatively low-biomass environment compared to the gut microbiota, we preferred to use a sensitive approach as given by Kraken 2. As a result, this will inevitably introduce false positive species, as pointed out by the reviewer as well as by a previous study (LaPierre et al., 2020), which found that Kraken 2 can produce false positive species that are low-abundant. Without an experimental negative control during the sampling step (that we acknowledge as a limitation, see below), it is difficult for us to flag the true false positive species. As mentioned above, even the use of KrakenUniq that was claimed to be well suited to help filter false positives still detected 4950 species in total. Thus, to counteract and compensate this shortcoming we decided to remove low-abundance species to reduce the effect of false positive. Of course, we cannot rule out that this approach will possibly remove some low-abundance species that are actually present. In fact, the MAMBO algorithm we used for predicting the lung microbiome most supporting metabolite profiles relies on Pearson's correlation and thus weighs nutritional needs of highly abundant species higher than those of low abundance species. Since our downstream strain-GEM simulations relied on the metabolite profiles rather than the detected lung microbiome profile, this also increased the credibility of our analysis. Although the actual lowly abundant species are of less important to us than potentially to other studies, removal of low

abundant species involving applying Kraken 2 has also been used before (Peabody et al., 2015, LaPierre et al., 2020). We extended our results and methods to address these points:

Results: “Our study lacked true negative controls, so to minimize the chances of detecting false-positives species, we applied abundance and prevalence filters and compared detected genera and species using two alternative tools to ensure robustness in lung microbe detection (Methods).”

Methods: “KrakenUniq (v0.5.7, default parameters) and Centrifuge (v1.0.4, default parameters) were used to assess reliability and for comparison of Kraken 2 results.”

10- Lines 615-618. Unfortunately, reference eukaryotic genomes are often contaminated, making it hard to detect and quantify eukaryotic microbes in shotgun metagenomic data. You may use a tool like EukDetect, which is specifically built to deal with these problems. In the context of *Aspergillus*, one need to know whether patients’ samples contained other fungi like *Scedosporium* that are suspected to be clinically more relevant for the ABPA phenotype than *A. fumigatus*.

Response:

*We agree that reference eukaryotic genomes are frequently contaminated. In this manuscript, shotgun metagenomic data were used only to detect and quantify prokaryotic microbes, whereas colonization of *A. fumigatus* was confirmed by cell culture. We also followed the reviewer’s recommendation and applied EukDetect to detect other eukaryotic microbes in our samples, however, in most samples no fungi were detected by EukDetect including *Scedosporium*. We added the additional test by EukDetect to our methods:*

*“We applied EukDetect (v1.3)⁷⁶ to test and confirm that patient samples contained no other fungi such as *Scedosporium* that may be clinically relevant to *A. fumigatus*-associated disease phenotypes such as allergic bronchopulmonary aspergillosis.”*

11- Line 631. How did you define infection with *A. fumigatus*?

Response:

As pointed out in the response to point 6 by this Reviewer we defined infection by a positive culture from the specimen of our sputum samples in at least one of the tested agar plates and clarified this also in the manuscript.

12- Line 629. The rationale for the thresholds of prevalence and abundance filters should be given. If the dataset is appropriately curated, less prevalent and less abundant taxa can be included.

Response:

The reviewer has expressed also above his/her concern of including false positives during our taxonomic annotation. Balancing not missing true positives and including false positives when using a sensitive method like Kraken 2 for taxonomic profiling requires some compromises. According to previous research studies not using prevalence and abundance filters (LaPierre et al., 2020) can produce many false positive species that are low-abundant. In order to reduce the effect of false positive species, we decided to use a prevalence and abundance filter. Our choice

of abundance thresholds was similar to other papers (Beghini et al., 2021, Erawijantari et al., 2020) and our choice for the threshold of prevalence filter is also commonly used. We added this notion to our methods:

“Low-abundance species were removed at cutoff = 0.1% (Supplementary Table S6) as in similar studies^{74,75}.”

13- Figure 3A. Burkholderia spp. and particularly Sphingomonas spp. are rare members of the CF airway community. Demonstrate that Af positive patients at the Heidelberg CF clinic are more frequently harboring these proteobacteria than patients seen at any other CF clinic in the world.

Response:

We thank the reviewer for pointing out this potential issue and agree on Burkholderia spp. and Sphingomonas spp. being rare members of the CF airway microbiota community. According to a previous study, the CF airway microbiota community is often dominated by one or a few principle colonizers/pathogens (Feigelman et al., 2017). Similarly, in our cohort, Burkholderia spp. and Sphingomonas spp. have low prevalence (22 out of 80 samples had a relative abundance above 0.1% for Burkholderia, while Sphingomonas was above 0.1% relative abundance in only 15 samples) but were highly abundant in only few patients, which biased the calculation of mean relative abundance of all samples. To avoid this issue and potential skews in our data visualizations, we have switched to a more robust median relative abundance to obtain top-abundant taxa and create a new Fig. 4a (shown below).

Updated Figure 4a (3a in the original submission).

14- Figure 3B shows numerous taxa that have not been detected in previous 16S rRNA or metagenome studies of CF airway specimens. Verify these species by another pipeline (Centrifuge, MetaPhlan3) and by a tool that flags false positive and false negative taxonomic assignments. In other words, the datasets need to be curated. The reviewer assumes that the figure shows about 17, 14, 7 and 9 false positive assignments in modules 1, 2, 3, 4, respectively.

Response:

As in several of the comments above the reviewer refers to the accuracy of the taxonomic annotation and we believe we have already addressed it in comment 8 by including more taxonomic annotation tools. Nevertheless, as recently reviewed (Thornton et al., 2022), most of previous CF airway studies were based on 16S, which has lower taxonomy resolution than shotgun metagenomics. For the limited metagenomes studies, the sample size was also smaller than ours, which is another possibility why we see new taxa in our analysis. Having said that we also followed the reviewer suggestion to increase the trust in our approach. We verified our identified species by KrakenUniq and a representative index classifier (Centrifuge), where 95 (66) species out of 96 species in the network can be detected as well, respectively. We are not aware of a tool that can flag accurately true false positive/negatives nor how the reviewer proposed false positives by simply looking into the modules. However, to provide confidence assessments of the taxonomic assignments, we adopted a voting system when visualizing the species in the network, where species annotated by different numbers of tools were sized differently (as shown in the new Fig. 4b below, where larger size represents several supporting tools). We included this additional piece of information into an updated Supplementary Table S6 (formerly S5), now detailing the species detection by different tools.

Of note, unlike the research into gut microbiota, admittedly lung microbiome is still in its infancy (hinted by 3278 available papers listed at pubmed searching for “metagenomics” together with “gut” compared to 295 papers using the search terms “metagenomics” and “lung”). Future studies with more appropriate designs, quality controls and higher sequencing depths with larger number of samples are definitely needed to widen our knowledge on the exact nature of lung microbe communities and to establish the lung microbiome gene catalog. We improved Figure 4b, highlighting now species detected by multiple tools and reflected this with the following figure 4b caption addition:

“Larger node size indicates species detected by multiple tools for taxonomic profiling.”

Updated Figure 4b (formerly 3b).

15- Figure 4. Authors should discuss why the first two dimensions in the PCA just explains 13-14% of the variance.

Response:

Following the suggestion from the reviewer, we have added the text below to our manuscript. The explained variance in our study is not anything unique and there are other studies of microbiome and metabolites that showed similar levels of explained variance (e.g. Wang, et al., 2019, Fig. 1b). We added a note in our results towards this point:

Results: "Of note, more than 82% of the variance could be explained by the first 50 dimensions of the principal component analysis. The first two components explained fewer, albeit significant, changes in metabolites between patient samples before and after A. fumigatus infection despite the high complexity of these data."

16- Table S1. ATP demand. Provide error estimates and restrict numbers to physically plausible digits.

Response:

We rounded the respective numbers to two decimal places except the values for DNA and RNA in column C as they were an order of magnitude lower compared to the other reported values.

17- Table S2. Quantify the term 'optimal' for growth prediction and explain your terminology in more depth in the Read me section.

Response:

We apologize for the unclear explanation of the term 'optimal'. The term is related to the type of mathematical solution in constraint-based metabolic modeling (CBM) and flux balance analysis (FBA, Orth et al. 2010). To "solve" a genome-scale metabolic model, a target objective function (commonly biomass) needs to be defined. This is usually solved by FBA, meaning that it is maximized (less common: minimized). We extended our explanation accordingly and revised the text in the related Supplementary Table S2:

"optimal" indicates that the maximization of growth rate by flux balance analysis was feasible given the set constraints, i.e. the allowed influx of metabolites."

18- Table S3. Metadata of the A. fumigatus strains should be given (date of collection, habitat, geographic origin, collector (name or institution)). I did not find this standard information in the authors' ref. 6 (Barber et al., 2021) cited throughout the table.

Response:

Following the reviewer's suggestion, we added the requested information in Supplementary Table S3, sheet "afu_isolate_metadata", in separate columns. This includes "Isolate", "Sample Name", "Source", "NCBI BioSample Accession", "Geographic origin", "Habitat", "Year of collection", and "Collector".

19- Table S4. Description. Provide reference for reaction ID, spell out abbreviation (FVA). Provide error estimates for the ratio mean(clinical)/mean(environmental). Five or six digits do not make sense.

Response:

To avoid overly complex information per supplementary item we provided references per reaction only in Supplementary Table S9 (formerly S7). We rounded the numbers to two decimal places. Also, we spelled out 'flux variability analysis' in the Description sheet of Supplementary Table S5 (formerly S4). We also added a note that references of reactions IDs can be found in Supplementary Table S9. We changed the description in Supplementary Table S5 accordingly:

"The input for the model were flux variability analysis (FVA) derived flux bounds for each reaction that support the objective function (biomass).

Details on reaction IDs including full equation, database and literature references are given in Supplementary Table S9."

20- Table S5 lists the presence of 600 taxa the majority of which never seen before in any CF patient's airways. The dataset must be rigorously curated and checked for false positive assignments and contaminations from other sources (lab, environment, other samples).

Response:

We have discussed parts of this request in several of the previous comments. We already added information about whether the species can be detected by other tools (KrakenUniq, Centrifuge) in Supplementary Table S6 (formerly S5). As we have mentioned in our paper (Line 272-275), the top 10 most abundant genera showed striking similarities to two recent 16s rRNA studies. Besides, 574 out of 598 species (96%) that Kraken 2 detected can also be detected by KrakenUniq, while Centrifuge also detected 445 species. To further interrogate our results, we obtained taxa results from three independent airway metagenomics dataset using the same tools and parameters that we used in this paper (Pust et al., 2020, Feigelman et al., 2017, Dmitrijeva et al., 2021). Clustering of samples by PCoA based on the beta diversity (Bray-Curtis distance) of our samples and other datasets showed that our dataset shared similar microbial structure and composition with other airway metagenomes (Fig. R3.20a below). It also turned out that the percentage of 598 species that we detected in other datasets is very high (Pust dataset: 94.48%, Feigelman dataset: 93.14%, Dmitrijeva dataset: 94.65%).

Figure R3.20a: The beta diversity (bray-curtis distance) of our samples and other lung metagenome datasets.

To further check for possible contaminations as requested by the reviewer, we used FEAST (Shenhav, et al., 2019). FEAST partitions microbial samples into their source components that can be used to quantify contamination or other potential source environments. We combined the three other CF airway datasets mentioned above and a clinical environmental dataset (Brooks, et al., 2017), as the potential source environments. The FEAST results (Fig. R3.20b below) indicated a dominant source of our samples to be CF patients' lung microbiome and a significantly higher contribution of them than the environment.

Figure R3.20b (new Supplementary Fig. S3): The FEAST result for estimating the contribution of potential source environments to the target dataset (i.e. our dataset) by combining three other lung metagenome datasets and a clinical environmental dataset (see response for description). The p value was obtained by Wilcoxon signed rank test.

To provide confidence assessments of the taxonomic assignments, we updated Supplementary Table S6 (formerly S5) with information on detection by different tools in sheet 'Metagenomic species' and added a new sheet 'Detection by other studies' showing the details of detection by other studies.

We added the FEAST results as new Supplementary Figure S3, added its description to our methods section and noted the lack of negative controls in our results:

Methods: "FEAST (v0.1.0, default parameters) was used to check for possible contamination and to estimate the contribution of potential source environments using the cystic fibrosis lung microbiome dataset from three published studies⁷⁰⁻⁷² and a clinical environmental dataset⁷³. Taxonomic assignment of the datasets followed the same pipeline used for our dataset. FEAST results (Supplementary Fig. S3) indicated a dominant source of our samples was the lung microbiome of cystic fibrosis patients with a significantly higher contribution from them than from the clinical environment."

Results: "Our study lacked true negative controls, so to minimize the chances of detecting false-positives species, we applied abundance and prevalence filters and compared detected genera and species using two alternative tools to ensure robustness in lung microbe detection (Methods)."

21- Table S6. Were P values corrected for multiple testing?

Response: *Yes, the p values were corrected by FDR. We clarified this in the description sheet of Supplementary Table S8 (formerly S6):*

"P values and adjusted p values for multiple test correction by FDR are indicated."

References

Agren R, Mardinoglu A, Asplund A, Kampf C, Uhlen M, Nielsen J. Identification of anticancer drugs for hepatocellular carcinoma through personalized genome-scale metabolic modeling. *Mol Syst Biol*. 2014 Mar 19;10(3):721. doi: 10.1002/msb.145122.

Almeida A, Nayfach S, Boland M, Strozzi F, Beracochea M, Shi ZJ, Pollard KS, Sakharova E, Parks DH, Hugenholtz P, Segata N, Kyrpides NC, Finn RD. A unified catalog of 204,938 reference genomes from the human gut microbiome. *Nat Biotechnol*. 2021 Jan;39(1):105-114. doi: 10.1038/s41587-020-0603-3.

Barber AE, Sae-Ong T, Kang K, Seelbinder B, Li J, Walther G, Panagiotou G, Kurzai O. *Aspergillus fumigatus* pan-genome analysis identifies genetic variants associated with human infection. *Nat Microbiol*. 2021 Dec;6(12):1526-1536. doi: 10.1038/s41564-021-00993-x.

Barker BM, Kroll K, Vödisch M, Mazurie A, Kniemeyer O, Cramer RA. Transcriptomic and proteomic analyses of the *Aspergillus fumigatus* hypoxia response using an oxygen-controlled fermenter. *BMC Genomics*. 2012 Feb 6;13:62. doi: 10.1186/1471-2164-13-62.

Beghini F, McIver LJ, Blanco-Míguez A, Dubois L, Asnicar F, Maharjan S, Mailyan A, Manghi P, Scholz M, Thomas AM, Valles-Colomer M, Weingart G, Zhang Y, Zolfo M, Huttenhower C, Franzosa EA, Segata N. Integrating taxonomic, functional, and strain-level profiling of diverse microbial communities with bioBakery 3. *Elife*. 2021 May 4;10:e65088. doi: 10.7554/eLife.65088.

Björnson E, Mukhopadhyay B, Asplund A, Pristovsek N, Cinar R, Romeo S, Uhlen M, Kunos G, Nielsen J, Mardinoglu A. Stratification of Hepatocellular Carcinoma Patients Based on Acetate Utilization. *Cell Rep*. 2015 Dec 1;13(9):2014-26. doi: 10.1016/j.celrep.2015.10.045.

Brooks B, Olm MR, Firek BA, Baker R, Thomas BC, Morowitz MJ, Banfield JF. Strain-resolved analysis of hospital rooms and infants reveals overlap between the human and room microbiome. *Nat Commun*. 2017 Nov 27;8(1):1814. doi: 10.1038/s41467-017-02018-w.

Choera T, Zelante T, Romani L, Keller NP. A Multifaceted Role of Tryptophan Metabolism and Indoleamine 2,3-Dioxygenase Activity in *Aspergillus fumigatus*-Host Interactions. *Front Immunol*. 2018 Jan 22;8:1996. doi: 10.3389/fimmu.2017.01996.

Dmitrijeva M, Kahlert CR, Feigelman R, Kleiner RL, Nolte O, Albrich WC, Baty F, von Mering C. Strain-Resolved Dynamics of the Lung Microbiome in Patients with Cystic Fibrosis. *mBio*. 2021 Mar 9;12(2):e02863-20. doi: 10.1128/mBio.02863-20.

Erawijantari PP, Mizutani S, Shiroma H, Shiba S, Nakajima T, Sakamoto T, Saito Y, Fukuda S, Yachida S, Yamada T. Influence of gastrectomy for gastric cancer treatment on faecal microbiome and metabolome profiles. *Gut*. 2020 Aug;69(8):1404-1415. doi: 10.1136/gutjnl-2019-319188.

Feigelman R, Kahlert CR, Baty F, Rassouli F, Kleiner RL, Kohler P, Brutsche MH, von Mering C. Sputum DNA sequencing in cystic fibrosis: non-invasive access to the lung microbiome and to pathogen details. *Microbiome*. 2017 Feb 10;5(1):20. doi: 10.1186/s40168-017-0234-1.

Glanemann C, Loos A, Gorret N, Willis LB, O'Brien XM, Lessard PA, Sinskey AJ. Disparity between changes in mRNA abundance and enzyme activity in *Corynebacterium glutamicum*: implications for DNA microarray analysis. *Appl Microbiol Biotechnol*. 2003 Mar;61(1):61-8. doi: 10.1007/s00253-002-1191-5.

Jones JT, Liu KW, Wang X, Kowalski CH, Ross BS, Mills KAM, Kerkaert JD, Hohl TM, Lofgren LA, Stajich JE, Obar JJ, Cramer RA. *Aspergillus fumigatus* Strain-Specific Conidia Lung Persistence Causes an Allergic Broncho-Pulmonary Aspergillosis-Like Disease Phenotype. *mSphere*. 2021 Feb 17;6(1):e01250-20. doi: 10.1128/mSphere.01250-20.

Lambou K, Lamarre C, Beau R, Dufour N, Latge JP. Functional analysis of the superoxide dismutase family in *Aspergillus fumigatus*. *Mol Microbiol*. 2010 Feb;75(4):910-23. doi: 10.1111/j.1365-2958.2009.07024.x.

LaPierre N, Alser M, Eskin E, Koslicki D, Mangul S. Metalign: efficient alignment-based metagenomic profiling via containment min hash. *Genome Biol*. 2020 Sep 10;21(1):242. doi: 10.1186/s13059-020-02159-0.

Lieven C, Beber ME, Olivier BG, Bergmann FT, Ataman M, Babaei P, Bartell JA, Blank LM, Chauhan S, Correia K, Diener C, Dräger A, Ebert BE, Edirisinghe JN, Faria JP, Feist AM, Fengos G, Fleming RMT, García-Jiménez B, Hatzimanikatis V, van Helvoirt W, Henry CS, Hermjakob H, Herrgård MJ, Kaafarani A, Kim HU, King Z, Klamt S, Klipp E, Koehorst JJ, König M, Lakshmanan M, Lee DY, Lee SY, Lee S, Lewis NE, Liu F, Ma H, Machado D, Mahadevan R, Maia P, Mardinoglu A, Medlock GL, Monk JM, Nielsen J, Nielsen LK, Nogales J, Nookaew I, Palsson BO, Papin JA, Patil KR, Poolman M, Price ND, Resendis-Antonio O, Richelle A, Rocha I, Sánchez BJ, Schaap PJ, Malik Sheriff RS, Shoaie S, Sonnenschein N, Teusink B, Vilaça P, Vik JO, Wodke JAH, Xavier JC, Yuan Q, Zakhartsev M, Zhang C. MEMOTE for standardized genome-scale metabolic model testing. *Nat Biotechnol*. 2020 Mar;38(3):272-276. doi: 10.1038/s41587-020-0446-y. Erratum in: *Nat Biotechnol*. 2020 Apr;38(4):504.

Lu H, Li F, Sánchez BJ, Zhu Z, Li G, Domenzain I, Marcišauskas S, Anton PM, Lappa D, Lieven C, Beber ME, Sonnenschein N, Kerkhoven EJ, Nielsen J. A consensus *S. cerevisiae* metabolic model Yeast8 and its ecosystem for comprehensively probing cellular metabolism. *Nat Commun*. 2019 Aug 8;10(1):3586. doi: 10.1038/s41467-019-11581-3. Erratum in: *Nat Commun*. 2020 Oct 22;11(1):5443.

Machado D, Herrgård M. Systematic evaluation of methods for integration of transcriptomic data into constraint-based models of metabolism. *PLoS Comput Biol*. 2014 Apr 24;10(4):e1003580. doi: 10.1371/journal.pcbi.1003580. Erratum in: *PLoS Comput Biol*. 2014 Oct;10(10):e1003989.

McCulloch JA, Davar D, Rodrigues RR, Badger JH, Fang JR, Cole AM, Balaji AK, Vetizou M, Prescott SM, Fernandes MR, Costa RGF, Yuan W, Salcedo R, Bahadiroglu E, Roy S, DeBlasio RN, Morrison RM, Chauvin JM, Ding Q, Zidi B, Lowin A, Chakka S, Gao W, Pagliano O, Ernst SJ, Rose A, Newman NK, Morgun A, Zarour HM, Trinchieri G, Dzutsev AK. Intestinal microbiota signatures of clinical response and immune-related adverse events in melanoma patients treated with anti-PD-1. *Nat Med*. 2022 Mar;28(3):545-556. doi: 10.1038/s41591-022-01698-2.

Opdam S, Richelle A, Kellman B, Li S, Zielinski DC, Lewis NE. A Systematic Evaluation of Methods for Tailoring Genome-Scale Metabolic Models. *Cell Syst*. 2017 Mar 22;4(3):318-329.e6. doi: 10.1016/j.cels.2017.01.010.

Orth JD, Thiele I, Palsson BØ. What is flux balance analysis? *Nat Biotechnol*. 2010 Mar;28(3):245-8. doi: 10.1038/nbt.1614.

Peabody MA, Van Rossum T, Lo R, Brinkman FS. Evaluation of shotgun metagenomics sequence classification methods using in silico and in vitro simulated communities. *BMC Bioinformatics*. 2015 Nov 4;16:363. doi: 10.1186/s12859-015-0788-5.

Pust MM, Wiehlmann L, Davenport C, Rudolf I, Dittrich AM, Tümmler B. The human respiratory tract microbial community structures in healthy and cystic fibrosis infants. *NPJ Biofilms Microbiomes*. 2020 Dec 15;6(1):61. doi: 10.1038/s41522-020-00171-7.

Shenhav L, Thompson M, Joseph TA, Briscoe L, Furman O, Bogumil D, Mizrahi I, Pe'er I, Halperin E. FEAST: fast expectation-maximization for microbial source tracking. *Nat Methods*. 2019 Jul;16(7):627-632. doi: 10.1038/s41592-019-0431-x.

Slesiona S, Gressler M, Mihlan M, Zaehle C, Schaller M, Barz D, Hube B, Jacobsen ID, Brock M. Persistence versus escape: *Aspergillus terreus* and *Aspergillus fumigatus* employ different strategies during interactions with macrophages. *PLoS One*. 2012;7(2):e31223. doi: 10.1371/journal.pone.0031223.

Slesiona S, Ibrahim-Granet O, Olias P, Brock M, Jacobsen ID. Murine infection models for *Aspergillus terreus* pulmonary aspergillosis reveal long-term persistence of conidia and liver degeneration. *J Infect Dis*. 2012 Apr 15;205(8):1268-77. doi: 10.1093/infdis/jis193.

Stacy A, Andrade-Oliveira V, McCulloch JA, Hild B, Oh JH, Perez-Chaparro PJ, Sim CK, Lim AI, Link VM, Enamorado M, Trinchieri G, Segre JA, Rehermann B, Belkaid Y. Infection trains the host for microbiota-enhanced resistance to pathogens. *Cell*. 2021 Feb 4;184(3):615-627.e17. doi: 10.1016/j.cell.2020.12.011.

Sun Z, Huang S, Zhang M, Zhu Q, Haiminen N, Carrieri AP, Vázquez-Baeza Y, Parida L, Kim HC, Knight R, Liu YY. Challenges in benchmarking metagenomic profilers. *Nat Methods*. 2021 Jun;18(6):618-626. doi: 10.1038/s41592-021-01141-3.

Thiele I, Palsson BØ. A protocol for generating a high-quality genome-scale metabolic reconstruction. *Nat Protoc*. 2010 Jan;5(1):93-121. doi: 10.1038/nprot.2009.203.

Thornton CS, Acosta N, Surette MG, Parkins MD. Exploring the Cystic Fibrosis Lung Microbiome: Making the Most of a Sticky Situation. *J Pediatric Infect Dis Soc*. 2022 Sep 7;11(Supplement_2):S13-S22. doi: 10.1093/jpids/piac036.

Wang M, Wan J, Rong H, He F, Wang H, Zhou J, Cai C, Wang Y, Xu R, Yin Z, Zhou W. Alterations in Gut Glutamate Metabolism Associated with Changes in Gut Microbiota Composition in Children with Autism Spectrum Disorder. *mSystems*. 2019 Jan 29;4(1):e00321-18. doi: 10.1128/mSystems.00321-18.

Wu HQ, Cheng ML, Lai JM, Wu HH, Chen MC, Liu WH, Wu WH, Chang PM, Huang CF, Tsou AP, Shiao MS, Wang FS. Flux balance analysis predicts Warburg-like effects of mouse hepatocyte deficient in miR-122a. *PLoS Comput Biol*. 2017 Jul 7;13(7):e1005618. doi: 10.1371/journal.pcbi.1005618.

Xiao W, Gong DY, Mao B, Du XM, Cai LL, Wang MY, Fu JJ. Sputum signatures for invasive pulmonary aspergillosis in patients with underlying respiratory diseases (SPARED): study protocol for a prospective diagnostic trial. *BMC Infect Dis*. 2018 Jun 11;18(1):271. doi: 10.1186/s12879-018-3180-z.

Yang JE, Park SJ, Kim WJ, Kim HJ, Kim BJ, Lee H, Shin J, Lee SY. One-step fermentative production of aromatic polyesters from glucose by metabolically engineered *Escherichia coli* strains. *Nat Commun*. 2018 Jan 8;9(1):79. doi: 10.1038/s41467-017-02498-w.

Ye SH, Siddle KJ, Park DJ, Sabeti PC. Benchmarking Metagenomics Tools for Taxonomic Classification. *Cell*. 2019 Aug 8;178(4):779-794. doi: 10.1016/j.cell.2019.07.010.

Zelante T, Choera T, Beauvais A, Fallarino F, Paolicelli G, Pieraccini G, Pieroni M, Galosi C, Beato C, De Luca A, Boscaro F, Romoli R, Liu X, Warris A, Verweij PE, Ballard E, Borghi M, Pariano M, Costantino G, Calvitti M, Vacca C, Oikonomou V, Gargaro M, Wong AYW, Boon L, den Hartog M, Spáčil Z, Puccetti P, Latgè JP, Keller NP, Romani L. *Aspergillus fumigatus* tryptophan metabolic route differently affects host immunity. *Cell Rep*. 2021 Jan 26;34(4):108673. doi: 10.1016/j.celrep.2020.108673.

REVIEWER COMMENTS

Reviewer #1 (Remarks to the Author):

The authors have sufficiently improved their paper, in response to the comments made. I enjoyed reading it.

although there are some major concerns remained with the methodology itself. the main concern would be the number of reactions that must be added to a network to fill its gaps (previously 500 reactions - reduced to 200 reactions after the first revision). in this regard, there are several available gap-filling approaches that are capable of filling networks gaps by adding less than 50 reactions regardless of network size and the species, But all are template based and computationally expensive. current approach would be highly beneficial when it comes to reconstructing GEMs for non-model organisms lacking a previous template for gap-filling. The following comments may help to improve the methodology to some extent.

1- a downstream workflow to assess the necessity of reactions that are added to networks to fill its gaps. this could be made by running a simple reaction essentiality analysis (both for Biomass objective function and targeted fermentation metabolites) on a set of 200 reactions that are added to the network. this may infer some of the mentioned reactions are not essential either for growth or for enhancing prediction on fermentation profile and thus may result in a reduced number of gap reactions.

2- given that NADH and NADPH are the main redox cofactors of catabolic and anabolic pathways respectively, it might be helpful to consider a weighting score for reactions with different cofactors as further selection criteria, this may potentially help to reduce the number of added reactions by reducing the number of redundant reactions.

3- checking the directionality of CHESHIRE-200 is a simple and doable analysis and may help to make the number of gaps fall into a reasonable range (less than 100). this could be done by checking whether any two reactions within CHESHIRE-200 have the same stoichiometry but different directionality. any two reactions that meet these conditions could be merged into a bi-directional reaction.

Reviewer #2 (Remarks to the Author):

I thank the authors for providing a thorough rebuttal to all the reviewers comments. The paper is much improved, and I acknowledge that a lot of work has gone into this, both initially and in the review process. I believe enough has been done, and I think the editors will be happy.

Personally, I am still confused by the omnilog section of the paper. Whilst the authors have indeed provided more clarity on this, it actually leads to more questions, the main one being, why the choice of redox dye D? This is a dye specifically for yeast, not filamentous fungi. In which case, maybe the FF plates would have been better for this particular part of the study?

Reviewer #3 (Remarks to the Author):

NCOMMS-22-25651A

Mirhakkak et al

Genome-scale metabolic modeling of 252 *Aspergillus fumigatus* strains reveals growth dependencies on the lung microbiome

After having studied the point-to-point response, manuscript and supplement for more than a full-working day, I can say that I am impressed with the ambitions, scope and beauty of this work. For me, the strength resides in the Systems Biology part, i.e. the Pan-GEM reconstruction. However, unfortunately the experimental part still contains some major flaws that will backfire over time so that this work (as it is now) will not become a milestone of *Aspergillus* research.

Please take my critique seriously and do not look for excuses. If you cannot fix the errors, you may reorganize scope and contents of the manuscript and delete some parts.

Major

In my opinion, the most critical part is the CF airway metagenome study:

Sample collection. Lines 639 – 643: “Spontaneously expectorated sputum was collected during visits to the Cystic Fibrosis Center at the University Hospital Heidelberg and frozen in liquid nitrogen on the day of visit ... Samples were frozen within 24 hr after reception at the microbiology department.”

1. First, spontaneously expectorated sputum will always be contaminated by oral microbiota. This is not problematic for routine culture-dependent diagnostics of the typical CF pathogens, but it is inappropriate for any in-depth metagenome analysis that includes commensals. Induced sputum is the best compromise to minimize contamination (Weiser et al. The lung microbiota in children with cystic fibrosis captured by induced sputum sampling. *J Cyst Fibros.* 2022;21:1006-1012; Ronchetti K et al. The CF-Sputum Induction Trial (CF-SpIT) to assess lower airway bacterial sampling in young children with cystic fibrosis: a prospective internally controlled interventional trial. *Lancet Respir Med.* 2018;6:461-471).

2. Second, the composition of microbial communities in CF respiratory secretion shifts within minutes. Hence, samples must be frozen asap at -80°C or liquid nitrogen, i.e. within less than a minute.

Processing and evaluation of sequence data. Line 692-693: 'BWA (v07.17) was used to align quality filtered reads to the human reference genome (hg38) for removal of human-derived reads.'

3. BWA is an adequate aligner for any study on human genomic DNA, but it is inappropriate for metagenome studies, particularly if the reads are longer than 50 bp. The aligner will clip sequence reads and the generated short k-mers will give an explosion of false-positive assignments. (You may test this behaviour by the comparison of 50, 70, 100, 150, 200 bp long reads of the same metagenome dataset.) Recommendation: The primary data should be re-evaluated by another aligner.

4. Figure 4b. I counted 18 species that have never been detected by others in the CF airway microbiome. In other words, the taxonomic classification of your datasets needs a rigorous re-classification.

5. Table S6. Of the 596 species I counted more than 150 implausible taxa. In addition, I noted an inflation of taxa among the abundant genera, probably due to false-positive alignment of homologous reads. The latter argument particularly applies to the genera *Pseudomonas*, *Stenotrophomonas* and *Neisseria*. You may easily curate your dataset by eliminating all microbial species with non-uniform read distributions.

6. You write in your response to point 20: "To further interrogate our results, we obtained taxa results from three independent airway metagenomics dataset using the same tools and parameters

that we used in this paper (Pust et al., 2020, Feigelman et al., 2017, Dmitrijeva et al., 2021). Clustering of samples by PCoA based on the beta diversity (Bray-Curtis distance) of our samples and other datasets showed that our dataset shared similar microbial structure and composition with other airway metagenomes (Fig. R3.20a below). It also turned out that the percentage of 598 species that we detected in other datasets is very high (Pust dataset: 94.48%, Feigelman dataset: 93.14%, Dmitrijeva dataset: 94.65%).”

In case of the Pust dataset 94.48% of 598 species are 565 species. Pust’s original paper (Supplementary Table S2 published in *Comput. Struct. Biotechnol.* 2021, 20:176-186), however, lists just 257 species. Please contain the inflation of species generated by false-positive alignment by eliminating all microbial species with non-uniform read distributions.

7. I am struggling with the meaning of the ‘targeted metabolomics’ data. What do this data of amino acid content in the supernatant tell us about the cellular amino acid metabolism? Some amino acids were apparently not detected, whereas others showed high coefficients of variation. I missed the information about the number of biological and technical replicates. This part is probably dispensable.

Minor

8. Line 118. You mention that the first draft model included 3,233 exchange reactions, but Figure 1b (probably representing the final model) indicates only 239 exchange reactions. Please clarify.

9. Figure 3c. The decision tree deserves some more interpretation in the text.

10. Table S7. The inclusion of clinical metadata is appreciated. However, first, please explain why you can provide paired spirometry and BMI data for only 14 and 26 of the 40 study participants, respectively. The CF clinic Heidelberg receives high financial support for its patient registry; hence, complete data should be available. According to guidelines, spirometry is obligatory at each patient’s visit of the CF center. Second, and more importantly, I do not see a clinical signal of the detection of *A. fumigatus* in the patient’s respiratory secretion. FEV1 was better in 8 patients at the day of the Af-negative sample and better in 8 patients at the day of the Af-positive sample. Likewise, BMI was higher in 8 patients at the day of the Af-negative sample and higher in 18 patients at the day of the Af-positive sample. Apparently, the investigated dataset does not show any association between Af-status and clinical status. This observation is consistent with clinical experience that the detection of Af needs to be interpreted in the light of clinical symptoms and Af serology. Your text avoids any claims *expressis verbis* between presence/absence of Af and clinical status, but you should state more clearly in the text that Af modifies the lung microbiome, but no consequences for the host –

microbe interaction can yet be deduced from your data. Your title is sexy, but evokes expectations that cannot be fulfilled.

11. Supplementary Tables. For most tables, the number of the Table is not given in the description. Please rectify.

REVIEWER COMMENTS

Reviewer #1 (Remarks to the Author):

The authors have sufficiently improved their paper, in response to the comments made. I enjoyed reading it.

Response: *We thank the reviewer for this remark as indeed tremendous serious efforts went into improving the manuscript on all provided comments.*

although there are some major concerns remained with the methodology itself. the main concern would be the number of reactions that must be added to a network to fill its gaps (previously 500 reactions - reduced to 200 reactions after the first revision). in this regard, there are several available gap-filling approaches that are capable of filling networks gaps by adding less than 50 reactions regardless of network size and the species, But all are template based and computationally expensive. current approach would be highly beneficial when it comes to reconstructing GEMs for non-model organisms lacking a previous template for gap-filling. The following comments may help to improve the methodology to some extent.

Response: *We appreciate the thoughts of the reviewer towards further improvement of our GEMs and the comments made below. To respond to all points made, we aimed to minimize the presence of non-annotated reactions for gap-filling purposes in our pan-GEM. All additional analyses described below were based on 688 identified reactions that were included during refinement steps of our pan-GEM. These included 609 transport reactions between compartments and 79 non-genome associated reactions occurring in only one compartment in our pan-GEM.*

1- a downstream workflow to assess the necessity of reactions that are added to networks to fill its gaps. this could be made by running a simple reaction essentiality analysis (both for Biomass objective function and targeted fermentation metabolites) on a set of 200 reactions that are added to the network. this may infer some of the mentioned reactions are not essential either for growth or for enhancing prediction on fermentation profile and thus may result in a reduced number of gap reactions.

Response: *To accomplish essentiality analysis on the set of available gap-filling reactions, we performed minimal cut set analysis (MCS). MCS allowed us to not only detect single essential reactions, but to identify minimal sets of reactions that must occur simultaneously. By simulating unconstrained exchange reaction fluxes we identified 210 blocked reactions, which are not present in any strain GEM. We deleted these from our pan-GEM accordingly. Of note this step did not alter any downstream analysis with either pan-GEM or the strain GEMs. In addition, we identified 314 reactions involved in at least one MCS. All of these reactions were essential individually or together in identified MCSs with sizes greater than one. In summary, we could safely remove 210 out of 688 reactions by this analysis. 314 further reactions were individually essential, or essential as a set of reactions and were kept accordingly. The remainder of 164 reactions that were added during gap-filling procedures could not be associated to any MCS assuming hypoxic or normoxic growth conditions on otherwise unconstrained exchange reaction fluxes. Since we aimed to provide a flexible platform for *A. fumigatus* simulations we kept these reactions to allow for a diverse range of simulated diets that may be much more restricted than unconstrained media influx resulting in further diet-specific reactions essentiality. We added this information to our results and methods section accordingly:*

Results: “After these steps we analyzed again the consistency in our pan-GEM and identified 210 blocked reactions by flux variability analysis (FVA)¹⁸ based on relaxed flux bounds of exchange reactions.”

Methods: “Next, we analyzed re-introduced non-genome annotated reactions during curation to identify reactions that re-introduced metabolic redundancy after all curation steps were done. We identified 609 putative transport reactions between compartments and 79 further reactions occurring in only one compartment, which we analyzed further. 210 were blocked based on analyzing relaxed influx through all defined exchange reaction. The remainder of 478 reactions were analyzed for essentiality, that is, identifying reactions that require carrying flux to support a non-zero biomass value given relaxed metabolite influx. Towards this goal we conducted a minimal cut set analysis²¹ assuming hypoxic or normoxic growth conditions and otherwise again unconstrained exchange reaction fluxes. 314 reactions were individually essential, or essential as a set of reactions identified by MCS analysis. For 164 reactions the essentiality status remained unknown with relaxed exchange flux bounds and biomass optimization.”

2- given that NADH and NADPH are the main redox cofactors of catabolic and anabolic pathways respectively, it might be helpful to consider a weighting score for reactions with different cofactors as further selection criteria, this may potentially help to reduce the number of added reactions by reducing the number of redundant reactions.

Response: *We investigated the 79 non-genome associated reactions occurring only in one compartment, whether these are identical except for co-factors. Only two pairs of two reactions were identical except for NAD or NADP co-factors:*

Pair 1:

H₂O [cytoplasm] + NAD(+) [cytoplasm] + L-saccharopine [cytoplasm]  NADH [cytoplasm] + 2-oxoglutarate [cytoplasm] + L-lysine [cytoplasm]

H₂O [cytoplasm] + NADP(+) [cytoplasm] + L-saccharopine [cytoplasm] <=> NADPH [cytoplasm] + 2-oxoglutarate [cytoplasm] + L-lysine [cytoplasm]

Pair 2:

NAD(+) [cytoplasm] + L-arogenate [cytoplasm]  NADH [cytoplasm] + CO₂ [cytoplasm] + L-tyrosine [cytoplasm]

NADP(+) [cytoplasm] + L-arogenate [cytoplasm] <=> NADPH [cytoplasm] + CO₂ [cytoplasm] + L-tyrosine [cytoplasm]

Of note during EGC refinement efforts, we identified the reactions using NAD/NADH as causing a problem with their original bidirectional definition. To resolve the issue and keep the models viable we restricted reaction direction for both accordingly. To keep our models viable, and since both variants occur in the subsequently optimized strain GEMs and usage of NADH/NADPH did not generate major redundancies in our non-genome associated reactions, we kept for gap-filling purposes, both variants. We added a note to our methods mentioning this analysis:

Methods: “In addition, we investigated the 79 non-transport reactions for identity in metabolite conversion except co-factors. We identified only two pairs of reactions that differed in the use of co-factors NAD/NADH or NADP/NADPH, but had different directionality due to curation of energy generating cycles.”

3- checking the directionality of CHESHIRE-200 is a simple and doable analysis and may help to make the number of gaps fall into a reasonable range (less than 100). this could be done by checking whether any two reactions within CHESHIRE-200 have the same stoichiometry but different directionality. any two reactions that meet these conditions could be merged into a bi-directional reaction.

Response: *We investigated the necessity of reactions to be present in our network by running the deep learning based method CHESHIRE¹. To this end we used the tool to identify 200 reactions based on similarity score and investigated whether these possess different reaction directionality, but are otherwise stoichiometrically identical. We identified only one transport reaction for CoA between the compartments cytoplasm and mitochondria. R3150 is a unidirectional reaction from cytoplasm to mitochondria, which originated from the consensus yeast model during our initial merging of multiple draft GEMs. R3369, in contrast, is a bidirectional reaction and was added later during our automated gap-filling procedure. We deleted R3150 accordingly. Of note, reaction R3369 is present in all downstream derived strain GEMs of *A. fumigatus*.*

Next to the set of reactions suggested by CHESHIRE, we investigated the remainder of the gap-filling reactions for the same directionality issue. Together with R3369, we identified 35 reactions with directionality differences and otherwise stoichiometric identity. By providing these reactions as bidirectional reactions, we could remove 31 reactions from the pan-GEM and all derived strain-GEMs. We added a note to the respective methods section that we applied CHESHIRE to reduce redundancy in our pan-GEM and thus each derived strain-GEM:

Methods: "These were kept accordingly and we removed the 210 blocked reactions from our generic draft pan-GEM. We finally analyzed our pan-GEM for redundant reactions using CHESHIRE⁶³. We used the tool to identify 200 reactions based on similarity score and identified only one further internal transport redundancy for CoA, comprising two reactions with different directionality and resolved this by allowing bidirectional transport. No further redundant reaction was identified by CHESHIRE, which concluded our refinement efforts for our pan-GEM."

Reviewer #2 (Remarks to the Author):

I thank the authors for providing a thorough rebuttal to all the reviewers comments. The paper is much improved, and I acknowledge that a lot of work has gone into this, both initially and in the review process. I believe enough has been done, and I think the editors will be happy.

Personally, I am still confused by the omnilog section of the paper. Whilst the authors have indeed provided more clarity on this, it actually leads to more questions, the main one being, why the choice of redox dye D? This is a dye specifically for yeast, not filamentous fungi. In which case, maybe the FF plates would have been better for this particular part of the study?

Response: We appreciate the concern raised by the reviewer. In fact, during assay optimization, we tested and compared OD with redox dye D on the PM1 plate in two independent replicates with the reference strain Af293. As can be seen from Figure R2.1, the measured growth agrees very well between the two methods for most tested metabolites. Moreover, the addition of redox dye D specifically over other dyes was made after consultation with a Biolog company representative. These insights allowed us to use our OmniLog plate reader (which measures colorimetric change and not OD). The OmniLog plate reader was specifically designed for phenotypic microarrays and can measure up to 50 assay plates at once, allowing us to perform our experiments at the scale we required.

Comparison of OD and redox dye for *Aspergillus fumigatus* Biolog growth on various carbon sources (PM1)

Black is OD. Purple is redox dye. Each datapoint is the mean from 2 biological replicates. Data is scaled to allow comparison between OD and dye.

Figure R2.1: Comparison of OD and redox dye using the Biolog phenotypic microarray plate PM1 for various carbon sources.

Reviewer #3 (Remarks to the Author):

NCOMMS-22-25651A

Mirhakkak et al Genome-scale metabolic modeling of 252 *Aspergillus fumigatus* strains reveals growth dependencies on the lung microbiome

After having studied the point-to-point response, manuscript and supplement for more than a full-working day, I can say that I am impressed with the ambitions, scope and beauty of this work. For me, the strength resides in the Systems Biology part, i.e. the Pan-GEM reconstruction. However, unfortunately the experimental part still contains some major flaws that will backfire over time so that this work (as it is now) will not become a milestone of *Aspergillus* research.

Please take my critique seriously and do not look for excuses. If you cannot fix the errors, you may reorganize scope and contents of the manuscript and delete some parts.

Response: *All the suggestions from this reviewer in the first round were implemented to improve the quality of our manuscript. Running the annotation tools Centrifuge, KrakenUniq and EukDetect were suggested by the reviewer and the results were included in the revised version. Additional methods like FEAST were tested to support the microbiome annotation. It seems that the substantial additional analyses and a 30+ pages long point-by-point response during our revision to answer all reviewer concerns including the ones of reviewer 3 were seen by this reviewer as “excuses” but our efforts have been acknowledged by two reviewers who were impressed and happy by our revision. We continued in this second revision improving the manuscript by adding new analysis based on the reviewer’s recommendations (see below) and we remain optimistic that our updated results would be appreciated.*

Major

In my opinion, the most critical part is the CF airway metagenome study:

Sample collection. Lines 639 – 643: “Spontaneously expectorated sputum was collected during visits to the Cystic Fibrosis Center at the University Hospital Heidelberg and frozen in liquid nitrogen on the day of visit ... Samples were frozen within 24 hr after reception at the microbiology department.”

1. First, spontaneously expectorated sputum will always be contaminated by oral microbiota. This is not problematic for routine culture-dependent diagnostics of the typical CF pathogens, but it is inappropriate for any in-depth metagenome analysis that includes commensals. Induced sputum is the best compromise to minimize contamination (Weiser et al. The lung microbiota in children with cystic fibrosis captured by induced sputum sampling. *J Cyst Fibros.* 2022;21:1006-1012; Ronchetti K et al. The CF-Sputum Induction Trial (CF-SpIT) to assess lower airway bacterial sampling in young children with cystic fibrosis: a prospective internally controlled interventional trial. *Lancet Respir Med.* 2018;6:461-471).

Response: *We agree with the reviewer that it is important to consider oral microbiota contamination for both spontaneously expectorated sputum and induced sputum, but disagree that induced sputum can always minimize contamination. To check for the level of possible oral microbiota contamination, we conducted the FEAST (Shenhav, et al., *Nature Methods*, 2019)² analysis as in the first revision and found out: (1) when using the dataset from*

Feigelman et al. (2017)³, where both induced sputum and spontaneous sputum were used, the contribution of oral microbiome was significantly higher in induced sputum samples than in spontaneous sputum samples (see Figure R3.1a below); (2) when comparing the contribution of oral microbiome in our samples against the induced sputum samples published in a Nature Publishing Group journal (Yan et al, Nature Microbiology, 2022)⁴, our samples had significantly lower contamination from oral microbiome ($p=0.05$, see Figure R3.1b below).

Moreover, both, Weiser et al. (2022) and Ronchetti et al. (2018), focus on investigating the best sampling technique for **young children** and compare BAL with induced sputum, the latter being the more tolerable and thus preferred method^{5,6}. However, the majority of our samples were not derived from patients <10 years, since it was not our aim to identify the best sampling technique in children. The samples obtained for this study are part of a study-cohort comparing NGS diagnostic to gold standard culture. Expecterated sputum is the most common sample from the lower airway in microbiological diagnostics in cystic fibrosis. All visits were routine visits and therefore induction was not required. Furthermore, other studies have shown that the microbiome of induced sputum and expecterated sputum are similar and are both suitable for microbiome studies^{3,7,8}, as being applied in studies recently published in prestigious journals from the Nature Publishing Group (Aogáin et al, Nature Medicine, 2021).

Figure R3.1: FEAST results using showing source proportion of oral microbiome. a) data from Feigelman et al., 2017³. b) data from Yan et al., 2022⁴.

2.Second, the composition of microbial communities in CF respiratory secretion shifts within minutes. Hence, samples must be frozen asap at -80°C or liquid nitrogen, i.e. within less than a minute.

Response: In our study, flash freezing on site was not possible and would have rendered culture impossible for determining *A. fumigatus* colonization. Performing necessary work/experiments before freezing the samples was also seen in another study into airway microbiome via sputum sampling (Yan et al., Nature Microbiology, 2022)⁴. Furthermore, in another study published in an NPG journal (Aogáin et al, Nature Medicine, 2021)⁹, transport of specimens was also involved which was completed within 4 hours instead of minutes. We appreciate that it is ideally the best practice to freeze samples as soon as possible. Yet realistically this is not achievable and next to our encouraging FEAST results it has been

sufficiently shown that sending samples within hours up to a day followed by immediate freezing in liquid nitrogen is an adequate and feasible compromise for microbiome studies.

Processing and evaluation of sequence data. Line 692-693: 'BWA (v07.17) was used to align quality filtered reads to the human reference 693 genome (hg38) for removal of human-derived reads.'

3. BWA is an adequate aligner for any study on human genomic DNA, but it is inappropriate for metagenome studies, particularly if the reads are longer than 50 bp. The aligner will clip sequence reads and the generated short k-mers will give an explosion of false-positive assignments. (You may test this behaviour by the comparison of 50, 70, 100, 150, 200 bp long reads of the same metagenome dataset.) Recommendation: The primary data should be re-evaluated by another aligner.

Response: We respectfully disagree with the reviewer that BWA is inappropriate for metagenome studies. There are many pipelines and numerous recent studies of metagenomics that use BWA **to remove host reads** as we did¹⁰⁻¹². A recently published protocol article in *Nature Protocols* by Lu et al. (2022) also stated that the Bowtie 2 tool used to filter host reads can be replaced with similar read alignment tools such as BWA-MEM or minimap2¹³.

Furthermore, Bush et al. (2020) compared the methods for detecting human reads in metagenomic sequencing datasets and show that the most sensitive methods of human read detection is BWA¹⁴. For our study of airway microbiome, BWA is thus more than appropriate for detecting human reads.

4. Figure 4b. I counted 18 species that have never been detected by others in the CF airway microbiome. In other words, the taxonomic classification of your datasets needs a rigorous re-classification.

Response: Following the reviewers' suggestions, we used *raspir*, which is also used in the *Wochenende* pipeline, to flag and eliminate the microbial species with non-uniform read distributions^{15,16}. As a result, we retained only *raspir*-confirmed 200 species in our dataset and updated all results in our manuscript. We then constructed a new network using *raspir*-confirmed species (Figure 4b), and found that module 3 is significantly correlated with Valine, Phenylalanine and Tryptophan related pathway, as well as with FEV1. Therefore, our main conclusion for network analysis remains the same: "The associated metabolic functions enriched in differential correlation microbial modules pinpointed again towards amino acid, particularly aromatic amino acid pathways, but also fatty acid, nitrogen and sulfur metabolic pathways, suggesting that lung microbiome metabolic activity is reshaped in the presence of *A. fumigatus*". This also demonstrates the robustness of our network analysis.

In addition, we also examined whether the species in the network ($n=124$) have been detected by previous studies into CF airway microbiome (Pust et al., 2020, Feigelman et al., 2017, Dmitrijeva et al., 2021)^{3,17,18}. We obtained taxa results from these three independent airway metagenomics datasets using the same tools and parameters as used in our study (including *raspir*). From the 124 species, 89 of them (72%) were also found in at least one of those studies (Figure R3.4a); for the remaining 35 species, 12 were able to colonize the airway as indicated by culture studies while the final 23 were also detected in the human airway microbiome of COPD patients (Yan et al., *Nature Microbiology*, 2022)⁴.

Figure R3.4a: Comparison of taxa used in our network analysis and those detected in three independent airway metagenomics datasets^{3,17,18}.

Notably, while our study has a unique set of 35 species (28.2% of total 124) that have not been detected by those 3 studies, we found this is common to each study/cohort (see Figure R3.4b). For example, the study by Dmitrijeva et al. (2021) also contains a unique set of 49 species (28.5% of total 172).

Figure R3.4b: Comparison of taxas detected in three independent airway metagenomics datasets^{3,17,18}.

5. Table S6. Of the 596 species I counted more than 150 implausible taxa. In addition, I noted an inflation of taxa among the abundant genera, probably due to false-positive alignment of homologous reads. The latter argument particularly applies to the genera *Pseudomonas*, *Stenotrophomonas* and *Neisseria*. You may easily curate your dataset by eliminating all microbial species with non-uniform read distributions.

Response: Following the reviewers' suggestions, we used *raspir* to eliminate the microbial species with non-uniform read distributions and updated our taxonomic results (updated Table S6). Based on the curated dataset (by using *raspir*), the top 10 abundant species and genera are highly similar to our previous results, with 8 out of 10 abundant genera being the same and all top 10 abundant species being exactly the same. For the three genera mentioned by the reviewer, *Pseudomonas* and *Stenotrophomonas* were excluded while *Neisseria* remained in the top-10 list. Moreover, consistent with the previous results, we did not observe a significant difference in alpha or beta diversity. We adapted our methods sections accordingly:

“To further control false positive rate, *raspir*(v1.0.2)³⁰ was used to support our Kraken 2 results. Only the species that can be detected in at least one samples by *raspir* were kept (Supplementary Table S6).”

6. You write in your response to point 20: “To further interrogate our results, we obtained taxa results from three independent airway metagenomics dataset using the same tools and parameters that we used in this paper (Pust et al., 2020, Feigelman et al., 2017, Dmitrijeva et al., 2021). Clustering of samples by PCoA based on the beta diversity (Bray-Curtis distance) of our samples and other datasets showed that our dataset shared similar microbial structure and composition with other airway metagenomes (Fig. R3.20a below). It also turned out that the percentage of 598 species that we detected in other datasets is very high (Pust dataset: 94.48%, Feigelman dataset: 93.14%, Dmitrijeva dataset: 94.65%).” In case of the Pust dataset 94.48% of 598 species are 565 species. Pust’s original paper (Supplementary Table S2 published in *Comput. Struct. Biotechnolog.* 2021, 20:176-186), however, lists just 257 species. Please contain the inflation of species generated by false-positive alignment by eliminating all microbial species with non-uniform read distributions.

Response: As stated in our response to comment 4-5, we have updated the taxonomic profiling result after using *raspir* to flag false-positive alignments. Now we retained only *raspir*-confirmed 200 species in our dataset, which still showed a high degree of overlap with the Pust et al. dataset¹⁸ as was reported in <https://github.com/mmpust/airway-metagenome-infants>.

After obtaining taxa results from it and two further independent airway metagenomics datasets using the same tools and parameters as used in our study (including the use of *raspir*), the ratio of overlap species remains high, despite each study showing a unique set of identified taxa (as already detailed out above in comment 4, Figure R3.6).

Figure R3.6: Intersection of our taxonomic profiling with data from Pust et al. (2020)¹⁸.

7. I am struggling with the meaning of the ‘targeted metabolomics’ data. What do this data of amino acid content in the supernatant tell us about the cellular amino acid metabolism? Some amino acids were apparently not detected, whereas others showed high coefficients of variation. I missed the information about the number of biological and technical replicates. This part is probably dispensable.

Response: One of our findings when we reconstructed the 252 strain-specific models was the high variability observed in the amino acid potential (see section “*A. fumigatus* strains show notable accessory reaction content”). Therefore, we cultivated 20 *A. fumigatus* strains on *A. fumigatus* minimal media. By acquiring and analyzing targeted amino acids found in the resulting supernatants (as a proxy of amino acid metabolism) after 24 hr, we found a

considerable variability of metabolites that reflect the metabolic variability of at most 40% shared metabolic reactions across all strain-GEMs. There were performed during our last revision as requested by one reviewer, and we added this information to Supplementary Table S4 and in our results:

“The large variability among strains in amino acid metabolism was confirmed by cultivation and targeted metabolomics profiling of 20 A. fumigatus strains (Supplementary Table S4).”

Minor

8.Line 118. You mention that the first draft model included 3,233 exchange reactions, but Figure 1b (probably representing the final model) indicates only 239 exchange reactions. Please clarify.

Response: *The draft model was generated by merging the Aspergillus models reconstructed by the CoReCo pipeline¹⁹. These models originally contain exchange reactions for every compound in the network. Therefore, we started with a model containing 3233 exchange reactions. However, after the model curations including the model adjustment to the phenotypic growth data, we ended up with 239 exchange reactions and removed the rest which are not necessary with respect to our collected data. We clarified this in the methods section:*

“Of note, the high number of initial exchange reactions originated from models created with the CoReCo pipeline, which includes an exchange reaction for each defined metabolite by default.”

9.Figure 3c. The decision tree deserves some more interpretation in the text.

Response: *We extended our description in the text as follows:*

“Specifically, the ability to add sulfur to methionine as well as the absence of the ability to convert selenocystathione to selenocysteine or tryptamine to Indole-3-acetaldehyde appeared a characteristic of environmental strains, which in part was not present in clinical strains (Fig. 3c). This may hint to altered Thioredoxin levels, which have been linked to the fungus' redox homeostatis before²⁷.”

10.Table S7. The inclusion of clinical metadata is appreciated. However, first, please explain why you can provide paired spirometry and BMI data for only 14 and 26 of the 40 study participants, respectively. The CF clinic Heidelberg receives high financial support for its patient registry; hence, complete data should be available. According to guidelines, spirometry is obligatory at each patient's visit of the CF center. Second, and more importantly, I do not see a clinical signal of the detection of A. fumigatus in the patient's respiratory secretion. FEV1 was better in 8 patients at the day of the Af-negative sample and better in 8 patients at the day of the Af-positive sample. Likewise, BMI was higher in 8 patients at the day of the Af-negative sample and higher in 18 patients at the day of the Af-positive sample. Apparently, the investigated dataset does not show any association between Af-status and clinical status. This observation is consistent with clinical experience that the detection of Af needs to be interpreted in the light of clinical symptoms and Af serology. Your text avoids any claims expressis verbis between presence/absence of Af and clinical status, but you should state more clearly in the text that Af modifies the lung microbiome, but no consequences for the host

– microbe interaction can yet be deduced from your data. Your title is sexy, but evokes expectations that cannot be fulfilled.

Response: *The missing data from the clinical report were missing due to the non-inclusion in the digital database. The data were now retrieved from the paper documents. BMI is now available from 57/80 visits; spirometry is available for 77/80 visits. The missing values for the spirometry are due to the inability to perform the spirometry at the time of the visit. The new data have been added and corrected in the manuscript. We agree with the reviewer that *A. fumigatus* infection should be validated by serology. Yet our study aimed to discover the impact of *A. fumigatus* presence on the colonisation of the lung microbiome. Therefore, we did not dissociate infection from colonization and unified our description to *A. fumigatus* colonisation throughout our manuscript.*

11. Supplementary Tables. For most tables, the number of the Table is not given in the description. Please rectify.

Response: *We added the table numbers of each Supplementary Table in its respective description sheet.*

References

1. Chen, C., Liao, C. & Liu, Y.-Y. Teasing out Missing Reactions in Genome-scale Metabolic Networks through Graph Convolutional Networks. *bioRxiv* 2022.06.27.497720 (2023) doi:10.1101/2022.06.27.497720.
2. Shenhav, L. *et al.* FEAST: fast expectation-maximization for microbial source tracking. *Nature Methods* 2019 16:7 16, 627–632 (2019).
3. Feigelman, R. *et al.* Sputum DNA sequencing in cystic fibrosis: Non-invasive access to the lung microbiome and to pathogen details. *Microbiome* 5, 1–14 (2017).
4. Yan, Z. *et al.* Multi-omics analyses of airway host-microbe interactions in chronic obstructive pulmonary disease identify potential therapeutic interventions. *Nat Microbiol* 7, 1361–1375 (2022).
5. Ronchetti, K. *et al.* The CF-Sputum Induction Trial (CF-SpIT) to assess lower airway bacterial sampling in young children with cystic fibrosis: a prospective internally controlled interventional trial. *Lancet Respir Med* 6, 461–471 (2018).
6. Weiser, R. *et al.* The lung microbiota in children with cystic fibrosis captured by induced sputum sampling. *Journal of Cystic Fibrosis* 21, 1006–1012 (2022).
7. Zemanick, E. T. *et al.* Assessment of airway microbiota and inflammation in cystic fibrosis using multiple sampling methods. *Ann Am Thorac Soc* 12, 221–229 (2015).
8. Shaw, J. *et al.* Sputum biospecimen quality and suitability for microbiome characterisation in COPD. *European Respiratory Journal* 52, PA5304 (2018).
9. Mac Aogáin, M. *et al.* Integrative microbiomics in bronchiectasis exacerbations. *Nature Medicine* 2021 27:4 27, 688–699 (2021).
10. Clarke, E. L. *et al.* Sunbeam: an extensible pipeline for analyzing metagenomic sequencing experiments. *Microbiome* 7, (2019).
11. Moraitou, M. *et al.* Ecology, Not Host Phylogeny, Shapes the Oral Microbiome in Closely Related Species. *Mol Biol Evol* 39, (2022).
12. Chen, C. *et al.* Expanded catalog of microbial genes and metagenome-assembled genomes from the pig gut microbiome. *Nat Commun* 12, (2021).
13. Lu, J. *et al.* Metagenome analysis using the Kraken software suite. *Nature Protocols* 2022 17:12 17, 2815–2839 (2022).
14. Bush, S. J., Connor, T. R., Peto, T. E. A., Crook, D. W. & Walker, A. S. Evaluation of methods for detecting human reads in microbial sequencing datasets. *Microb Genom* 6, 5–18 (2020).
15. Pust, M.-M. & Tümmler, B. Identification of core and rare species in metagenome samples based on shotgun metagenomic sequencing, Fourier transforms and spectral comparisons. *ISME Communications* 2021 1:1 1, 1–4 (2021).
16. Rosenboom, I. *et al.* Wochenende — modular and flexible alignment-based shotgun metagenome analysis. *BMC Genomics* 23, 1–12 (2022).
17. Dmitrijeva, M. *et al.* Strain-resolved dynamics of the lung microbiome in patients with cystic fibrosis. *mBio* 12, 1–20 (2021).

18. Pust, M. M. *et al.* The human respiratory tract microbial community structures in healthy and cystic fibrosis infants. *npj Biofilms and Microbiomes* 2020 6:1 6, 1–10 (2020).
19. Castillo, S. *et al.* Whole-genome metabolic model of *Trichoderma reesei* built by comparative reconstruction. *Biotechnol Biofuels* 9, 252 (2016).

REVIEWERS' COMMENTS

Reviewer #1 (Remarks to the Author):

I am pleased to acknowledge the authors' significant efforts in addressing the comments made during the previous review process. The manuscript has been substantially improved, and I commend the authors for their thoroughness and dedication. Their work undoubtedly contributes to the field and holds great potential for impacting related research areas. Their study holds great potential for advancing our understanding of the organism's metabolism. The work is significant, compares well to the established literature, supports the conclusions and claims made, and meets the expected standards in the field. Therefore, I strongly recommend its publication.

Reviewer #2 (Remarks to the Author):

I personally think the paper is much improved from the initial reading months ago - it is certainly clearer and easier to read (making it more enjoyable).

I don't think my previous comments regarding making the omnilog section clearer have really been addressed though. I mentioned why use the redox dye and not FF plates. I appreciate that this was after consultation with a representative from the omnilog company, and the comparison to OD has been done, but doesn't address my original comment of why not use FF plates, which are specific for filamentous fungi (as currently recommended on omnilog website)? Does the redox dye get metabolised by the fungi and therefore influence result?

I appreciate re-doing this experiment could be costly (both time and financially) so maybe mentioning in the discussion that this is a consideration. Or maybe it's an editorial decision to do an additional experiment just to check redox dye does not differ from FF plate.

Reviewer #4 (Comments to the editors. Keep confidential.)

Dear editors,

Below please find my opinion regarding Reviewer #3's comments and the author's response.

Major critique 1 - sample collection in not induced sputum.

The studies about induced sputum have been published to show this as a reasonable alternative to bronchoalveolar lavage in understanding the lung microbiome. Sputum likely has contamination of upper airway flora (see Jorth et al, Cell Reports 2019, PMID 31018133; Lu et al, mSystems 2020, PMID 32636336). The influence of this on microbiome determination remains debated (see two prior articles). However, this is true for both induced and spontaneously expectorated sputum. Thus, here I would agree with the authors.

Major critique 2 - sputum samples require immediate freezing.

Here, I again agree with the authors. Lipuma and colleagues have published a number of articles using DNA isolated from samples kept for up to a month at 4 degrees, and have not seen significant variation in results (Caverly et al, Ann Am Thorac Soc 2019, PMID 31415187 as an exemplar).

Major critique 3 - BWA is inappropriate for metagenomic studies.

BWA has been published as a tool used in the metagenomic analysis of the human microbiome. No tool has been clearly established as the gold standard at this point, and it is unlikely that the aligner would be a large confounding factor (Miossec et al., PeerJ 2020, PMID 32864214). I would again side with the authors.

Major critiques 4-6 - the authors report implausible taxa.

I agree with the reviewer that low level taxa in a dataset that have not been previously reported are likely not truly present in the sample. However, this is the nature of the bioinformatic tools we have available at the moment. I think the authors have appropriately addressed the reviewer's concerns in doing their best to control for these factors and present the cleanest data possible.

Major critique 7 - addressed already.

Based on these assessments, I would recommend publication of this manuscript.

Point by point response

Reviewer #2

I personally think the paper is much improved from the initial reading months ago - it is certainly clearer and easier to read (making it more enjoyable).

Response: *We are grateful and appreciate the reviewer acknowledges our efforts to improve our manuscript.*

I don't think my previous comments regarding making the omnilog section clearer have really been addressed though. I mentioned why use the redox dye and not FF plates. I appreciate that this was after consultation with a representative from the omnilog company, and the comparison to OD has been done, but doesn't address my original comment of why not use FF plates, which are specific for filamentous fungi (as currently recommended on omnilog website)? Does the redox dye get metabolised by the fungi and therefore influence result?

Response: *We appreciate the reviewer's continued attention to the issue regarding the use of the Omnilog section. Upon further investigation, we believe it is a miscommunication regarding the products being discussed and utilised in this study. The confusion appeared to arise from the fact that we and the reviewer are referring to two different Biolog products.*

We believe that the reviewer is referring to the Biolog "FF Filamentous Fungi Identification Test Panel." This is a single plate designed for the identification of fungi based on their growth patterns on some selected substrates. However, in our study, we employed the Biology Phenotypic Microarray plates, which consist of a series of 25 plates designed for testing various metabolic and chemical sensitivity traits (we used plates PM1-4). Of note, the Phenotypic Microarray plates we used do not contain the redox dye, while the FF ID plates do. Consequently, we followed the manufacture's protocol for using the PM plates with filamentous fungi, including resuspending conidia in FF media and adding redox dye to allow measurement in the Omnilog colorimetric plate reading device.

We apologize for any confusion caused by the lack of clarity in our previous response.

I appreciate re-doing this experiment could be costly (both time and financially) so maybe mentioning in the discussion that this is a consideration. Or maybe it's an editorial decision to do an additional experiment just to check redox dye does not differ from FF plate.

Response: *We clarify our methodology further in the respective methods section of the phenotypic microarrays as follows:*

"Phenotypic microarrays were performed using Biolog Phenotypic Microarray plates PM1, PM2, PM3, and PM4 (Biolog Inc., Hayward, CA, USA) prepared following the manufacturer's protocol for filamentous fungi, including resuspending conidia in filamentous fungi (FF) media and the addition of 0.16 ml of Biolog Redox Dye D to the master mix of each plate to quantify fungal metabolic activity. By using PM1-PM4 we could investigate 379 different growth conditions rather than only 95 as available with Biolog's FF plate (Fungi Identification Test Panel)."